# Fully Unconstrained Online Learning

**Ashok Cutkosky**
Boston University
ashok@cutkosky.com

**Zakaria Mhammedi**
Google Research
mhammedi@google.com

## Abstract

We provide a technique for online convex optimization that obtains regret $G\|w_\star\|\sqrt{T \log(\|w_\star\|G\sqrt{T})} + \|w_\star\|^2 + G^2$ on $G$-Lipschitz losses for any comparison point $w_\star$ without knowing either $G$ or $\|w_\star\|$. Importantly, this matches the optimal bound $G\|w_\star\|\sqrt{T}$ available *with* such knowledge (up to logarithmic factors), unless either $\|w_\star\|$ or $G$ is so large that even $G\|w_\star\|\sqrt{T}$ is roughly linear in $T$. Thus, at a high level it matches the optimal bound in all cases in which one can achieve sublinear regret.

## 1 Unconstrained Online Learning

This paper provides new algorithms for *online learning*, which is a standard framework for the design and analysis of iterative first-order optimization algorithms used throughout machine learning. Specifically, we consider a variant of online learning often called "online convex optimization" [1, 2]. Formally, an online learning algorithm is designed to play a kind of "game" between the learning algorithm and the environment, which we can describe using the following protocol:

---
**Protocol 1.** Online Learning/Online Convex Optimization.
**Input:** Convex domain $\mathcal{W} \subseteq \mathbb{R}^d$, number of rounds $T$.
For $t = 1, \ldots, T$:

1. Learner outputs $w_t \in \mathcal{W}$.
2. Nature reveals loss vector $g_t \in \partial\ell_t$ for some convex function $\ell_t : \mathcal{W} \to \mathbb{R}$ to the learner.
3. Learner suffers loss $\langle g_t, w_t \rangle$.

The learner is evaluated with the *regret* $\sum_{t=1}^{T}(\ell_t(w_t) - \ell_t(w_\star))$ against comparators $w_\star \in \mathcal{W}$. By convexity, the regret is bounded by the *linearized* regret $\sum_{t=1}^{T}\langle g_t, w_t - w_\star \rangle$. Our goal is to ensure that for all $w_\star \in \mathcal{W}$ simultaneously:

$$\text{Regret}_T(w_\star) := \sum_{t=1}^{T}\langle g_t, w_t - w_\star \rangle \underbrace{\leq}_{\text{goal}} \widetilde{O}\left(\|w_\star\|\sqrt{\sum_{t=1}^{T}\|g_t\|^2}\right). \tag{1}$$

---

Qualitatively, we consider a learner to be performing well if $\frac{1}{T}\sum_{t=1}^{T}(\ell_t(w_t) - \ell_t(w_\star))$ is very small, usually going to zero as $T \to \infty$. This indicates that the average loss of the learner is close to the average loss of any chosen comparison point $w_\star \in \mathcal{W}$. This property is called "sublinear regret". The bound (1) is unimprovable in general [3, 4, 5], and clearly implies sublinear regret.

Algorithms that achieve low regret are used in a variety of machine learning applications. Perhaps the most famous such application is in the analysis of stochastic gradient descent, which achieves (1) for appropriately tuned learning rate [6]. More generally, stochastic convex optimization can be reduced

38th Conference on Neural Information Processing Systems (NeurIPS 2024).

to online learning via the *online to batch conversion* [7]. Roughly speaking, this result says that an online learning algorithm that guarantees low regret can be immediately converted into a stochastic convex optimization algorithm that converges at a rate of $\frac{\mathbb{E}[\text{Regret}_T(w_\star)]}{T}$, where $w_\star$ is the minimizer of the objective. Online learning can also be used to solve non-convex optimization problems [8] and can even be used to prove concentration inequalities [9, 10, 11, 12]. In all of these cases, achieving the bound (1) produces methods that are optimal for their respective tasks. Thus, it is desirable to be able to achieve (1) in as robust a manner as possible.

Our goal is to come as close as possible to achieving the bound (1) while requiring minimal prior user knowledge about the loss sequence $g_1, \ldots, g_t$ and $w_\star$. In the past, several prior works have achieved the bound (1) when given prior knowledge of either $\|w_\star\|$ or $\max_t \|g_t\|$ [13, 14, 15, 16, 17, 18, 19, 20, 21]. However, such knowledge is frequently unavailable. Instead, many problems are "fully unconstrained" in the sense that we do not have any reasonable upper bounds on either $\|w_\star\|$ or $\max_t \|g_t\|$. In particular, when considering the application to stochastic convex optimization, the values for $\|w_\star\|$ and $\max_t \|g_t\|$ can be interpreted as knowledge of the correct learning rate for stochastic gradient descent [6]. Thus, achieving the bound (1) with less prior knowledge roughly corresponds to building algorithms that are able to achieve optimal convergence guarantees without requiring manual hyperparameter tuning. For this reason, it is common to refer to such algorithms as "parameter-free". This paper focuses on this difficult but realistic setting.

**Our new upper bound.** Unfortunately, the bound (1) is actually unobtainable in general without prior knowledge of either the magnitude $\|w_\star\|$ or the value of $\max_t \|g_t\|$ [18, 22]. Nevertheless, we will obtain a new compromise bound. For any user-specified $\gamma > 0$, our method will achieve:

$$\sum_{t=1}^{T} \langle g_t, w_t - w_\star \rangle \leq \widetilde{O}\left( \max_{t \in [T]} \|g_t\|^2 / \gamma + \gamma \|w_\star\|^2 + \|w_\star\| \sqrt{\sum_{t=1}^{T} \|g_t\|^2} \right). \tag{2}$$

To dissect this compromise, let us consider the case $\|g_t\| = G$ for all $t$ and $\gamma = 1$. In this situation, our bound (2) is roughly $G^2 + \|w_\star\|^2 + \|w_\star\| G\sqrt{T}$, while the "ideal" bound (1) is merely $\|w_\star\| G\sqrt{T}$. However, for our bound to be significantly worse than (1), we must have either $G \geq \|w_\star\| \sqrt{T}$ or $\|w_\star\| \geq G\sqrt{T}$. In either case, we might expect that $\|w_\star\| G\sqrt{T}$ is roughly $\Omega(T)$ (assuming that neither $G$ nor $\|w_\star\|$ is very small). So, intuitively the only cases in which our bound is worse than the ideal bound are those for which the ideal bound is already rather large—the problem is in some sense "too hard".

**Comparison with previous bounds** Our bound (2) is not the first attempted compromise in our fully unconstrained setting. Prior work [18, 23] instead provides the bound:

$$\sum_{t=1}^{T} \langle g_t, w_t - w_\star \rangle \leq \widetilde{O}\left( \frac{1}{\gamma} \sqrt{\max_{t' \in [T]} \|g_{t'}\| \cdot \sum_{t=1}^{T} \|g_t\|} + \gamma^2 \max_{t \in [T]} \|g_t\| \|w_\star\|^3 + \|w_\star\| \sqrt{\sum_{t=1}^{T} \|g_t\|^2} \right). \tag{3}$$

In fact, readers familiar with this literature may be surprised that our bound is even possible; [18] show that the bound (3) is optimal for the fully-unconstrained case. However, the lower-bound provided by [18] actually has a small loophole; it only applies to algorithms that insist on a *linear* dependence on $\max_t \|g_t\|$. Our method avoids this lower bound by instead suffering a quadratic dependence on $\max_t \|g_t\|$.

While our new bound (2) does not uniformly improve the prior bound (3), it has several qualitative differences that may be more appealing.

1. The bound (3) does *not* have the desirable property outlined above for our new bound; for $\|g_t\| = G$, it is possible for the bound (3) to be much greater than the ideal bound (1) even when (1) is small.

2. The dependency on the user-specified value $\gamma$ is arguably more sensitive; in (3), decreasing $\gamma$ comes at a $\frac{1}{\gamma} G\sqrt{T}$ cost while increasing gamma comes at an $\gamma^2 \|w_\star\|^3$ cost. In contrast, in our bound (3), the $\gamma$-dependencies are milder; $G^2/\gamma$ for decreasing $\gamma$ (which does not depend on $T$) and $\gamma \|w_\star\|^2$ for increasing $\gamma$.

3. The previous compromise bound (3) has a term that depends on $\max_t \|g_t\| \left( \sum_{t=1}^{T} \|g_t\| \right)$ rather than $\sum_{t=1}^{T} \|g_t\|^2$. The dependence on the second power of $\|g_t\|$ is sometimes referred to as a

"second-order" bound and is known to imply *constant* regret in certain settings [1, 24] (so-called "fast rates").

4. Consider the case that both bounds are tuned with their respective "optimal" values for $\gamma$. Our new bound would then reduce to $\tilde{O}\left(\|w_\star\|\sqrt{\sum_{t=1}^T \|g_t\|^2} + \|w_\star\|G\right)$, while the previous bound would instead become $\tilde{O}\left(\|w_\star\|\sqrt{\sum_{t=1}^T \|g_t\|^2} + \|w_\star\|G^{2/3}\left(\sum_{t=1}^T \|g_t\|\right)^{1/3}\right)$. Thus, our new bound appears more desirable even with individually optimal tuning.

5. Our bound ensures that when $w_\star = 0$, the dependence on $T$ is $O(1)$. This has a number of useful consequences. For example, by running a separate instance of our algorithm for each dimension of a $d$-dimensional problem, we can achieve:

$$\sum_{t=1}^T \langle g_t, w_t - w_\star \rangle \leq \widetilde{O}\left(\frac{1}{\gamma}\sum_{i=1}^d \max_{t\in[T]} g_{t,i}^2 + \gamma\|w_\star\|_2^2 + \sum_{i=1}^d |w_{\star,i}|\sqrt{\sum_{t=1}^T g_{t,i}^2}\right).$$

Attempting this with the bound (3) would incur a more significant dependence on the dimension $d$. More generally, this property means that our bound fits into the framework for "combining" regret bounds of [25].

## 2   Notation

Throughout this paper, we use $\mathcal{W}$ to refer to a convex domain contained in $\mathbb{R}^d$. Our results can in fact be extended to Banach spaces relatively easily using the reduction techniques of [16], but we focus on $\mathbb{R}^d$ here to keep things more familiar. We use $\|\cdot\|$ to indicate the Euclidean norm. Occasionally we also make use of other norms—these will always be indicated by some subscript (e.g. $\|\cdot\|_t$). We use $\mathbb{R}_{\geq 0}$ to indicate the set of non-negative reals. For a convex function $F$ over $\mathbb{R}^d$, the Fenchel conjugate of $F$ is $F^\star(\theta) = \sup_{x\in\mathbb{R}^d}\langle\theta, x\rangle - F(x)$. We occasionally make use of a "compressed sum" notation: $g_{a:b} := \sum_{t=a}^b g_t$. We use $O$ to hide constant factors and $\widetilde{O}$ to hide both constant and logarithmic factors. All proofs not present in the main paper may be found in the appendix.

We will refer to the values $g_t$ provided to an online learning algorithm interchangeably as "gradients", "feedback" and "loss" values. We will refer to online learning algorithms occasionally as either "learners" or just "algorithms".

## 3   Overview of Approach

Our overall approach to achieve (2) is a sequence of reductions. As a first step, we observe that it suffices to achieve our goal in the special case $\mathcal{W} = \mathbb{R}$. Specifically, [16] Theorems 2 and 3 reduce the general $\mathcal{W}$ case to $\mathcal{W} = \mathbb{R}$ case. We provide an explicit description of how to apply these reductions in Section C. So, we focus our analysis on the case $\mathcal{W} = \mathbb{R}$. Next, we reduce the problem to a variant of the online learning protocol in which we also must contend with some potentially non-Lipschitz regularization function (Section 3.1). Finally, we show how to achieve low regret in this special regularized setting (Section 3.3).

### 3.1   Hints and Regularization

Our bound is achieved via a reduction to a variant of Protocol 1 with two changes. First, the learner is provided with prior access to *magnitude hints* $h_t \in \mathbb{R}$ that satisfy $\|g_t\| \leq h_t$. This notion of magnitude hints is also a key ingredient in the previous bound (3). Our second change is that the loss is not only the linear loss $\langle g_t, w \rangle$, but a *regularized* non-linear loss $\langle g_t, w \rangle + a_t\psi(w)$ for some fixed function $\psi : \mathcal{W} \to \mathbb{R}_{\geq 0}$ that we call a "regularizer". Formally, this protocol variant is specified in Protocol 2.

> **Protocol 2.** Regularized Online Learning with Magnitude Hints.
> **Input:** Convex function $\psi : \mathcal{W} \to \mathbb{R}_{\geq 0}$.
> For $t = 1, \ldots, T$:
>
> 1. Nature reveals magnitude hint $h_t \geq h_{t-1} \geq 0$ to the learner.
> 2. Learner outputs $w_t \in \mathcal{W}$.
> 3. Nature reveals loss $\tilde{g}_t$ with $\|\tilde{g}_t\| \leq h_t$ and $a_t \in [0, \gamma]$ to the learner.
> 4. Learner suffers loss $\langle \tilde{g}_t, w_t \rangle + a_t \psi(w_t)$.
>
> The learner is evaluated with the *regularized regret* $\sum_{t=1}^T \langle \tilde{g}_t, w_t - w_\star \rangle + a_t \psi(w_t) - a_t \psi(w_\star)$. The goal is to obtain:
>
> $$\sum_{t=1}^T \langle \tilde{g}_t, w_t - w_\star \rangle + a_t \psi(w_t) - a_t \psi(w_\star) \underbrace{\leq}_{\text{goal}} \widetilde{O}\left( \|w_\star\| \sqrt{h_T^2 + \sum_{t=1}^T \|\tilde{g}_t\|^2} + \psi(w_\star) \sqrt{\gamma^2 + \sum_{t=1}^T a_t^2} \right). \tag{4}$$

In the special case that $\psi(w) = 0$ (i.e. the $a_t$ are irrelevant, or all 0), then various algorithms achieving the desired bound (4) are available in the literature [18, 21, 23, 26]. We provide in Algorithm 3 a new algorithm for this situation that achieves the optimal logarithmic factors—there is in fact a pareto-frontier of incomparable bounds that differ in the logarithmic factors. [26] provides the first algorithm to reach this frontier, while our method can achieve all points on the frontier[1]. We include this result because it is of some independent interest, but it not the major focus of our contributions. Any of the prior work in this area would roughly suffice for our broader purposes; the difference is only in the logarithmic terms.

**Challenge of achieving (4).** Achieving the bound (4) is challenging when $\|w_\star\|$ is not known ahead of time. To see why, let us briefly consider two potential solutions.

The most immediate approach might be to reduce Protocol 2 to the case in which $a_t = 0$ for all $t$ by replacing $\tilde{g}_t$ with $\tilde{g}_t + a_t \nabla \psi(w_t)$, and then possibly modifying the magnitude hint $h_t$ in some way to now be a bound on $\|\tilde{g}_t\|$. However, this approach is problematic because the expected bound would now depend on $\sum_{t=1}^T \|\tilde{g}_t + a_t \nabla \psi(w_t)\|^2$ rather than $\sum_{t=1}^T \|\tilde{g}_t\|^2$ and $\sum_{t=1}^T a_t^2$. This means that the naive regret bound would be very hard to interpret as $w_t$ would appear on both the left and right hand sides of the inequality.

Another possibility is a follow-the-regularized leader/potential-based algorithm, making updates:

$$w_{t+1} = \operatorname*{argmin}_{w \in \mathcal{W}} P_t(w) + \sum_{i=1}^t \langle \tilde{g}_i, w \rangle + a_i \psi(w), \tag{5}$$

for some sequence of "potential functions" $P_t : \mathcal{W} \to \mathbb{R}$. In fact, this approach can be very effective; this is roughly the method employed by [27] for a similar problem. However, deriving the correct potential $P_t$ and proving the desired regret bound can be very difficult, and could easily require separate analysis for each different possible $\psi$ function. For example, [27]'s analysis specifically applies to $\psi(w) = \|w\|^2$. There is other work on similar protocols using approximately this method, such as [28, 29], that also requires particular analysis for each setting. Finally, even if the bound can be achieved in general using this scheme, solving the optimization problem (5) may incur some undesirable computational overhead, even for intuitively "simple" regularizers such as $\psi(w) = \|w\|^2$. In fact, the method of [27] suffers from exactly this issue, which is why we provide an alternative approach in Section 3.3, for the special case of interest that $\mathcal{W} = \mathbb{R}$.

**Re-parametrizing to achieve (4).** In order to achieve the bound (4) in the special case $\mathcal{W} = \mathbb{R}$, we will employ a standard trick in convex optimization: re-parametrizing the objective as a convex constraint using the fact that the epigraph of a convex function is convex. Instead of having our learner output $w_t \in \mathcal{W}$, we will output $(x_t, y_t) \in \mathcal{W} \times \mathbb{R}$, but subject to the constraint that $y_t \geq \psi(x_t)$. We provide details of this approach in Section 3.3.

---

[1]It is likely that the approach of [26] in concert with the varying potentials of [20] would achieve all points on the frontier as well, although our analysis takes a different direction using the centered mirror descent method of [21].

With all of these technicalities introduced, we are ready to provide an outline of our method. The key idea is to show that for a very peculiar choice of coefficients $a_1, \dots, a_T$ and some simple clipping of the gradients $g_t$, we are able to achieve the following result.

**Theorem 1.** *There exists an online learning algorithm that requires $O(d)$ space and takes $O(d)$ time per update, takes as input scalar values $\gamma$, $h_1$, and $\epsilon$ and ensures that for any sequence $g_1, g_2, \dots \subset \mathbb{R}^d$, the outputs $w_1, w_1, \dots \subset \mathbb{R}^d$ satisfy for all $w_\star$ and $T$:*

$$\sum_{t=1}^{T} \langle g_t, w_t - w_\star \rangle \le O\left[ \epsilon G + \epsilon^2 \gamma + \frac{G^2}{\gamma} \log\left(e + \frac{G}{h_1}\right) + \|w_\star\| \sqrt{V \log\left(e + \frac{|w_\star| \sqrt{V} \log^2(T)}{h_1 \epsilon}\right)} \right.$$
$$\left. + \|w_\star\| G \log\left(e + \frac{\|w_\star\| \sqrt{V} \log^2(T)}{h_1 \epsilon}\right) + \gamma \|w_\star\|^2 \log\left(e + \frac{\|w_\star\|^2}{\epsilon^2} \log\left(e + \frac{G}{h_1}\right)\right) \right],$$

*where $G = \max(h_1, \max_{t \in [T]} \|g_t\|)$ and $V = G^2 + \sum_{t=1}^{T} \|g_t\|^2$.*

Before proving this result, let us briefly unpack the algebra in the statement to see how it relates to our originally stated bound (2). Notice that if we drop all the logarithmic terms, the bound becomes:

$$\sum_{t=1}^{T} \langle g_t, w_t - w_\star \rangle \le \tilde{O}\left[ \epsilon G + \epsilon^2 \gamma + \frac{G^2}{\gamma} + \|w_\star\| \sqrt{\sum_{t=1}^{T} \|g_t\|^2} + \|w_\star\| G + \gamma \|w_\star\|^2 \right]$$

Here if we should think of $h_1$ and $\epsilon$ as conservative under-estimates of $\max_t \|g_t\|$ and $\|w_\star\|$. Notice that decreasing $h_1$ and $\epsilon$ only increases the terms inside the logarithms, so that in some sense the algorithm is very robust to even extremely conservative under-estimation. When it holds that $h_1 \le \max_t \|g_t\|$ and $\epsilon \le \|w_\star\|$, then the above bound is exactly the previously stated equation (2).

### 3.2 Proof Sketch of Theorem 1

Let us suppose for now that we have access to an algorithm that achieves the bound (4) under Protocol 2. Let us call it REG. In this section, we will detail how to use REG to achieve our desired goal (2) under Protocol 1 with $\mathcal{W} = \mathbb{R}$: in this sketch, we treat all values as *scalars*, and never vectors. Recall that it suffices to consider $\mathcal{W} = \mathbb{R}$ to achieve the result in general. Given an output $w_t$ from REG, we play $w_t$ and observe the gradient $g_t$. We will then produce a modified gradient $\tilde{g}_t$, a scalar $a_t$, and a magnitude hint $h_{t+1}$ to provide to REG such that $\tilde{g}_t$ and $a_t$ satisfy the constraints of Protocol 2. We will set $\psi(w) = w^2$, and then by careful choice of $\tilde{g}_t$, $a_t$, and $h_{t+1}$, we will be able to establish Theorem 1.

There are two key steps in our reduction. The first step is now a standard trick originally used by [18, 23, 30] to reduce the original Protocol 1 to Protocol 2. The idea is as follows: let us set $h_t = \max(h_1, |g_1|, \dots, |g_{t-1}|)$ for some given "initial value" $h_1 \ge 0$. Notice that $h_t$ may be computed before $g_t$ is revealed and that the value $G$ specified in the theorem satisfies $G = h_{T+1}$. Then, upon receiving a gradient $g_t$, we replace $g_t$ with the "clipped" gradient $\tilde{g}_t = \left(1 \wedge \frac{h_t}{|g_t|}\right) \cdot g_t$. The clipped gradient $\tilde{g}_t$ satisfies $|\tilde{g}_t| \le h_t$ by definition. We then pass $\tilde{g}_t$ in place of $g_t$ to an algorithm that interacts with Protocol 2. It is then relatively straightforward to see that for all $w_\star \in \mathcal{W}$:

$$\sum_{t=1}^{T} g_t(w_t - w_\star) \le \sum_{t=1}^{T} \tilde{g}_t(w_t - w_\star) + \sum_{t=1}^{T} |\tilde{g}_t - g_t| |w_\star| + \sum_{t=1}^{T} |\tilde{g}_t - g_t| |w_t|,$$
$$\le \sum_{t=1}^{T} \tilde{g}_t(w_t - w_\star) + h_{T+1} |w_\star| + \sum_{t=1}^{T} (h_{t+1} - h_t) |w_t|.$$

At this point, prior work [18, 23] observed that if we could constrain $|w_t|$ to have some chosen maximum value $D$, then the final summation above is at most $h_{T+1} D$. By carefully choosing $D$ in tandem with an algorithm that achieve (4) in the case $\psi(w) = 0$, one can achieve the previous "compromise" bound (3).

This is where our *second* key step (which is our main technical innovation) comes in. Instead of explicitly enforcing $|w_t| \le D$, we will apply a "soft constraint" by adding a regularizer. Surprisingly, we will add a very tiny amount of regularization and yet still achieve meaningful regret bounds.

Recall that we are assuming access to an algorithm that achieves the bound (4) when interacting with Protocol 2. Let us set $\psi(w) = w^2$. Then, observe that for any choices of $a_1, \dots, a_T$:

$$\sum_{t=1}^{T} g_t(w_t - w_\star) \le \sum_{t=1}^{T} \left( \tilde{g}_t(w_t - w_\star) + a_t \psi(w_t) - a_t \psi(w_\star) \right) + \sum_{t=1}^{T} |\tilde{g}_t - g_t| |w_\star| + w_\star^2 \sum_{t=1}^{T} a_t$$

$$+ \sum_{t=1}^{T} \left( |\tilde{g}_t - g_t| |w_t| - a_t w_t^2 \right),$$

$$\le \underbrace{\sum_{t=1}^{T} \left( \tilde{g}_t(w_t - w_\star) + a_t \psi(w_t) - a_t \psi(w_\star) \right) + h_{T+1}|w_\star|}_{\text{controlled by (4)}} + \underbrace{w_\star^2 \sum_{t=1}^{T} a_t}_{\text{needs small } a_t} + \underbrace{\sum_{t=1}^{T} \left( (h_{t+1} - h_t)|w_t| - a_t w_t^2 \right)}_{\text{needs big } a_t}.$$

From the above decomposition, we see that to make the overall regret small, we would like to choose $a_t$ such that $\sum_{t=1}^{T} a_t$ is small, but also $a_t$ is large enough that $\sum_{t=1}^{T} \left( (h_{t+1} - h_t)|w_t| - a_t w_t^2 \right)$ is also small. It turns out that this is accomplished by the following choice for $a_t$:

$$a_t = \gamma \cdot \frac{(h_{t+1} - h_t)/h_{t+1}}{1 + \sum_{i=1}^{t} (h_{i+1} - h_i)/h_{i+1}}.$$

Here, $\gamma$ is an arbitrary user-specified constant. Notice that the value of $h_{t+1}$ is available immediately after $g_t$ is revealed, so that it is possible to set this value of $a_t$. Moreover, it is clear that $a_t \in [0, \gamma]$ for all $t$.

Let us see how this value for $a_t$ satisfies our desired properties. First, recall the bound $\log(p+q) - \log(q) \ge \frac{p}{p+q}$ for any $p, q > 0$, which implies $\sum_{t=1}^{T} \frac{p_t}{\sum_{i=0}^{t} p_i} \le \log \left( \sum_{t=1}^{T} p_t / p_0 \right)$ for any sequence of positive numbers $p_0, \dots, p_T$. From this, we have:

$$\sum_{t=1}^{T} a_t = \gamma \sum_{t=1}^{T} \frac{(h_{t+1} - h_t)/h_{t+1}}{1 + \sum_{i=1}^{t} (h_{i+1} - h_i)/h_{i+1}},$$

$$\le \gamma \log \left( 1 + \sum_{t=1}^{T} (h_{t+1} - h_t)/h_{t+1} \right),$$

$$\le \gamma \log \left( 1 + \log(G/h_1) \right).$$

Thus, $\sum_{t=1}^{T} a_t$ is in fact doubly logarithmic in the ratio between $h_1$ and $h_{T+1} = \max(h_1, \max_t |g_t|) = G$.

Next, let us check that $a_t$ is "large enough" to make $\sum_{t=1}^{T} (h_{t+1} - h_t)|w_t| - a_t w_t^2$ small. To this end, observe that:

$$(h_{t+1} - h_t)|w_t| - a_t w_t^2 \le \sup_X (h_{t+1} - h_t)X - a_t X^2,$$

$$= \frac{(h_{t+1} - h_t)^2}{4a_t},$$

$$= \frac{h_{t+1}(h_{t+1} - h_t)}{4\gamma} \left( 1 + \sum_{i=1}^{t} (h_{i+1} - h_i)/h_{i+1} \right),$$

$$\le \frac{h_{T+1}(h_{t+1} - h_t)}{4\gamma} \left( 1 + \sum_{i=1}^{T} (h_{i+1} - h_i)/h_{i+1} \right),$$

$$= \frac{G(h_{t+1} - h_t)}{4\gamma} \left( 1 + \log(G/h_1) \right),$$

where we used that $G = h_{T+1}$. Thus, we have:

$$\sum_{t=1}^{T} \left( (h_{t+1} - h_t)|w_t| - a_t w_t^2 \right) \le \frac{G^2}{4\gamma} \left( 1 + \log(G/h_1) \right).$$

This shows that $a_t$ is large enough that it is able to counteract the effect of $\sum_{t=1}^{T} (h_{t+1} - h_t)|w_t|$ (which makes the regret large if $|w_t|$ is large). It is tempting to conclude that the regularizer is somehow "implicitly constraining" $w_t$ to be small enough that the regret is bounded. However, it is difficult

to envision exactly what constraint is being enforced; notice that to make $\sum_{t=1}^{T}(h_{t+1}-h_t)|w_t| = \widetilde{O}(G^2/\gamma)$ by applying some constraint $|w_t| \le D$, we would need to set $D = \widetilde{O}(G/\gamma)$. However, such an aggresive constraint would surely prevent us from achieving low regret for even relatively moderate $\|w_\star\| \ge G/\gamma$. So, our regularization seems to be doing something more subtle than simply applying a global constraint to the $w_t$'s. Indeed, notice that in the case $|g_t| \le h_1$ for all $t$, we actually have $a_t = 0$ and so no constraint effect at all is enforced!

The final step we need to check is bounding $\sum_{t=1}^{T}\tilde{g}_t(w_t - w_\star) + a_t\psi(w_t) - a_t\psi(w_\star)$. To this end, we provide in Section 3.3 an algorithm that achieves the following bound, which is slightly weaker than (4):

$$\sum_{t=1}^{T}\left(\tilde{g}_t(w_t - w_\star) + a_t\psi(w_t) - a_t\psi(w_\star)\right)$$

$$\le O\left[\epsilon h_T + |w_\star|\sqrt{V\log\left(e + \frac{|w_\star|\sqrt{V}\log^2(T)}{h_1\epsilon}\right)} + |w_\star|h_T\log\left(e + \frac{|w_\star|\sqrt{V}\log^2(T)}{h_1\epsilon}\right)\right.$$

$$\left. + \epsilon^2\gamma + w_\star^2\sqrt{S\log\left(e + \frac{|w_\star|^2\sqrt{S}\log^2(T)}{\gamma\epsilon^2}\right)} + \|w_\star\|^2\gamma\log\left(e + \frac{|w_\star|^2\sqrt{S}\log^2(T)}{\gamma\epsilon^2}\right)\right],$$

where $S = \gamma^2 + \gamma\sum_{t=1}^{T}a_t$. This bound is weaker than (4) due to the presence of $S$ rather than $\gamma^2 + \sum_{t=1}^{T}a_t^2$. Nevertheless, by our bound on $\sum_{t=1}^{T}a_t$, we have:

$$S \le \gamma^2 + \gamma^2\log\left(1 + \log\left(G/h_1\right)\right)$$

so that combining all of the above calculations we establish Theorem 1.

Thus, it remains to establish how we can achieve (4), or the slightly weaker (but sufficient) statement above. We accomplish this next in Section 3.3.

### 3.3 Regularized Online Learning via Epigraph Constraints

Recall that our approach to obtaining (4) is to replace the regularization terms in the loss with constraints. Formally, consider the following protocol:

---

**Protocol 3.** Epigraph-based Regularized Online Learning for $\mathcal{W} = \mathbb{R}$.
**Input:** Convex function $\psi : \mathbb{R} \to \mathbb{R}$.
For $t = 1, \ldots, T$:

    1. Nature reveals magnitude hint $h_t \ge h_{t-1}$ to the learner.
    2. Learner outputs $(x_t, y_t) \in \mathbb{R} \times \mathbb{R}$ with $y_t \ge \psi(x_t)$.
    3. Nature reveals $\tilde{g}_t \in [-h_t, h_t]$ and $a_t \in [0, \gamma]$ to the learner.
    4. Learner suffers loss $\tilde{g}_t x_t + a_t y_t$.

The learner is evaluated with the *linear regret* $\sum_{t=1}^{T}g_t(x_t - w_\star) + a_t(y_t - \psi(w_\star))$. The goal is to obtain:

$$\sum_{t=1}^{T}\left(\tilde{g}_t(x_t - w_\star) + a_t(y_t - \psi(w_\star))\right) \underset{\text{goal}}{\le} \widetilde{O}\left(\|w_\star\|\sqrt{h_T^2 + \sum_{t=1}^{T}\tilde{g}_t^2} + \psi(w_\star)\sqrt{\gamma^2 + \sum_{t=1}^{T}a_t^2}\right). \quad (6)$$

---

The key fact about this protocol is the observation that by setting $w_t = x_t$, the bound (6) immediately implies (4). To see this, recall that $\psi(w) \ge 0$, $a_t \ge 0$, and $y_t \ge \psi(x_t) = \psi(w_t)$ so that:

$$\sum_{t=1}^{T}\left(\langle g_t, w_t - w_\star\rangle + a_t\psi(w_t) - a_t\psi(w_\star)\right) \le \sum_{t=1}^{T}\left(\langle g_t, x_t - w_\star\rangle + a_t y_t - a_t\psi(w_\star)\right).$$

So, to achieve (4) under Protocol 2, it suffices to achieve the bound (6) under Protocol 3.

There is one tempting approach that *almost, but not quite*, achieves this goal. One could employ the "constraint-set reduction" developed in [16] that converts an algorithm that operates on the

"unconstrained" domain $\mathbb{R}^d \times \mathbb{R}$ to one respecting the constraint $y \geq \psi(x)$. In particular, it is relatively straightforward to build an algorithm that achieves (6) without requiring $y_t \geq \psi(x_t)$. This unconstrained setting can be handled by the classic "coordinate-wise updates" trick in which we run two instances of an algorithm achieving (4) in the special case that $\psi(x) = 0$, one of which will output $x_t$ and receive feedback $g_t$, and the other will output $y_t$ and receive feedback $a_t$. Then, by the individual regret bounds on both coordinates, we would have:

$$\sum_{t=1}^{T} \left( g_t(x_t - w_\star) + a_t(y_t - \psi(w_\star)) \right) = \sum_{t=1}^{T} g_t(x_t - w_\star) + \sum_{t=1}^{T} a_t(y_t - \psi(w_\star)),$$

$$\leq \widetilde{O}\left( \|w_\star\| \sqrt{h_T^2 + \sum_{t=1}^{T} \tilde{g}_t^2} + \psi(w_\star) \sqrt{\gamma^2 + \sum_{T=1}^{T} a_t^2} \right).$$

Then, one might hope that applying the constraint-set reduction of [16] would allow us to apply the constraint $\mathcal{W}$ without damaging the regret bound. Unfortunately, this reduction will modify the feedback $g_t$ and $a_t$ in such a way that $\sum_{t=1}^{T} a_t^2$ could become much larger, which makes this approach untenable in general.

Fortunately, it turns out that our particular usage will enforce some favorable conditions on $a_t$ that make the above strategy viable. Specifically, the choices of $\tilde{g}_t$, $h_t$ and $a_t$ described in Section 3.2 satisfy the condition that $a_t = 0$ unless $\|\tilde{g}_t\| = h_t$. By careful inspection of the constraint-set reduction, it is possible to show that the above strategy achieves a slightly weaker version of (6):

$$\sum_{t=1}^{T} \tilde{g}_t(x_t - w_\star) + a_t(y_t - \psi(w_\star)) \leq \widetilde{O}\left( \|w_\star\| \sqrt{h_T^2 + \sum_{t=1}^{T} \tilde{g}_t^2} + \psi(w_\star) \sqrt{\gamma^2 + \gamma \sum_{T=1}^{T} a_t} \right). \quad (7)$$

As detailed in Section 3.2, this weaker bound suffices for our eventual purposes. Nevertheless, for the reader interested in a fully general solution, in Appendix H, we provide a method for achieving (6) without restrictions. We do not employ it in our main development because it involves solving a convex subproblem at each iteration and so may be less efficient in some settings. This technique does however involve a small improvement to so-called "full-matrix" regret bounds [31], and so may be of some independent interest.

## 4 Lower Bounds

In this section, we show that the result of Theorem 1 is tight. In fact, we show a stronger result that generalizes our extra penalty term from $G^2/\gamma + \gamma\|w_\star\|^2$ to $\gamma\psi(\|w_\star\|) + \gamma\psi^\star(G/\gamma)$ for any symmetric convex function $\psi$, where $\psi^\star(x) = \sup_z xz - \psi(z)$ is the Fenchel conjugate of $\psi$. That is, we provide a Pareto frontier of different lower bounds and Theorem 1 is but one point on this frontier. In Appendix A we extend our upper-bound results to match any desired point on this frontier (up to a logarithmic factor). . We also provide matching upper bounds (up to a logarithmic factor) in Theorem 16, as well as more simplified

**Theorem 2.** *Suppose $\psi : \mathbb{R} \to \mathbb{R}$ is convex, symmetric, differentiable, non-negative, achieves its minimum at $\psi(0) = 0$, and $\psi(x)$ is strictly increasing for non-negative $x$. Further suppose that for any $X, Y, Z > 0$, there is some $\tau$ such that for all $T \geq \tau$,*

$$\exp(T) - 1 \geq X\sqrt{T}\nabla\psi^\star(YT)$$
$$X\psi^\star(YTZ) \geq TZ$$

*where $\psi^\star(z) = \sup zx - \psi(x)$ is the Fenchel conjugate of $\psi$. Let $h_1 > 0$, $\gamma > 0$ and $\epsilon > 0$ be given.*

*For any online learning algorithm $\mathcal{A}$ interacting with Protocol 1 with $\mathcal{W} = \mathbb{R}$, there is a $T_0$ such that for any $T \geq T_0$, there is a sequence of gradients $g_1, \ldots, g_T$ and a $w_\star$ such that the outputs $w_1, \ldots, w_T$ of $\mathcal{A}$ satisfy:*

$$\sum_{t=1}^{T} g_t(w_t - w_\star) \geq \epsilon G + \frac{\gamma}{8}\psi^\star(G/\gamma) + \frac{\gamma}{4}\psi(w_\star) + \frac{G|w_\star|}{4}\sqrt{T\log\left(1 + \frac{G|w_\star|\sqrt{T}}{h_1\epsilon}\right)},$$

*where $G = \max(h_1, g_1, \ldots, g_T)$. In particular, with $\psi(x) = x^{1+q}$ for any $q > 0$, we can ensure:*

$$\sum_{t=1}^{T} g_t(w_t - w_\star) \geq \Omega\left[\epsilon G + \frac{G^{1+1/q}}{\gamma^{1/q}} + \gamma|w_\star|^{1+q} + G|w_\star|\sqrt{T \log\left(1 + \frac{G|w_\star|\sqrt{T}}{h_1 \epsilon}\right)}\right].$$

The conditions on $\psi^\star$ in this bound are relatively mild. The first condition says that the gradient $\nabla\psi^\star$ should not grow exponentially fast. The second condition says that $\psi^\star$ should grow faster than some linear function. So, any polynomial of degree greater than 1 satisfies these conditions.

We note that this lower bound leaves something to be desired in terms of the quantification of the terms. Here, the value of $G$ and $\|w_\star\|$ depends on the algorithm $\mathcal{A}$. This is a critical factor in the proof; roughly speaking, the proof operates by providing the algorithm with a constant gradient $g_t = h_1$ at every round. Then, if the iterates $w_t$ grow in some sense "quickly", we "punish" the algorithm with a very large negative gradient, which causes high regret if $w_\star = 0$. Alternatively, if the iterates do not grow quickly, then we show that the regret is large for some $w_\star \gg 1$. This approach is a common idiom for lower bounds in the fully unconstrained setting [18, 22].

However, a much better bound might be possible; ideally, it would hold that for *any* $G$ and $\|w_\star\|$ and algorithm $\mathcal{A}$, we can find a sequence of gradients $g_t$ that enforces our desired regret. Indeed, when either $\|w_\star\|$ or $\max_t \|g_t\|$ is provided to the algorithm, the lower bounds available do take this form [3, 5]. We leave as an open question whether it is possible to do so in our setting.

## 5  Discussion

We have provided a new online learning algorithm that achieves a near-optimal regret bound (2). Our algorithm is "fully unconstrained", or "fully parameter-free", in the sense that we achieve a near-optimal regret bound without requiring bounds on the gradients $g_t$ or the comparison point $w_\star$. Prior work in this setting [18, 21, 22, 23, 26] achieve bounds that are technically incomparable, but may be aesthetically less desirable, as detailed in the discussion following (3). Nevertheless, ideally we would have a unified algorithm framework capturing both our old and new bounds. It is an open question whether more careful choice of regularization in our approach could achieve this goal.

Our algorithm takes as input parameters $\epsilon$, $h_1$ and $\gamma$. All of these have a pleasingly small impact on the regret bound. $\epsilon$ and $h_1$ can be interpreted as very rough estimates of $\|w_\star\|$ and $G$. As these quantities go to zero, the regret bound increases only logarithmically. Moreover, these estimates can be too high by a factor of $\sqrt{T}$ while still maintaining $\tilde{O}(\|w_\star\|G\sqrt{T})$ regret. The quantity $\gamma$ represents an estimate of $G/\|w_\star\|$. As discussed in Section 1, this value does not appear in any term that has a $T$-dependence in the regret bound and so also has a very mild impact on the regret.

While our bound has several intuitively desirable characteristics, it is missing one important property: our bound suffers from an issue highlighted by [18] called the "range-ratio" problem. That is, the bound depends on the ratio $G/h_1$, which could be very large if the losses are rescaled by some arbitrary large number without rescaling $h_1$. This issue is at the heart of how we are able to sidestep the lower-bound of [18], which appears to apply to all algorithms that do not suffer from the range-ratio problem.

### 5.1  Other forms of Unconstrained Online Learning

Our results focus on the case that we have no prior bounds on the value of $\|w_\star\|$ or $\|g_t\|$, and our bounds eventually depend on $\max_t \|g_t\|$. One might worry that this is too conservative in some settings. For example, it might be that $g_t$ is known to be a random variable with bounded mean $\|\mathbb{E}[g_t]\| \leq G$ and variance $\text{Var}(g_t) \leq \sigma^2$ for some known $G$ and $\sigma$. In this case, $\max_t \|g_t\|$ might become large even though intuitively our regret should still depend only on $G + \sigma$. This is the setting considered by several prior work on online learning with unconstrained domains [9, 17, 32]. Under various assumptions, these results all achieve an in-expectation regret bound of $\mathbb{E}[\text{Regret}_T(w_\star)] \leq \widetilde{O}(\|w_\star\|(G + \sigma)\sqrt{T})$.

In fact, our results come close to this ideal even without knowledge of $G$. For example, [9, 17] study the case of sub-exponential $g_t$ that satisfy $\sup_{\|a\| \leq 1} \mathbb{E}[\exp(\beta\langle g_t - \mathbb{E}[g_t], a\rangle)] \leq \exp(\beta^2\sigma^2/2)$ for all $|\beta| \leq 1/b$ for some $b > 0$. In this case, for 1-dimensional $g_t$, we have $\mathbb{E}[\max_t g_t^2] \leq \widetilde{O}(G^2 + \sigma^2)$, and

so in expectation we achieve $\widetilde{O}(\|w_\star\|(G+\sigma)\sqrt{T} + G^2 + \sigma^2 + \|w_\star\|^2)$ (the extension from 1-d to arbitrary dimensions can then be achieved via the black-box reduction of [16]). However, in the case that $g_t$ has some heavy-tailed distribution such as studied by [32], it is less clear that our bounds achieve the desired result out-of-the box. Discovering how to achieve this is an interesting direction for future study.

## 5.2 Parameter-free Algorithms and Stochastic Convex Optimization

As discussed in the introduction, a common motivation for the study of online learning is its immediate application to stochastic convex optimization through various online-to-batch conversions. The classic conversion of [7], as well as a few more recent results [33, 34, 35, 36] all show that if $g_t$ is the output of a stochastic gradient oracle for a convex function $F$, then for any $w_\star \in \mathrm{argmin}\, F$:[2]

$$\mathbb{E}\left[F\left(\frac{\sum_{t=1}^T w_t}{T}\right) - F(w_\star)\right] \le \frac{\mathbb{E}[\mathrm{Regret}_T(w_\star)]}{T}$$

If $\|g_t\| \le G$ with probability 1 (for an unknown $G$), our Theorem 1 immediately implies $\mathbb{E}\left[F\left(\frac{\sum_{t=1}^T w_t}{T}\right) - F(w_\star)\right] \le \widetilde{O}\left(\frac{\|w_\star\|G}{\sqrt{T}} + \frac{G^2/\gamma + \gamma\|w_\star\|^2}{T}\right)$. The first term is the optimal rate for stochastic convex optimization that can be achieved via SGD with learning rate $\eta = \frac{\|w_\star\|}{G\sqrt{T}}$ if $G$ and $\|w_\star\|$ are known ahead of time, and the second term is a lower-order "penalty" for not having up-front knowledge of these quantities.

Convergence results that match that of optimally tuned SGD are often called "parameter-free" (the parameter in question is the learning rate). As mentioned in the introduction, there has been a long line of works that attempt to achieve this goal by matching the regret bound (1), which can then be applied to the stochastic setting via an online-to-batch conversion. More recent work on parameter-free optimization has considered the stochastic case [37, 38], or deterministic case [39] directly without passing through a general regret bound. Many of these algorithms have shown significant empirical promise, even for non-convex deep learning tasks [38, 39, 40, 41, 42]. Almost all of these results require apriori knowledge of the value $G$[3]

To place our results in this context, let us focus on the case of a known $G$ value. In this case, [37] show that by eschewing regret analysis and focusing specifically on the stochastic setting, it is possible to achieve a high-probability guarantee that improves upon the logarithmic factors achieved by our result, and so there seems to be something lost by focusing on regret bounds. However, in a surprising counterpoint, [44] shows that if one is interested in an *in-expectation* result, then there is actually no way to improve upon the logarithmic factors achieved via online-to-batch conversion when applied to parameter-free regret bounds. Thus, our in-expectation stochastic convergence rate is optimal even up to logarithmic factors, while we also do not require prior knowledge of $G$.

Finally, let us evaluate the optimality of our bound in the stochastic setting while accounting for the fact that our methods do not get to know either $G$ or $\|w_\star\|$. Here, we can again make use of the lower bounds developed by [44]. Consider the class of stochastic convex optimization objectives with Lipschitz constant $G$ between 1 and $L$ and $\|w_\star\| \in [1, R]$. The "price of adaptivity" as defined by [44] is the maximum over this class of the ratio between the convergence guarantee of an algorithm that does not know $\|w_\star\|$ and $G$ with respect to the minimax optimal convergence guarantee for an algorithm that does know these values (which is $RG/\sqrt{T}$). We achieve a price of adaptivity of $\widetilde{O}(1 + \max(L, R)/\sqrt{T})$. The best-known lower bound for this class is $\Omega(1 + \min(L, R)/\sqrt{T})$ [44]. Thus, there is a gap here—although we provide matching lower bounds for the *online* setting, it is possible that in the *stochastic* setting, one can improve our bounds. That said, the stochastic lower bound is derived for algorithms that are given the ranges $[1, L]$ and $[1, R]$. Our algorithm does not use this information and it is also plausible that without such knowledge the lower bound itself would improve.

### Acknowledgements

AC is supported by NSF grant number CCF-2211718.

---

[2]The difference between these conversions lies in where the stochastic gradients $g_t$ are computed.
[3]A few exceptions achieve the prior bound (3) [18, 21, 23, 43].

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

# A Generalized Upper Bounds

In Theorem 1, we provide a bound that achieves the "ideal bound" of (1) with an extra penalty term of roughly $G^2/\gamma + \gamma\|w_\star\|^2$. It turns out that this penalty is but one point on a frontier of potential choices that are all immediately accessible by simply changing $\psi(w)$ from $\|w\|^2$ to any other symmetric convex function, as outlined in the lower bounds of Section 4. Here we state formally that each of these bounds is achievable. In particular, by setting $\psi(w) = \|w\|^{1+q}$ for any $q > 0$, we have:

**Theorem 3.** *There is an online learning algorithm that takes as input positive scalar values* $q$, $\gamma$, $h_1$, *and* $\epsilon$ *and ensures that for any sequence* $g_1, g_2, \cdots \subset \mathbb{R}^d$, *the outputs* $w_1, w_1, \cdots \subset \mathbb{R}^d$ *satisfy for all* $w_\star$ *and* $T$:

$$
\sum_{t=1}^{T} \langle g_t, w_t - w_\star \rangle \le O\Bigg[ \epsilon G + \|w_\star\| \sqrt{ V \log\left( e + \frac{\|w_\star\|\sqrt{V}\log^2(T)}{h_1 \epsilon} \right) }
$$
$$
+ \|w_\star\| G \log\left( e + \frac{\|w_\star\|\sqrt{V}\log^2(T)}{h_1 \epsilon} \right)
$$
$$
+ \epsilon^{1+q}\gamma + \gamma\|w_\star\|^{1+q} \log\left( e + \frac{\|w_\star\|^{1+q}}{\epsilon^{1+q}} \log\left( e + \frac{G}{h_1} \right) \right) + \frac{G^{1+1/q}}{\gamma^{1/q}} \log\left( 1 + \log\left( \frac{G}{h_1} \right) \right)^{1/q} \Bigg],
$$

*where* $G = \max(h_1, \max_{t\in[T]} \|g_t\|)$ *and* $V = G^2 + \sum_{t=1}^{T} \|g_t\|^2$.

Finally, it is also the case that the logarithmic terms in our bounds can be adjusted to remove the $T$ dependencies, at the cost of increasing the regret in the case $w_\star = 0$. This is achieved simply by adjusting the logarithmic factors achieved by the algorithm for regularized online learning (Protocol 2) in a manner similar to other recent works in unconstrained online optimization [20, 21, 26]. Formally, we can achieve:

**Theorem 4.** *There is an online learning algorithm that takes as input positive scalar values* $q$, $\gamma$, $h_1$, *and* $\epsilon$ *and ensures that for any sequence* $g_1, g_2, \cdots \subset \mathbb{R}^d$, *the outputs* $w_1, w_1, \cdots \subset \mathbb{R}^d$ *satisfy for all* $w_\star$ *and* $T$:

$$
\sum_{t=1}^{T} \langle g_t, w_t - w_\star \rangle \le O\Bigg[ \epsilon\sqrt{V} + \|w_\star\| \sqrt{ V \log\left( e + \frac{\|w_\star\|}{\epsilon} \right) } + \|w_\star\| G \log\left( e + \frac{\|w_\star\|}{\epsilon} \right)
$$
$$
+ \epsilon^{1+q}\gamma \sqrt{ \log\left( 1 + \log\left( \frac{G}{h_1} \right) \right) } + \gamma\|w_\star\|^{1+q} \log\left( e + \frac{\|w_\star\|^{1+q}}{\epsilon^{1+q}} \log\left( e + \frac{G}{h_1} \right) \right)
$$
$$
+ \frac{G^{1+1/q}}{\gamma^{1/q}} \log\left( 1 + \log\left( \frac{G}{h_1} \right) \right)^{1/q} \Bigg]
$$

*where* $G = \max(h_1, \max_t \|g_t\|)$ *and* $V = G^2 + \sum_{t=1}^{T} \|g_t\|^2$.

Both of these results are immediate corollaries of Theorem 16.

# B Proof of Lower Bound

We restate and prove Theorem 2.

**Theorem 2.** *Suppose $\psi : \mathbb{R} \to \mathbb{R}$ is convex, symmetric, differentiable, non-negative, achieves its minimum at $\psi(0) = 0$, and $\psi(x)$ is strictly increasing for non-negative $x$. Further suppose that for any $X, Y, Z > 0$, there is some $\tau$ such that for all $T \geq \tau$,*

$$\exp(T) - 1 \geq X\sqrt{T}\nabla\psi^{\star}(YT)$$
$$X\psi^{\star}(YTZ) \geq TZ$$

*where $\psi^{\star}(z) = \sup zx - \psi(x)$ is the Fenchel conjugate of $\psi$. Let $h_1 > 0$, $\gamma > 0$ and $\epsilon > 0$ be given.*

*For any online learning algorithm $\mathcal{A}$ interacting with Protocol 1 with $\mathcal{W} = \mathbb{R}$, there is a $T_0$ such that for any $T \geq T_0$, there is a sequence of gradients $g_1, \ldots, g_T$ and a $w_\star$ such that the outputs $w_1, \ldots, w_T$ of $\mathcal{A}$ satisfy:*

$$\sum_{t=1}^{T} g_t(w_t - w_\star) \geq \epsilon G + \frac{\gamma}{8}\psi^{\star}(G/\gamma) + \frac{\gamma}{4}\psi(w_\star) + \frac{G|w_\star|}{4}\sqrt{T\log\left(1 + \frac{G|w_\star|\sqrt{T}}{h_1\epsilon}\right)},$$

*where $G = \max(h_1, g_1, \ldots, g_T)$. In particular, with $\psi(x) = x^{1+q}$ for any $q > 0$, we can ensure:*

$$\sum_{t=1}^{T} g_t(w_t - w_\star) \geq \Omega\left[\epsilon G + \frac{G^{1+1/q}}{\gamma^{1/q}} + \gamma|w_\star|^{1+q} + G|w_\star|\sqrt{T\log\left(1 + \frac{G|w_\star|\sqrt{T}}{h_1\epsilon}\right)}\right].$$

*Proof.* First, let us define $\psi_\gamma(x) = \gamma\psi(x)$. Let $\nabla\psi_\gamma(x)$ and $\nabla\psi_\gamma^{\star}(\theta)$ indicate the derivatives of $\psi_\gamma$ and $\psi_\gamma^{\star}$. The following properties are standard facts about the Fenchel conjugate (see e.g. [45]):

$$\psi_\gamma^{\star}(\theta) = \gamma\psi^{\star}(\theta/\gamma),$$
$$\psi_\gamma^{\star}(0) = 0,$$
$$\psi_\gamma^{\star}(x) \geq 0 \text{ for all } x,$$
$$\psi_\gamma(x) = \nabla\psi_\gamma(x) \cdot x - \psi_\gamma^{\star}(\nabla\psi_\gamma(x)),$$
$$\psi_\gamma^{\star}(\theta) = \nabla\psi_\gamma^{\star}(\theta) \cdot \theta - \psi_\gamma(\nabla\psi_\gamma^{\star}(\theta)).$$

Moreover, $\nabla\psi_\gamma$ and $\nabla\psi_\gamma^{\star}$ are inverses of each other and are odd functions, and $\psi_\gamma^{\star}(x)$ is strictly increasing for non-negative $x$.

Next, observe that for any $X, Y, X', Y', Z'$, there is a $\tau$ such that for all $T \geq \tau$:

$$\exp(T) - 1 \geq X\sqrt{T}\nabla\psi^{\star}(YT), \tag{8}$$
$$X'\psi^{\star}(Y'TZ') \geq TZ'. \tag{9}$$

To see this, observe that by assumption, there is a $\tau_1$ such that (8) holds for all $T \geq \tau_1$, and also a $\tau_2$ such that (9) holds for all $T \geq \tau_2$, so we may take $\tau = \max(\tau_1, \tau_2)$ to achieve both simultaneously.

From this, we see that there is some $T_0$ such that for all $T \geq T_0$:

$$\exp(T) - 1 \geq \frac{2\sqrt{T}}{\epsilon}\nabla\psi^{\star}(2Th_1/\gamma) = \frac{2\sqrt{T}}{\epsilon}\nabla\psi_\gamma^{\star}(2Th_1) \tag{10}$$

$$Th_1 \leq \frac{\gamma}{32\epsilon}\psi^{\star}\left(\frac{2Th_1}{\gamma}\right) = \frac{1}{32\epsilon}\psi_\gamma^{\star}(2Th_1) \tag{11}$$

We now construct the algorithm-dependent sequence $g_1, g_2, \ldots$ that satisfies the claim of the theorem.

1. Define $g_0 \leftarrow 0$ and set $t \leftarrow 1$.

2. Algorithm $\mathcal{A}$ outputs $w_t$.

3. If $w_t < -2\epsilon - \nabla\psi_\gamma^{\star}(2h_1(t-1))$, set $g_t \leftarrow -2(t-1)h_1$ and for all $k \geq 1$ set $g_{t+k} \leftarrow 0$.

4. Else $g_t \leftarrow h_1$.

5. Set $t \leftarrow t + 1$ and go to Item 2.

Suppose that the condition in Item 3 has not been triggered for the first $\tau$ iterations. Then, we have:

$$\sum_{t=1}^{\tau} w_t \cdot g_t \geq -2\epsilon\tau h_1 - \frac{1}{2}\sum_{t=1}^{\tau}\nabla\psi_\gamma^\star(2(t-1)h_1) \cdot 2h_1,$$

$$\geq -2\epsilon\tau h_1 - \frac{1}{2}\sum_{t=1}^{\tau}(\psi_\gamma^\star(2th_1) - \psi_\gamma^\star(2(t-1)h_1)), \quad \text{(by convexity of } \psi_\gamma^\star)$$

$$= -2\epsilon\tau h_1 - \frac{1}{2}\psi_\gamma^\star(2\tau h_1). \tag{12}$$

Now, suppose that the condition in Item 3 is triggered at some iteration $\tau + 1 \geq 1$. Then $G = 2\tau h_1$ and with $w_\star = 0$, we have:

$$\sum_{t=1}^{T} g_t \cdot (w_t - w_\star) = \sum_{t=1}^{\tau+1} g_t \cdot w_t,$$

$$= g_{\tau+1} \cdot w_{\tau+1} + \sum_{t=1}^{\tau} w_t \cdot g_t,$$

$$\geq 4\epsilon\tau h_1 + \nabla\psi_\gamma^\star(2\tau h_1) \cdot 2\tau h_1 - 2\epsilon\tau h_1 - \frac{1}{2}\psi_\gamma^\star(2\tau h_1), \quad \text{(by (12))}$$

$$= 2\epsilon\tau h_1 + \nabla\psi_\gamma^\star(2\tau h_1) \cdot 2\tau h_1 - \psi_\gamma(\nabla\psi_\gamma^\star(2\tau h_1)) + \psi_\gamma(\nabla\psi_\gamma^\star(2\tau h_1)) - \frac{1}{2}\psi_\gamma^\star(2\tau h_1),$$

$$= 2\epsilon\tau h_1 + \psi_\gamma^\star(2\tau h_1) + \psi_\gamma(\nabla\psi_\gamma^\star(2\tau h_1)) - \frac{1}{2}\psi_\gamma^\star(2\tau h_1),$$

$$\geq 2\epsilon\tau h_1 + \frac{1}{2}\psi_\gamma^\star(2\tau h_1), \tag{13}$$

$$= \epsilon G + \frac{1}{2}\psi_\gamma^\star(G).$$

Therefore, overall we have for $w_\star = 0$:

$$\sum_{t=1}^{T} g_t \cdot (w_t - w_\star) \geq \epsilon G + \frac{\gamma}{2}\psi^\star(G/\gamma) + \gamma\psi(|w_\star|) + |w_\star|G\sqrt{T\log\left(1 + \frac{|w_\star|G\sqrt{T}}{h_1\epsilon}\right)}.$$

Alternatively, suppose the condition in Item 3 is never triggered. In this case, let us set $w_\star = -2\nabla\psi_\gamma^\star(2Th_1)$. Then, $G = h_1$ and by (12) we have:

$$\sum_{t=1}^{T} g_t \cdot (w_t - w_\star) \geq \nabla\psi_\gamma^\star(2Th_1) \cdot 2Th_1 - \frac{1}{2}\psi_\gamma^\star(2Th_1) - 2\epsilon Th_1.$$

Using that $\nabla\psi_\gamma^\star(x) \cdot x \geq \psi_\gamma^\star(x)$ by convexity of $\psi_\gamma^\star$ and $\phi_\gamma^\star(0) = 0$, the right-hand side of the previous display can be bounded below by ():

$$\nabla\psi_\gamma^\star(2Th_1) \cdot 2Th_1 - \frac{1}{2}\psi_\gamma^\star(2Th_1) - 2\epsilon Th_1 \geq \frac{1}{2}\psi_\gamma^\star(2Th_1),$$

$$\geq \frac{1}{4}\psi_\gamma^\star(G) + \frac{1}{4}\psi_\gamma^\star(2Th_1) - 2\epsilon Th_1.$$

Applying Eq. (11):

$$\geq \frac{1}{4}\psi_\gamma^\star(G) + 8\epsilon Th_1 - 2\epsilon Th_1,$$

$$\geq \frac{\gamma}{4}\psi^\star(G/\gamma) + 6\epsilon Th_1. \tag{14}$$

Further, since $\psi_\gamma^{\star\star} = \psi_\gamma$, we have:

$$\nabla\psi_\gamma^\star(2Th_1) \cdot 2Th_1 - \frac{1}{2}\psi_\gamma^\star(2Th_1) = \nabla\psi_\gamma^\star(2Th_1) \cdot Th_1 + \frac{1}{2}\left(\nabla\psi_\gamma^\star(2Th_1) \cdot 2Th_1 - \psi_\gamma^\star(2Th_1)\right),$$

$$= \nabla\psi_\gamma^\star(2Th_1) \cdot Th_1 + \frac{1}{2}\psi_\gamma(\nabla\psi_\gamma^\star(2Th_1)),$$

$$= \nabla\psi_\gamma^\star(2Th_1) \cdot Th_1 + \frac{1}{2}\psi_\gamma(-w_\star),$$

$$= \frac{1}{2}|w_\star| \cdot Th_1 + \frac{\gamma}{2}\psi(w_\star).$$

Finally, let us bound $\frac{1}{2}|w_\star| \cdot g_{1:T}$ using our choice of $w_\star$. By definition, we have:

$$h_1T = h_1\sqrt{T \cdot T},$$

$$= h_1\sqrt{T\log(1 + (\exp(T) - 1))},$$

applying Eq. (10):

$$\geq h_1\sqrt{T\log\left(1 + \frac{2\sqrt{T}\nabla\psi_\gamma^\star(2Th_1)}{\epsilon}\right)},$$

$$= h_1\sqrt{T\log\left(1 + \frac{|w_\star|\sqrt{T}}{\epsilon}\right)},$$

$$= G\sqrt{T\log\left(1 + \frac{G|w_\star|\sqrt{T}}{h_1\epsilon}\right)}. \tag{15}$$

Therefore, combining Eq. (14) with Eq. (15):

$$\nabla\psi_\gamma^\star(2Th_1) \cdot 2Th_1 - \frac{1}{2}\psi_\gamma^\star(2Th_1) - 2\epsilon Th_1$$

$$\geq \frac{1}{2}\left(\frac{\gamma}{4}\psi^\star(G/\gamma) + 6\epsilon Th_1\right) + \frac{1}{2}\left(\frac{\gamma}{2}\psi(w_\star) + \frac{G|w_\star|}{2}\sqrt{T\log\left(1 + \frac{G|w_\star|\sqrt{T}}{h_1\epsilon}\right)} - 4\epsilon Th_1\right),$$

$$\geq 3\epsilon Th_1 + \frac{\gamma}{8}\psi^\star(G/\gamma) + \frac{\gamma}{4}\psi(w_\star) + \frac{G|w_\star|}{4}\sqrt{T\log\left(1 + \frac{G|w_\star|\sqrt{T}}{h_1\epsilon}\right)},$$

$$\geq \epsilon G + \frac{\gamma}{8}\psi^\star(G/\gamma) + \frac{\gamma}{4}\psi(w_\star) + \frac{G|w_\star|}{4}\sqrt{T\log\left(1 + \frac{G|w_\star|\sqrt{T}}{h_1\epsilon}\right)}.$$

$\square$

## C   Reduction to $\mathcal{W} = \mathbb{R}$

As a first step in our algorithm design, we observe that the application of some known reductions from [16] can significantly simplify our task. [16] show that to build an algorithm whose regret bound depends on $g_t$ only through the norms $\|g_t\|$, it suffices to consider exclusively the case $\mathcal{W} = \mathbb{R}$. We provide the formal reduction in Algorithm 1, which ensures the following regret bound.

**Theorem 5** ([16])**.** *Algorithm 1 ensures that $|g_t^{magnitude}| \leq 2\|g_t\|$, and also for all $w_\star \in \mathcal{W}$:*

$$\sum_{t=1}^{T}\langle g_t, w_t - w_\star\rangle \leq 4\|w_\star\|\sqrt{2\sum_{t=1}^{T}\|g_t\|^2} + \sum_{t=1}^{T}g_t^{magnitude}(w_t^{magnitude} - \|w_\star\|).$$

---

**Algorithm 1** Reduction From General $\mathcal{W}$ to $\mathbb{R}$

---

**Input:** Convex domain $\mathcal{W} \subseteq \mathbb{R}^d$, online learning algorithm $\mathcal{A}^{1D}$ with domain $\mathbb{R}$.
Initialize $w_1^{\text{direction}} = 0 \in \mathbb{R}^d$
**for** $t = 1 \ldots T$ **do**
    Receive $w_t^{\text{magnitude}} \in \mathbb{R}$ from $\mathcal{A}^{1D}$.
    Set $\hat{w}_t = w_t^{\text{magnitude}} \cdot w_t^{\text{direction}} \in \mathbb{R}^d$.
    Set $w_t = \Pi_{\mathcal{W}} \hat{w} = \operatorname{argmin}_{w \in \mathcal{W}} \|w - \hat{w}\|$.
    Output $w_t$, receive feedback $g_t$.
    Set $g_t^{\text{unconstrained}} = g_t + \|g_t\| \frac{w_t - \hat{w}_t}{\|w_t - \hat{w}_t\|}$.
    Set $w_{t+1}^{\text{direction}} = \Pi_{\|w\| \le 1} w_t^{\text{direction}} - \frac{g_t^{\text{unconstrained}}}{\sqrt{2 \sum_{i=1}^t (g_i^{\text{unconstrained}})^2}}$.
    Set $g_t^{\text{magnitude}} = \langle g_t^{\text{unconstrained}}, w_t^{\text{direction}} \rangle \in \mathbb{R}$.
    Send $g_t^{\text{magnitude}}$ to $\mathcal{A}^{1D}$ as the $t$th feedback.
**end for**

---

From Theorem 5, it is clear that to achieve low regret on $\mathcal{W}$, we need only bound $\sum_{t=1}^T g_t^{\text{magnitude}}(w_t^{\text{magnitude}} - \|w_\star\|)$, which is exactly the regret of a 1-dimensional learner. So, our final results will be established by considering the case of $\mathcal{W} = \mathbb{R}$, although we will define many intermediate problems for general $\mathcal{W}$ as they may have other applications for which the general setting is of interest.

# D   An Efficient Algorithm for Protocol 2 With Restricted (But Sufficient) Assumptions

In this section, we describe our algorithm for Protocol 2 in the special case that $\mathcal{W} = \mathbb{R}$ and $a_t = 0$ whenever $|\tilde{g}_t| \ne h_t$, where $\tilde{g}_t = (1 \wedge \frac{h_t}{|g_t|}) \cdot g_t$. Our algorithm is in fact a reduction to the special case that $a_t = 0$ for all $t$. This is an important special case that has actually also been previously considered in the literature (see e.g. the discussion in Section 3.1), so we provide it as a separate Protocol below:

---

**Protocol 4.** Online Learning with Magnitude Hints.
**Input:** Convex domain $\mathcal{W}$ (recall that we focus on $\mathcal{W} = \mathbb{R}$).
For $t = 1, \ldots, T$:

    1. Nature reveals magnitude hint $h_t \ge h_{t-1}$ to the learner.
    2. Learner outputs $w_t \in \mathcal{W}$.
    3. Nature reveals loss scalar $g_t$ with $\|g_t\| \le h_t$ to the learner.
    4. Learner suffers loss $\langle g_t, w_t \rangle$.

The learner is evaluated with the *regret* $\sum_{t=1}^T g_t(w_t - w_\star)$. The goal is to obtain:

$$\underbrace{\sum_{t=1}^T \langle g_t, w_t - w_\star \rangle}_{\text{goal}} \le \widetilde{O}\left( \|w_\star\| \sqrt{h_T^2 + \sum_{t=1}^T \|g_t\|^2} \right). \tag{16}$$

---

In Section E, we provide an explicit algorithm (Algorithm 3) for Protocol 4 that suffices for our purposes and achieves the bound (16). In the rest of this section, we take the existence of such an algorithm as given, and use it to build our method for Protocol 2.

Our algorithm for Protocol 2 is given in Algorithm 2. The full regret bound is provided by Theorem 10. However, before providing the general bound, which is somewhat technical, we provide two more interpretable corollaries in order to provide a preview of what the method is capable of.

**Corollary 6.** *For any $\epsilon$ with $\psi(\epsilon) > 0$, there exists an algorithm for Protocol 2 such that for all t, the outputs $x_1, \ldots, x_T$ satisfy:*

$$\sum_{t=1}^{T} g_t(x_t - x_\star) + a_t(\psi(x_t) - \psi(x_\star))$$

$$\leq O\left[ \epsilon h_T + \psi(\epsilon)\gamma + |x_\star|\sqrt{V_g \log\left(e + \frac{|x_\star|\sqrt{V_g}\log^2(T)}{h_1\epsilon}\right)} + |x_\star|h_T \log\left(e + \frac{|x_\star|\sqrt{V_g}\log^2(T)}{h_1\epsilon}\right) \right.$$

$$\left. + \psi(x_\star)\sqrt{S_a \log\left(e + \frac{\psi(x_\star)\sqrt{S_a}\log^2(T)}{\gamma\psi(\epsilon)}\right)} + \psi(x_\star)\gamma \log\left(e + \frac{\psi(x_\star)\sqrt{S_a}\log^2(T)}{\gamma\psi(\epsilon)}\right) \right]$$

*Where $V_g = h_T^2 + \sum_{t=1}^{T} g_t^2$ and $S_a = \gamma^2 + \gamma \sum_{t=1}^{T} a_t$.*

*Proof.* Apply Algorithm 2 with the BASE set to Algorithm 3 using $p = 1/2$. Then, in the notation of Theorem 10, the regret bound of Theorem 11 shows that $A, B, C$ are all $O(1)$ while $D$ is $O(\log^2(T))$ and $p = 1/2$. Set $\epsilon_x = \epsilon$ and $\epsilon_\psi = \psi(\epsilon)$. The result immediately follows. □

**Corollary 7.** *For any $\epsilon$ with $\psi(\epsilon) > 0$, there exists an algorithm for Protocol 2 such that for all t, the outputs $x_1, \ldots, x_T$ satisfy:*

$$\sum_{t=1}^{T} g_t(x_t - x_\star) + a_t(\psi(x_t) - \psi(x_\star))$$

$$\leq O\left[ \epsilon\sqrt{V_g} + \psi(\epsilon)\sqrt{S_a} + |x_\star|\sqrt{V_g \log\left(e + \frac{|x_\star|}{\epsilon}\right)} + |x_\star|h_T \log\left(e + \frac{|x_\star|}{\epsilon}\right) \right.$$

$$\left. + \psi(x_\star)\sqrt{S_a \log\left(e + \frac{\psi(x_\star)}{\psi(\epsilon)}\right)} + \psi(x_\star)\gamma \log\left(e + \frac{\psi(x_\star)}{\psi(\epsilon)}\right) \right],$$

*where $V_g = h_T^2 + \sum_{t=1}^{T} g_t^2$ and $S_a = \gamma^2 + \gamma \sum_{t=1}^{T} a_t$.*

*Proof.* Apply Algorithm 2 with the BASE set to Algorithm 3 using $p = 0$. Then, in the notation of Theorem 10, the regret bound of Theorem 11 shows that $A, B, C$ and $D$ are all $O(1)$ while $p = 0$. Set $\epsilon_x = \epsilon$ and $\epsilon_\psi = \psi(\epsilon)$. The result immediately follows. □

**Lemma 8.** *Suppose $\psi : \mathbb{R} \to \mathbb{R}$ is a convex function that achieves its minimum at 0. Let $h > 0$ and $\gamma > 0$ be given and define the norm $\|(x,y)\| = h^2 x^2 + \gamma^2 y^2$ and the distance function $S(\hat{x}, \hat{y}) = \inf_{y \geq \psi(x)} \|(x,y) - (\hat{x}, \hat{y})\|$. For any $(\hat{x}, \hat{y})$, let $(\delta^x, \delta^y)$ be an arbitrary subgradient of $S$ at $(\hat{x}, \hat{y})$. Then, $\delta^y \leq 0$.*

*Proof.* Throughout this proof, we will assume $\hat{x} > 0$. The proof is completely symmetric in the sign of $\hat{x}$.

First, we dispense with the case in which there is no projection: suppose $\hat{y} \geq \psi(\hat{x})$. Then we must have $\hat{y} = y$ and $\hat{x} = x$ and $S(\hat{x}, \hat{y}) = 0$. Further, for any $\tilde{y} > \hat{y}$, $S(\hat{x}, \tilde{y}) = 0$. However, if $\delta_y > 0$, then by definition of subgradient, we must have $0 = S(\hat{x}, \tilde{y}) \geq S(\hat{x}, \hat{y}) + \delta_y(\tilde{y} - \hat{y}) > 0$, which cannot be. Therefore $\delta_y \leq 0$. So, it remains to consider the case $\hat{y} < \psi(\hat{x})$.

Define $(x,y) = \text{argmin}_{y \geq \psi(x)} \|(x,y) - (\hat{x}, \hat{y})\|$. Further, by [16, Theorem 4], we have:

$$(\delta^x, \delta^y) = \left( \frac{h^2(\hat{x} - x)}{\sqrt{h^2(x - \hat{x})^2 + \gamma^2(\hat{y} - y)^2}}, \frac{\gamma^2(\hat{y} - y)}{\sqrt{h^2(x - \hat{x})^2 + \gamma^2(\hat{y} - y)^2}} \right).$$

Therefore, it suffices to show that $\hat{y} \leq y$.

To start, consider the case $\psi(0) > \hat{y}$. Then, we have $\hat{y} < \psi(0) \leq \psi(x) \leq y$ as desired. So, in the following we consider the remaining case $\psi(0) \leq \hat{y} < \psi(\hat{x})$.

---

**Algorithm 2** Algorithm for Protocol 2 (REG)

---

**Input:** Initial online learning algorithm BASE for Procotol 4 with domain $\mathbb{R}$ taking initialization parameter $\epsilon_{\text{BASE}}$. Non-negative convex function $\psi$. Parameters $\gamma > 0$, $\epsilon_x > 0$ and $\epsilon_\psi > 0$
Initialize two copies of BASE: $\text{BASE}_x$ with $\epsilon_{\text{BASE}} = \epsilon_x$ and $\text{BASE}_y$ with $\epsilon_{\text{BASE}} = \epsilon_\psi$
**for** $t = 1 \ldots T$ **do**
    Receive $h_t \geq h_{t-1} \in \mathbb{R}$
    Send $3h_t$ to $\text{BASE}_x$ as the $t$th magnitude hint.
    Send $3\gamma$ to $\text{BASE}_y$ as the $t$th magnitude hint.
    Get $\hat{x}_t \in \mathbb{R}$ from $\text{BASE}_x$
    Get $\hat{y}_t \in \mathbb{R}$ from $\text{BASE}_y$.
    Define the norm $\|(x,y)\|_t^2 = h_t^2 x^2 + \gamma^2 y^2$, with dual norm $\|(g,a)\|_{\star,t}^2 = \frac{g^2}{h_t^2} + \frac{a^2}{\gamma^2}$.
    Define $S_t(\hat{x},\hat{y}) = \inf_{\hat{y} \geq \psi(x)} \|(x,y) - (\hat{x},\hat{y})\|_t$.
    Compute $x_t, y_t = \text{argmin}_{y \geq \psi(x)} \|x_t, y_t) - (\hat{x},\hat{y})\|_t$.
    Receive feedback $g_t \in [-h_t, h_t]$, $a_t \in [0,\gamma]$, such that $a_t = 0$ unless $|g_t| = h_t$.
    Compute $(\delta_t^x, \delta_t^y) = \|g_t\|_{\star,t} \nabla S_t(\hat{x}_t, \hat{y}_t)$
    Send $g_t + \delta_t^x$ to $\text{BASE}_x$ as $t$th feedback.
    Send $a_t + \delta_t^y$ to $\text{BASE}_y$ as $t$th feedback.
**end for**

---

Observe that since $\psi$ is convex, it must be continuous. Therefore, by intermediate value theorem there must be some $\tilde{x} \geq 0$ with $\psi(\tilde{x}) = \hat{y}$. Further, we have $\psi(\tilde{x}) = \hat{y} < \psi(\hat{x})$, so that $\tilde{x} < \hat{x}$.

Now, by convexity, if $x \geq \tilde{x}$, we must $\psi(x) \geq \psi(\tilde{x})$ because $\psi$ must be non-decreasing for positive $x$ since it achieves its minimum at $0$. Therefore, $y \geq \psi(x) \geq \psi(\tilde{x}) = \hat{y}$ and so we are done. So, let us suppose $x < \tilde{x}$.

Further, suppose that $\hat{y} > y$. Then, observe that:

$$h^2(\tilde{x} - \hat{x})^2 + \gamma^2(\max(y, \psi(\tilde{x})) - \hat{y})^2 < h^2(x - \hat{x})^2 + \gamma^2(y - \hat{y})^2,$$

so that the point $(\tilde{x}, \max(y, \psi(\tilde{x})))$ would contradict the optimality of $(x,y)$. Thus, it also cannot be that $\hat{y} > y$ and so we are done. $\qquad\square$

**Lemma 9.** *Let $h > 0$ and $\gamma > 0$ be given and define the norm $\|(x,y)\| = h^2 x^2 + \gamma^2 y^2$ with corresponding dual norm $\|\cdot\|_\star$. Let $(g,a)$ be any point satisfying $|g| \leq h$, $a \in [0,\gamma]$, and $a = 0$ unless $|g| = h$. Let $(\delta^x, \delta^y)$ be any points satisfying $\|(\delta^x, \delta^y)\|_\star = \|(g,a)\|_\star$. Then,*

$$|\delta^x| \leq |g|\sqrt{2},$$
$$|\delta^y| \leq \gamma\sqrt{2}.$$

*Proof.* The dual norm $\|\cdot\|_\star$ is $\|(g,a)\|_\star = \frac{g^2}{h^2} + \frac{a^2}{\gamma^2}$. So, we have:

$$\frac{(\delta^x)^2}{h^2} + \frac{(\delta^y)^2}{\gamma^2} = \frac{g^2}{h^2} + \frac{a^2}{\gamma^2} \leq 2.$$

This immediately implies $|\delta^y| \leq \gamma\sqrt{2}$. We also have

$$(\delta^x)^2 \leq g^2 + \frac{h^2 a^2}{\gamma^2}.$$

Now, since $a = 0$ unless $g^2 = h^2$, this yields either $|\delta^x| \leq |g|$ if $|g| < h$ or $|\delta^x| \leq \sqrt{g^2 + h^2 a^2/\gamma^2} = h\sqrt{2}$ if $|g| = h$, so either way $|\delta^x| \leq |g|\sqrt{2}$. $\qquad\square$

**Theorem 10.** *Let $A, B, CD, \epsilon > 0$, and $p \geq 1$, be given. Suppose that for any sequence $z_1, \ldots, z_T$ and magnitude hints $m_1 \leq \cdots \leq m_T$ satisfying $|z_t| \leq m_t$, BASE outputs $w_1, \ldots, w_T$ and guarantees regret:*

$$\sum_{t=1}^{T} z_t(w_t - u) \leq \epsilon_{\text{BASE}} C m_T^{2p} Z^{1/2-p} + A|u|\sqrt{Z \log\left(e + \frac{D|u|Z^p}{m_1^{2p}\epsilon_{\text{BASE}}}\right)} + B|u|h_T \log\left(e + \frac{D|u|Z^p}{m_1^{2p}\epsilon_{\text{BASE}}}\right)$$

*for any $u \in \mathbb{R}$, where $Z = m_T^2 + \sum_{t=1}^T z_t^2$.*

*Let $\epsilon_x$, $\epsilon_\psi$ and $\gamma$ be given non-negative inputs to Algorithm 2. Then, for any $T$, with $V_g = h_T^2 + \sum_{t=1}^T g_t^2$ and $S_a = \gamma^2 + \gamma \sum_{t=1}^T a_t$, Algorithm 2's output sequence $\hat{x}_1, \ldots, \hat{x}_T$ guarantees:*

$$\sum_{t=1}^T g_t(\hat{x}_t - x_\star) + a_t(\psi(\hat{x}_t) - \psi(x_\star)) \leq$$

$$\mathcal{C}_x \epsilon_x h_T^{2p} V_g^{1/2-p} + \mathcal{A}_x |x_\star| \sqrt{V_g \log\left(e + \frac{\mathcal{D}_x |x_\star| V_g^p}{\epsilon_x h_1^{2p}}\right)} + \mathcal{B}_x |x_\star| \log\left(e + \frac{\mathcal{D}_x |x_\star| V_g^p}{\epsilon_x h_1^{2p}}\right)$$

$$+ \mathcal{C}_\psi \epsilon_\psi \gamma^{2p} S_a^{1/2-p} + \mathcal{A}_\psi \psi(x_\star) \sqrt{S_a \log\left(e + \frac{\mathcal{D}_\psi \psi(x_\star) S_a^p}{\epsilon_\psi \gamma^{2p}}\right)} + \mathcal{B}_\psi \psi(x_\star) \log\left(e + \frac{\mathcal{D}_\psi \psi(x_\star) S_a^p}{\epsilon_\psi \gamma^{2p}}\right),$$

*for any $x_\star \in \mathbb{R}$, where the constants in the above expression are given by:*

$$\mathcal{A}_x = 3A,$$
$$\mathcal{B}_x = 3B,$$
$$\mathcal{C}_x = 3C,$$
$$\mathcal{D}_x = D,$$
$$\mathcal{A}_\psi = \frac{1}{2} + 144A^2,$$
$$\mathcal{B}_\psi = 144A^2 + 24B,$$
$$\mathcal{C}_\psi = 3C\left[(144A^2 + 24B)\log\left(e + 12CD(1152pA^2 + 48pb)^p\right) + \frac{1}{2} + \frac{(2p+1)(2-4p)^{\frac{1-2p}{1+2p}}}{2}\right],$$
$$\mathcal{D}_\psi = 4D\left[1152A^2p + 48pB\right]^p.$$

*Proof.* First, observe that since $(x_t, y_t)$ is the result of a projection to the domain $y \geq \psi(x)$, it must hold that $y_t \geq \psi(x_t)$ for all $t$. Thus, since $a_t > 0$ and $\psi$ is non-negative, we have for any $x_\star$:

$$g_t(x_t - x_\star) + a_t(\psi(x) - \psi(x_\star)) \leq g_t(x_t - x_\star) + a_t(y_t - x_\star).$$

Therefore, it suffices to bound $\sum_{t=1}^T g_t(x_t - x_\star) + a_t(y_t - x_\star)$, which we will now accomplish.

By [16, Theorem 3], we have for any $x_\star \in \mathbb{R}$

$$\sum_{t=1}^T g_t(x_t - x_\star) + a_t(y_t - \psi(x_\star)) \leq \sum_{t=1}^T (g_t + \delta_t^x)(\hat{x}_t - x_\star) + \sum_{t=1}^T (a_t + \delta_t^y)(\hat{y}_t - \psi(x_\star)),$$

and also $\|(\delta_t^x, \delta_t^y)\|_{t,\star} = \|(g_t, a_t)\|_{t,\star}$ by [16, Proposition 1]. Therefore, by Lemma 9, we have $|g_t + \delta_t^x| \leq 3|g_t| \leq 3h_t$. Defining $V_g = h_T^2 + \sum_{t=1}^T (g_t + \delta_t^x)^2$, and by the guarantee of BASE, we have for any $x_\star$:

$$\sum_{t=1}^T (g_t + \delta_t^x)(\hat{x}_t - x_\star)$$

$$\leq C(3h_T)^{2p}\left[9h_T^2 + \sum_{t=1}^T (g_t + \delta_t^x)^2\right]^{1/2-p} \epsilon_x$$

$$+ A|x_\star| \sqrt{\left[9h_T^2 + \sum_{t=1}^T (g_t + \delta_t^x)^2\right] \log\left(e + \frac{D|x_\star|\left[9h_T^2 + \sum_{t=1}^T (g_t + \delta_t^x)^2\right]^p}{(3h_1)^{2p}\epsilon_x}\right)}$$

$$+ 3Bh_T|x_\star| \log\left(e + \frac{D|x_\star|\left[9h_T^2 + \sum_{t=1}^T (g_t + \delta_t^x)^2\right]^p}{(3h_1)^p\epsilon_x}\right)$$

$$\leq 3Ch_T^{2p}V_g^{1/2-p}\epsilon_x + 3A|x_\star|\sqrt{V_g \log\left(e + \frac{D|x_\star|V_g^p}{h_1^{2p}\epsilon_x}\right)} + 3Bh_T|x_\star|\log\left(e + \frac{D|x_\star|V_g^p}{h_1^{2p}\epsilon_x}\right)$$

Next, observe that by Lemma 9, $|a_t + \delta_t^y| \le 3\gamma$. We also have for any $x_\star \in \mathbb{R}$:

$$\sum_{t=1}^T (a_t + \delta_t^y)(\hat{y}_t - \psi(x_\star)) = \frac{1}{2}\sum_{t=1}^T (a_t + \delta_t^y)(y_t - 2\psi(x_\star)) + \frac{1}{2}\sum_{t=1}^T (a_t + \delta_t^y)y_t$$

$$= \frac{1}{2}\sum_{t=1}^T (a_t + \delta_t^y)(y_t - 2\psi(x_\star)) + \frac{1}{2}\sum_{t=1}^T (a_t + \delta_t^y)(y_t - 2y_\star) + y_\star \sum_{t=1}^T (a_t + \delta_t^y)$$

Now, define $V_a = \gamma^2 + \sum_{t=1}^T (a_t + \delta_t^y)^2$. By the guarantee of BASE applied twice, we have that for any $x_\star \in \mathbb{R}$ ($\psi(x_\star) \ge 0$ below represents the comparator for the regret of BASE):

$$\sum_{t=1}^T (a_t + \delta_t^y)(\hat{y}_t - \psi(x_\star))$$

$$\le C(3\gamma)^{2p}\left[9\gamma^2 + \sum_{t=1}^T (a_t + \delta_t^y)^2\right]^{1/2-p}\epsilon_\psi$$

$$+ A\psi(x_\star)\sqrt{\left(9\gamma^2 + \sum_{t=1}^T (a_t + \delta_t^y)^2\right)\log\left(e + \frac{2D\psi(x_\star)\left[9\gamma^2 + \sum_{t=1}^T (a_t + \delta_t^y)^2\right]^p}{3^{2p}\gamma^{2p}\epsilon_\psi}\right)}$$

$$+ 3\gamma B\psi(x_\star)\log\left(e + \frac{2D\psi(x_\star)\left[9\gamma^2 + \sum_{t=1}^T (a_t + \delta_t^y)^2\right]^p}{3^{2p}\gamma^{2p}\epsilon_\psi}\right)$$

$$+ Ay_\star\sqrt{\left(9\gamma^2 + \sum_{t=1}^T (a_t + \delta_t^y)^2\right)\log\left(e + \frac{2Dy_\star\left[9\gamma^2 + \sum_{t=1}^T (a_t + \delta_t^y)^2\right]^p}{3^{2p}\gamma^{2p}\epsilon_\psi}\right)}$$

$$+ 3\gamma B|y_\star|\log\left(e + \frac{2Dy_\star\left[9\gamma^2 + \sum_{t=1}^T (a_t + \delta_t^y)^2\right]^p}{3^{2p}\gamma^{2p}\epsilon_\psi}\right)$$

$$+ y_\star \sum_{t=1}^T (a_t + \delta_t^y)$$

$$\le 3C\gamma^{2p}V_a^{1/2-p}\epsilon_\psi + 3A\psi(x_\star)\sqrt{V_a\log\left(e + \frac{2D\psi(x_\star)V_a^p}{\gamma^{2p}\epsilon_\psi}\right)} + 3B\psi(x_\star)\log\left(e + \frac{2D\psi(x_\star)V_a^p}{\gamma^{2p}\epsilon_\psi}\right)$$

$$+ 3Ay_\star\sqrt{V_a\log\left(e + \frac{2Dy_\star V_a^p}{\epsilon_\psi}\right)} + 3\gamma By_\star\log\left(e + \frac{2Dy_\star V_a^p}{\gamma^{2p}\epsilon_\psi}\right)$$

$$+ y_\star \sum_{t=1}^T (a_t + \delta_t^y)$$

$$\le 3C\gamma^{2p}V_a^{1/2-p}\epsilon_\psi + 3A(\psi(x_\star) + y_\star)\sqrt{V_a\log\left(e + \frac{2D(\psi(x_\star) + y_\star)V_a^p}{\gamma^{2p}\epsilon_\psi}\right)}$$

$$+ 3B(\psi(x_\star) + y_\star)\log\left(e + \frac{2D\psi(x_\star)V_a^p}{\gamma^{2p}\epsilon_\psi}\right)$$

$$+ y_\star \sum_{t=1}^T (a_t + \delta_t^y)$$

Now, we observe:

$$\sum_{t=1}^T (a_t + \delta_t^y) = \frac{1}{2\gamma}\sum_{t=1}^T \left[(a_t + \delta_t^y + \gamma)^2 - \gamma^2\right] - \frac{1}{2\gamma}\sum_{t=1}^T (a_t + \delta_t^y)^2,$$

$$= \frac{1}{2\gamma}\sum_{t=1}^T \left[(a_t + \delta_t^y + \gamma)^2 - \gamma^2\right] + \frac{\gamma}{2} - \frac{1}{2\gamma}V_a.$$

Next, we bound $(a_t + \delta_t^y + \gamma)^2 - \gamma^2$:

$$(a_t + \delta_t^y + \gamma)^2 - \gamma = a_t^2 + (\delta_t^y)^2 + 2a_t\delta_t^y + 2\gamma a_t + 2\gamma\delta_t^y,$$

using $\delta_t^y \leq 0$ (from Lemma 8) amd $|\delta_t^y| \leq \gamma\sqrt{2} \leq 2\gamma$ (from Lemma 9):

$$\leq a_t^2 + 2a_t\delta_t^y + 2\gamma a_t$$

using $0 \leq a_t \leq \gamma$:

$$\leq 3\gamma a_t,$$

so that we have (recalling that $S_a = \gamma^2 + \gamma\sum_{t=1}^T a_t$:

$$y_\star \sum_{t=1}^T (a_t + \delta_t^y) \leq -\frac{y_\star}{2\gamma}V_a + \frac{3y_\star}{2\gamma}S_a.$$

So, overall we have:

$$\sum_{t=1}^T (a_t + \delta_t^y)(\hat{y}_t - \psi(x_\star))$$

$$\leq 3C\gamma^{2p}V_a^{1/2-p}\epsilon_\psi + 3A(\psi(x_\star) + y_\star)\sqrt{V_a \log\left(e + \frac{2D(\psi(x_\star) + y_\star)V_a^p}{\gamma^{2p}\epsilon_\psi}\right)}$$

$$+ 3B(\psi(x_\star) + y_\star)\log\left(e + \frac{2D\psi(x_\star)V_a^p}{\gamma^{2p}\epsilon_\psi}\right) - \frac{y_\star}{2\gamma}V_a + \frac{3y_\star}{2\gamma}S_a.$$

Since the above holds for *any* $y_\star \geq 0$, we may write:

$$\sum_{t=1}^T (a_t + \delta_t^y)(\hat{y}_t - \psi(x_\star))$$

$$\leq \inf_{y_\star}\sup_{V_a}\left[3C\gamma^{2p}V_a^{1/2-p}\epsilon_\psi + 3A(\psi(x_\star) + y_\star)\sqrt{V_a \log\left(e + \frac{2D(\psi(x_\star) + y_\star)V_a^p}{\gamma^{2p}\epsilon_\psi}\right)}\right.$$

$$\left.+ 3B(\psi(x_\star) + y_\star)\log\left(e + \frac{2D\psi(x_\star)V_a^p}{\gamma^{2p}\epsilon_\psi}\right) - \frac{y_\star}{2\gamma}V_a + \frac{3y_\star}{2\gamma}S_a\right]$$

Now, applying Lemma 20 to bound the minimax expression above, we have:

$$\sum_{t=1}^T (a_t + \delta_t^y)(\hat{y}_t - \psi(x_\star)) \leq \left(\frac{1}{2} + 144A^2\right)\psi(x_\star)\sqrt{S_a \log\left(e + \frac{4D\psi(x_\star)S_a^p}{\epsilon_\psi\gamma^{2p}}\left[1152A^2p + 48pB\right]^p\right)}$$

$$+ \gamma\psi(x_\star)(144A^2 + 24B)\log\left(e + \frac{4D\psi(x_\star)S_a^p}{\epsilon_\psi\gamma^{2p}}\left[1152A^2p + 48pB\right]^p\right)$$

$$+ 3C\gamma^{2p}S^{1/2-p}\epsilon_\psi\left[(144A^2 + 24B)\log\left(e + 12CD(1152pA^2 + 48pb)^p\right) + \frac{1}{2} + \frac{(2p+1)(2-4p)^{\frac{1-2p}{1+2p}}}{2}\right]$$

So, overall we achieve:

$$\sum_{t=1}^T g_t(x_t - x_\star) + a_t(y_t - \psi(x_\star))$$

$$\leq 3Ch_T^{2p}V_g^{1/2-p}\epsilon_x + 3A|x_\star|\sqrt{V_g \log\left(e + \frac{D|x_\star|V_g^p}{h_1^{2p}\epsilon_x}\right)} + 3Bh_T|x_\star|\log\left(e + \frac{D|x_\star|V_g^p}{h_1^{2p}\epsilon_x}\right)$$

$$+ \left(\frac{1}{2} + 144A^2\right)\psi(x_\star)\sqrt{S_a \log\left(e + \frac{4D\psi(x_\star)S_a^p}{\epsilon_\psi\gamma^{2p}}\left[1152A^2p + 48pB\right]^p\right)}$$

$$+ \gamma\psi(x_\star)(144A^2 + 24B)\log\left(e + \frac{4D\psi(x_\star)S_a^p}{\epsilon_\psi\gamma^{2p}}\left[1152A^2p + 48pB\right]^p\right)$$

$$+ 3C\gamma^{2p}S^{1/2-p}\epsilon_\psi\left[(144A^2 + 24B)\log\left(e + 12CD(1152pA^2 + 48pb)^p\right) + \frac{1}{2} + \frac{(2p+1)(2-4p)^{\frac{1-2p}{1+2p}}}{2}\right].$$

So, with:

$$\mathcal{A}_x = 3A,$$
$$\mathcal{B}_x = 3B,$$
$$\mathcal{C}_x = 3C,$$
$$\mathcal{D}_x = D,$$
$$\mathcal{A}_\psi = \frac{1}{2} + 144A^2,$$
$$\mathcal{B}_\psi = 144A^2 + 24B,$$
$$\mathcal{C}_\psi = 3C\left[(144A^2 + 24B)\log\left(e + 12CD(1152pA^2 + 48pb)^p\right) + \frac{1}{2} + \frac{(2p+1)(2-4p)^{\frac{1-2p}{1+2p}}}{2}\right],$$
$$\mathcal{D}_\psi = 4D\left[1152A^2p + 48pB\right]^p,$$

we have

$$\sum_{t=1}^{T} g_t(x_t - x_\star) + a_t(y_t - \psi(x_\star))$$

$$\leq \mathcal{C}_x \epsilon_x h_T^{2p} V_g^{1/2-p} + \mathcal{A}_x |x_\star| \sqrt{V_g \log\left(e + \frac{\mathcal{D}_x |x_\star| V_g^p}{\epsilon_x h_1^{2p}}\right)} + \mathcal{B}_x |x_\star| \log\left(e + \frac{\mathcal{D}_x |x_\star| V_g^p}{\epsilon_x h_1^{2p}}\right),$$

$$+ \mathcal{C}_\psi \epsilon_\psi \gamma^{2p} S_a^{1/2-p} + \mathcal{A}_\psi \psi(x_\star) \sqrt{S_a \log\left(e + \frac{\mathcal{D}_\psi \psi(x_\star) S_a^p}{\epsilon_\psi \gamma^{2p}}\right)}$$

$$+ \mathcal{B}_\psi \psi(x_\star) \log\left(e + \frac{\mathcal{D}_\psi \psi(x_\star) S_a^p}{\epsilon_\psi \gamma^{2p}}\right).$$

from which the conclusion follows.

$\square$

# E   A Parameter-Free Algorithm With Optimal Log Factors for Protocol 4

In this section we quote an algorithm that obtains a performance guarantee suitable for use as BASE in Theorem 10. We emphasize that the development in this section is only a very mild improvement (affecting only logarithmic factors) on previous work: our key contribution is how to use this algorithm to obtain better adaptivity to unknown Lipschitz constants.

In fact, algorithms satisfying the requirements of Theorem 10 up to logarithmic factors have been described by several previous authors: see [18, 21, 23, 26]. Here, we provide a slightly improved analysis of the algorithm of [21] which achieves tighter (and in fact optimal) logarithmic terms.
,

**Theorem 11.** *Suppose* $g_1, \ldots, g_T$ *is any sequence of real numbers and* $0 < h_1 \leq \cdots \leq h_T$ *is another sequence of real numbers satisfying* $|g_t| \leq h_t$. *Then, if* $p = 1/2$, *Algorithm 3 guarantees for all* $u$

$$\sum_{t=1}^{T} g_t(w_t - u) \leq 8h_T \epsilon + 6|u| \sqrt{\left(h_T^2 + \sum_{t=1}^{T} g_t^2\right) \log\left(\frac{|u|\sqrt{3 + \sum_{t=1}^{T} g_t^2/h_t^2} \log^2\left(3 + \sum_{t=1}^{T} g_t^2/h_t^2\right)}{\epsilon} + 1\right)}$$

$$+ 6|u|h_T \log\left(\frac{|u|\sqrt{3 + \sum_{t=1}^{T} g_t^2/h_t^2} \log^2\left(3 + \sum_{t=1}^{T} g_t^2/h_t^2\right)}{\epsilon} + 1\right),$$

**Algorithm 3** 1-Dimensional Learner for Protocol 4 (BASE)

---

**Input:** $\epsilon > 0$, $p \in [0, 1/2]$
Initialize $h_0 = 0$, $k = 3$
**if** $p = 1/2$ **then**
   Define constant $c = 3$
**else**
   Define constant $c = 1$
**end if**
**for** $t = 1 \ldots T$ **do**
   Receive $h_t \geq h_{t-1} \in \mathbb{R}$
   Define $V_t = h_t^2 + \sum_{i=1}^{t-1} g_i^2$
   **if** $p = 1/2$ **then**
      Set $\alpha_t = \dfrac{\epsilon}{\sqrt{c + \sum_{i=1}^{t-1} g_i^2/h_i^2}\, \log^2\left(c + \sum_{i=1}^{t-1} g_i^2/h_i^2\right)}$
   **else**
      Define $\alpha_t = \dfrac{\epsilon}{\left(c + \sum_{i=1}^{t-1} g_i^2/h_i^2\right)^p}$
   **end if**
   Define $\Theta_t = \begin{cases} \dfrac{\left(\sum_{i=1}^{t-1} g_i\right)^2}{4k^2 V_t} & \text{if } \left|\sum_{i=1}^{t-1} g_i\right| \leq \dfrac{2kV_t}{h_t} \\[2ex] \dfrac{\left|\sum_{i=1}^{t-1} g_i\right|}{kh_t} - \dfrac{V_t}{h_t^2} & \text{otherwise} \end{cases}$
   Output $w_t = -\text{sign}\left(\sum_{i=1}^{t-1} g_i\right) \alpha_t \left(\exp(\Theta_t) - 1\right)$
   Receive $g_t$ with $|g_t| \leq h_t$.
**end for**

---

*while if $p < 1/2$ Algorithm 3 guarantees instead:*

$$\sum_{t=1}^{T} g_t(w_t - u) \leq \frac{4h_T^{2p}\epsilon \left(\sum_{t=1}^{T} g_t^2\right)^{1/2-p}}{1 - 2p}$$

$$+ 6|u|\sqrt{\left(h_T^2 + \sum_{t=1}^{T} g_t^2\right) \log\left(\frac{|u|\left(1 + \sum_{t=1}^{T} g_t^2/h_t^2\right)^p}{\epsilon} + 1\right)}$$

$$+ 6|u|h_T \log\left(\frac{|u|\left(1 + \sum_{t=1}^{T} g_t^2/h_t^2\right)^p}{\epsilon} + 1\right).$$

Notice that the term $\log^2\left(3 + \sum_{t=1}^{T} g_t^2/h_t^2\right) \leq \log^2(3 + T)$, and so we upper bound this term with a constant for the purposes of use in Theorem 10. Further, the term $\sum_{t=1}^{T} g_t^2/h_t^2 \leq \sum_{t=1}^{T} g_t^2/h_1^2$, and so the logarithmic terms always fit into the framework of Theorem 10.

*Proof.* Observe that Algorithm 3 is an instance of FTRL with regularizer:

$$\psi_t(w) = k \int_0^{|w|} \min_{\eta \leq 1/h_t} \left[\frac{\log(x/\alpha_t + 1)}{\eta} + \eta V_t\right] dx.$$

That is,

$$w_t = \operatorname*{argmin}_{w} \psi_{t+1}(w) + \sum_{i=1}^{t-1} g_i w.$$

In the "centered mirror descent" framework of [21] (their Algorithm 1), this corresponds to setting $\varphi(w) = 0$. Further, [21] provides an analysis of this update for the particular family of regularizer functions $\psi_t$ we consider above in their Theorem 6. Although formally speaking, their Theorem 6 specifies a particular equation for $\alpha_t$, inspection of the proof shows that most of their argument applies so long as $\alpha_t$ is non-increasing. We reproduce this verification in Lemma 12, which yields:

$$\sum_{t=1}^{T} g_t(w_t - u) \leq \psi_T(u) + \sum_{t=1}^{T} \frac{2\alpha_t}{\sqrt{V_t}}.$$

Next, define $h_{T+1} = 0$ and $g_{T+1} = 0$ in order to define $\alpha_{T+1}$ and $\psi_{T+1} \geq \psi_T$. So, we can replace $\psi_T(u)$ with $\psi_{T+1}(u)$ in the above expression. Next, to bound $\psi_{T+1}(u)$, we observe that:

$$\psi_{T+1}(u) = k \int_0^{|u|} \min_{\eta \leq 1/h_{T+1}} \left[ \frac{\log(x/\alpha_{T+1} + 1)}{\eta} + \eta V_{T+1} \right] dx,$$

$$\leq k|u| \min_{\eta \leq 1/h_{T+1}} \left[ \frac{\log(u/\alpha_{T+1} + 1)}{\eta} + \eta V_{T+1} \right].$$

Now, notice that if the minimizing $\eta$ of $\min_{\eta \leq 1/h_{T+1}} \left[ \frac{\log(u/\alpha_{T+1}+1)}{\eta} + \eta V_{T+1} \right]$ occurs on the boundary $\eta = 1/h_{T+1}$, then it must be that $\frac{\log(u/\alpha_{T+1}+1)}{\eta} > \eta V_{T+1}$, since $\frac{\log(u/\alpha_{T+1}+1)}{\eta}$ is decreasing in $\eta$ and $\eta V_{T+1}$ is increasing in $\eta$. Thus in this case $\min_{\eta \leq 1/h_{T+1}} \left[ \frac{\log(u/\alpha_{T+1}+1)}{\eta} + \eta V_{T+1} \right] \leq 2h_T \log(u/\alpha_{T+1} + 1)$. Alternatively, when the minimizing $\eta$ is not on the boundary we have $\min_{\eta \leq 1/h_{T+1}} \left[ \frac{\log(u/\alpha_{T+1}+1)}{\eta} + \eta V_{T+1} \right] = 2\sqrt{V_{T+1} \log(u/\alpha_{T+1} + 1)}$. So, in general we have:

$$\psi_{T+1}(u) \leq 2k|u|\sqrt{V_{T+1} \log(|u|/\alpha_{T+1} + 1)} + 2k|u|h_T \log(|u|/\alpha_{T+1} + 1).$$

So far this analysis is identical to that of [21], and has been agnostic to the value of $\alpha_t$, so long as $\alpha_t$ is non-increasing. Now, however, we come to the place at which we diverge in analysis: our choice of $\alpha_t$ is slightly larger and so results in better logarithmic factors in $\psi$. The trade-off is that we need to provide a fresh analysis of $\sum_{t=1}^T \frac{2\alpha_t g_t^2}{\sqrt{V_t}}$ to show that this term is still controlled. We accomplish this in Lemma 21 (for $p = 1/2$) and Lemma 22 (for $p < 1/2$). For $p = 1/2$, we then obtain:

$$\sum_{t=1}^T g_t(w_t - u) \leq 8h_T \epsilon + 2k|u| \sqrt{\left( h_T^2 + \sum_{t=1}^T g_t^2 \right) \log \left( \frac{|u|\sqrt{3 + \sum_{t=1}^T g_t^2/h_t^2} \log^2 \left(3 + \sum_{t=1}^T g_t^2/h_t^2 \right)}{\epsilon} + 1 \right)}$$

$$+ 2k|u|h_T \log \left( \frac{|u|\sqrt{3 + \sum_{t=1}^T g_t^2/h_t^2} \log^2 \left(3 + \sum_{t=1}^T g_t^2/h_t^2 \right)}{\epsilon} + 1 \right),$$

while for $p < 1/2$ we obtain:

$$\sum_{t=1}^T g_t(w_t - u) \leq \frac{4h_T^{2p}\epsilon \left( \sum_{t=1}^T g_t^2 \right)^{1/2-p}}{1 - 2p}$$

$$+ 2k|u| \sqrt{\left( h_T^2 + \sum_{t=1}^T g_t^2 \right) \log \left( \frac{|u| \left(1 + \sum_{t=1}^T g_t^2/h_t^2 \right)^p}{\epsilon} + 1 \right)}$$

$$+ 2k|u|h_T \log \left( \frac{|u| \left(1 + \sum_{t=1}^T g_t^2/h_t^2 \right)^p}{\epsilon} + 1 \right).$$

The conclusion now follows by substituting in $k = 3$. □

The following technical Lemma is lifted almost entirely from [21]. Unfortunately, this result was not explicitly declared as a separate Lemma in the prior literature and is instead merely a subset of the proof of a larger Theorem (specifically, Theorem 6 of [21]). So, we include the argument here for completeness. The steps are nearly identical to the prior literature, with only very mild improvement to some constants.

**Lemma 12.** *Let $g_1, \ldots, g_T$ be an arbitrary sequence of scalars. Suppose $0 < h_1 \leq \cdots \leq h_T$ is non-decreasing sequence with $|g_t| \leq h_t$ for all $t$, and let $\alpha_1 \geq \cdots \geq \alpha_T$, be a non-increasing sequence. Let $k \geq 3$. Set $V_t = h_t^2 + \sum_{i=1}^{t-1} g_i^2$ and define*

$$\psi_t(w) = k \int_0^{|w|} \min_{\eta \leq 1/h_t} \left[ \frac{\log(x/\alpha_t + 1)}{\eta} + \eta V_t \right] dx$$

$$w_t = \underset{w}{argmin}\, \psi_t(w) + \sum_{i=1}^{t-1} g_i w.$$

*Then for all $u \in \mathbb{R}$:*

$$\sum_{t=1}^{T} g_t(w_t - u) \le \psi_T(u) + \sum_{t=1}^{T} \frac{2\alpha_t g_t^2}{\sqrt{V_t}}.$$

*Proof.* Define $\psi_{T+1} = \psi_T$ and let $D_f(a|b)$ indicate the Bregman divergence $D_f(a|b) = f(a) - f(b) - f'(b)(a-b)$. Define $\Delta_t(w) = D_{\psi_{t+1}}(w|w_1)$. Then, by [21] Lemma 1, we have:

$$\sum_{t=1}^{T} g_t(w_t - u) \le \psi_T(u) + \sum_{t=1}^{T} g_t(w_t - w_{t+1}) - D_{\psi_t}(w_{t+1}|w_t) - \Delta_t(w_{t+1})$$

So, it suffices to establish that:

$$g_t(w_t - w_{t+1}) - D_{\psi_t}(w_{t+1}|w_t) - \Delta_t(w_{t+1}) \le \frac{2\alpha_t g_t}{\sqrt{V_t}} \tag{17}$$

Following the notation and argument of [21], define $F_t(w) = \log(w/\alpha_t + 1)$ and

$$\Psi_t(x) = k \int_0^x \min_{\eta \le 1/h_t} \left[ \frac{F_t(x)}{\eta} + \eta V_t \right] dx$$

Then we have $\psi(w) = \Psi_t(\|w\|)$ and elementary calculation yields:

$$\Psi_t'(x) = \begin{cases} 2k\sqrt{V_t F_t(x)} & \text{if } h_t\sqrt{F_t(x)} \le \sqrt{V_t} \\ kh_t F_t(x) + \frac{kV_t}{h_t} & \text{otherwise} \end{cases}$$

$$\Psi_t''(x) = \begin{cases} \frac{k\sqrt{V_t}}{(x+\alpha_t)\sqrt{F_t(x)}} & \text{if } h_t\sqrt{F_t(x)} \le \sqrt{V_t} \\ \frac{kh_t}{x+\alpha_t} & \text{otherwise} \end{cases}$$

$$\Psi_t'''(x) = \begin{cases} \frac{-k\sqrt{V_t}(1+2F_t(x))}{2(x+\alpha_t)^2 F_t(x)^{3/2}} & \text{if } h_t\sqrt{F_t(x)} \le \sqrt{V_t} \\ \frac{-kh_t}{(x+\alpha_t)^2} & \text{otherwise} \end{cases}$$

Therefore, $\Psi_t(x) \ge 0$, $\Psi_t'(x) \ge 0$, $\Psi_t''(x) \ge 0$ and $\Psi_t'''(x) \le 0$. Further, if we define $x_0 = \alpha_t(e - 1)$, then for any $> x_0$ we have $\sqrt{F_t(x)} \ge \frac{1}{\sqrt{F_t(x)}}$ and:

$$-\frac{\Psi_t'''(x)}{\Psi_t''(x)^2} \le \begin{cases} \frac{1}{2k\sqrt{V_t}} \left( \frac{1}{\sqrt{F_t(x)}} + 2\sqrt{F_t(x)} \right) & \text{if } h_t\sqrt{F_t(x)} \le \sqrt{V_t} \\ \frac{1}{kh_t} & \text{otherwise} \end{cases}$$

$$\le \begin{cases} \frac{3\sqrt{F_t(x)}}{2k\sqrt{V_t}} & \text{if } h_t\sqrt{F_t(x)} \le \sqrt{V_t} \\ \frac{1}{kh_t} & \text{otherwise} \end{cases}$$

using $k \ge 3$:

$$\le \frac{1}{2} \min \left( \sqrt{\frac{F_t(x)}{V_t}}, \frac{1}{h_t} \right)$$

Now, if we define $Z_t(x) = \int_0^x \min \left( \sqrt{\frac{F_t(\overline{x})}{V_t}}, \frac{1}{h_t} \right) d\overline{x}$, then we have

$$-\frac{\Psi_t'''(x)}{\Psi_t''(x)^2} \le \frac{1}{2} Z_t'(x)$$

Clearly $Z_t$ is convex, $1/h_t$ Lipschitz, and achieves its minimum value of $0$ at $0$. Therefore, by [21] Lemma 2, we have:

$$g_t(w_t - w_{t+1}) - D_{\psi_t}(w_{t+1}|w_t) - Z_t(|w_{t+1}|)g_t^2 \le \frac{2g_t^2}{\Psi''(x_0)},$$

$$\le \frac{2g_t^2(x_0 + \alpha_t)}{k\sqrt{V_t}},$$

$$= \frac{2g_t^2\alpha_t e}{k\sqrt{V_t}},$$

$$\le \frac{2g_t^2\alpha_t}{\sqrt{V_t}}.$$

So, now if we could show that $\Delta_t(w) \geq Z_t(|w|)g_t^2$, this would establish (17). In turn, since $\Delta_t(w) = \Psi_{t+1}(|w|) - \Psi_t(|w|)$, it suffices to establish:

$$\Psi_{t+1}'(x) - \Psi_t'(x) \geq Z'(x)g_t^2 = g_t^2 \min\left(\sqrt{\frac{F_t(x)}{V_t}}, \frac{1}{h_t}\right).$$

To this end, we compute:

$$\Psi_{t+1}'(x) - \Psi_t'(x) = k \min_{\eta \leq 1/h_{t+1}} \left[\frac{F_{t+1}(x)}{\eta} + \eta V_{t+1}\right] - k \min_{\eta \leq 1/h_t} \left[\frac{F_t(x)}{\eta} + \eta V_t\right],$$

$$\geq k \min_{\eta \leq 1/h_t} \left[\frac{F_{t+1}(x)}{\eta} + \eta V_{t+1}\right] - k \min_{\eta \leq 1/h_t} \left[\frac{F_t(x)}{\eta} + \eta V_t\right].$$

Next, let us define $\delta_m = h_{t+1}^2 - h_t^2$ so that $V_{t+1} = V_t + \delta_m + g_t^2$. Then we have $\frac{F_{t+1}(x)}{\eta} + \eta V_{t+1} \geq \min_{\eta'}\left[\frac{F_{t+1}(x)}{\eta'} + \eta' V_t\right] + \eta(\delta_m + g_t^2)$. Armed with this calculation, we proceed:

$$\Psi_{t+1}'(x) - \Psi_t'(x) \geq k(\delta_m + g_t^2)\min\left[\sqrt{\frac{F_{t+1}(x)}{V_{t+1}}}, \frac{1}{h_t}\right] + k \min_{\eta \leq 1/h_t}\left[\frac{F_{t+1}(x)}{\eta} + \eta V_t\right] - k \min_{\eta \leq 1/h_t}\left[\frac{F_t(x)}{\eta} + \eta V_t\right],$$

now, since $\alpha_t \geq \alpha_{t+1}$, we have $F_{t+1} \geq F_t$ so that:

$$\geq k(\delta_m + g_t^2)\min\left[\sqrt{\frac{F_{t+1}(x)}{V_{t+1}}}, \frac{1}{h_t}\right].$$

Next, observe that

$$\frac{d}{d\delta_m}\frac{\delta_m + g_t^2}{\sqrt{V_t + \delta_m + g_t^2}} = \frac{\delta_m^2 + 2V_t + g_t^2}{2(V_t + \delta_m + g_t^2)^{3/2}} \geq 0.$$

Therefore

$$\frac{\delta_m + g_t^2}{\sqrt{V_{t+1}}} = \frac{\delta_m + g_t^2}{\sqrt{V_t + \delta_m + g_t^2}},$$

$$\geq \frac{g_t^2}{\sqrt{V_t + g_t^2}},$$

$$\geq \frac{g_t^2}{\sqrt{V_t}}\sqrt{\frac{V_t}{V_t + g_t^2}},$$

$$\geq \frac{g_t^2}{\sqrt{V_t}}\sqrt{\frac{h_t^2}{h_t^2 + g_t^2}},$$

$$\geq \frac{g_t^2}{\sqrt{2V_t}}.$$

This implies that

$$(\delta_m + g_t^2)\sqrt{\frac{F_t(x)}{V_{t+1}}} \geq \frac{g_t^2}{\sqrt{2}}\sqrt{\frac{F_t(x)}{V_t}}.$$

So, altogether we have:

$$\Psi_{t+1}'(x) - \Psi_t'(x) \geq \frac{kg_t^2}{\sqrt{2}}\min\left[\sqrt{\frac{F_{t+1}(x)}{V_t}}, \frac{1}{h_t}\right],$$

$$\geq g_t^2 \min\left[\sqrt{\frac{F_{t+1}(x)}{V_t}}, \frac{1}{h_t}\right],$$

$$= Z_t'(x)g_t^2,$$

as desired. $\qquad\square$

# F  Fully Unconstrained Learning via Regularization

In this section, we provide a formal description of how to achieve a fully unconstrained bound via application of some peculiar regularization terms, as sketched in Section 3.2.

The goal is to ensure regret given by (4), restated below:

$$\sum_{t=1}^{T} g_t(w_t - u) + a_t(\psi(w_t) - \psi(u)) \le \tilde{O}\left(|u|\sqrt{h_T^2 + \sum_{t=1}^{T} g_t^2} + \psi(u)\sqrt{\gamma^2 + \sum_{t=1}^{T} a_t^2}\right). \quad (4)$$

In Section H, we will see how to obtain the bound (4) via a general technique for obtaining constrained "full-matrix" regret bounds (which is of independent interest). However, this approach comes with a mild computational overhead. To counteract this, in Section E, we provide an alternative approach that has the same computational complexity as gradient descent, but achieves the slightly weaker bound:

$$\sum_{t=1}^{T} g_t(w_t - u) + a_t(\psi(w_t) - \psi(u)) \le \tilde{O}\left(|u|\sqrt{h_T^2 + \sum_{t=1}^{T} g_t^2} + \psi(u)\sqrt{\gamma^2 + \gamma \sum_{t=1}^{T} a_t}\right). \quad (18)$$

Fortunately, (18) will also be sufficient for our purposes.

Armed with an algorithm that achieves (18), we are ready to describe our approach for fully unconstrained learning.

**Corollary 13.** *There exists an online learning algorithm that requires $O(d)$ space and takes $O(d)$ time per update, takes as input scalar values $\gamma$, $h_1$, and $\epsilon$ and ensures that for any sequence $g_1, g_2, \cdots \subset \mathbb{R}^d$, the outputs $w_1, w_1, \cdots \subset \mathbb{R}^d$ satisfy for all $w_\star$ and $T$:*

$$\sum_{t=1}^{T} \langle g_t, w_t - w_\star \rangle \le O\left[\epsilon\sqrt{V} + \|w_\star\|\sqrt{V \log\left(e + \frac{\|w_\star\|}{\epsilon}\right)} + \|w_\star\|G \log\left(e + \frac{\|w_\star\|}{\epsilon}\right)\right.$$
$$\left. + \epsilon^2\gamma\sqrt{\log\left(e + \log\left(e + \frac{G}{h_1}\right)\right)} + \frac{G^2}{\gamma} \log\left(e + \frac{G}{h_1}\right) + \gamma w_\star^2 \log\left(e + \frac{\|w_\star\|^2}{\epsilon^2} \log\left(e + \frac{G}{h_1}\right)\right)\right].$$

*where $G = h_1 + \max_t \|g_t\|$ and $V = G^2 + \sum_{t=1}^{T} \|g_t\|^2$.*

*Proof.* Apply Algorithm 5 with $q = 1$, and REG set to Algorithm 2 using Algorithm 3 with $p = 0$ as BASE. The result in 1 dimension then follows from Theorem 16 and Corollary 6. Then by the reduction from $d$-dimensional online learning to 1-dimensional online learning ([16] Theorem 2), the result in high dimensions also follows. $\square$

**Theorem 1.** *There exists an online learning algorithm that requires $O(d)$ space and takes $O(d)$ time per update, takes as input scalar values $\gamma$, $h_1$, and $\epsilon$ and ensures that for any sequence $g_1, g_2, \cdots \subset \mathbb{R}^d$, the outputs $w_1, w_1, \cdots \subset \mathbb{R}^d$ satisfy for all $w_\star$ and $T$:*

$$\sum_{t=1}^{T} \langle g_t, w_t - w_\star \rangle \le O\left[\epsilon G + \epsilon^2\gamma + \frac{G^2}{\gamma} \log\left(e + \frac{G}{h_1}\right) + \|w_\star\|\sqrt{V \log\left(e + \frac{|w_\star|\sqrt{V} \log^2(T)}{h_1\epsilon}\right)}\right.$$
$$\left. + \|w_\star\|G \log\left(e + \frac{\|w_\star\|\sqrt{V} \log^2(T)}{h_1\epsilon}\right) + \gamma\|w_\star\|^2 \log\left(e + \frac{\|w_\star\|^2}{\epsilon^2} \log\left(e + \frac{G}{h_1}\right)\right)\right],$$

*where $G = \max(h_1, \max_{t \in [T]} \|g_t\|)$ and $V = G^2 + \sum_{t=1}^{T} \|g_t\|^2$.*

*Proof.* Apply Algorithm 5 with $q = 1$, and REG set to Algorithm 2 using Algorithm 3 with $p = 1/2$ as BASE. The result in 1 dimension then follows from Theorem 16 and Corollary 6. Then by the reduction from $d$-dimensional online learning to 1-dimensional online learning ([16] Theorem 2), the result in high dimensions also follows. $\square$

---
**Algorithm 4** Fully Unconstrained Learning in One Dimension
---
**Input:** Regularized learning algorithm REG with domain $\mathbb{R}$. Parameter $\gamma > 0$, $h_1 > 0$.
Initialize REG with parameters $\epsilon$ and $\gamma$.
**for** $t = 1 \ldots T$ **do**
    Send $h_t$ to REG as the $t$th magnitude hint.
    Get $w_t$ from REG.
    Play $w_t$, see feedback $g_t$.
    Set $h_{t+1} = \max(h_t, |g_t|)$.
    Set $\tilde{g}_t = \text{clip}_{[-h_t, h_t]} g_t$
    Set $a_t = \gamma \frac{(h_{t+1} - h_t)/h_{t+1}}{1 + \sum_{i=1}^t (h_{i+1} - h_i)/h_{i+1}}$.
    Send $\tilde{g}_t, a_t$, to REG as $t$th loss and regularization coefficient.
**end for**
---

**Theorem 14.** *There exists an online learning algorithm that requires $O(d)$ space and takes $O(d)$ time per update, takes as input scalar values $\gamma$, $h_1$, and $\epsilon$ and a symmetric convex function $\psi$ and ensures that for any sequence $g_1, g_2, \cdots \subset \mathbb{R}^d$, the outputs $w_1, w_1, \cdots \subset \mathbb{R}^d$ satisfy for all $w_\star$ and $T$:*

$$\sum_{t=1}^{T} \langle g_t, w_t - w_\star \rangle \leq O\left[ \epsilon G + \|w_\star\| \sqrt{V \log\left(e + \frac{\|w_\star\|\sqrt{V}\log^2(T)}{h_1 \epsilon}\right)} + \|w_\star\| G \log\left(e + \frac{\|w_\star\|\sqrt{V}\log^2(T)}{h_1 \epsilon}\right) \right.$$

$$\left. + \psi(\epsilon)\gamma + \gamma\psi(\|w_\star\|)\log\left(e + \frac{\psi(\|w_\star\|)}{\psi(\epsilon)}\log\left(e + \frac{G}{h_1}\right)\right) + \gamma\log\left(1 + \log\left(\frac{G}{h_1}\right)\right)\psi^\star\left(\frac{G}{\gamma}\left[1 + \log\left(\frac{G}{h_1}\right)\right]\right) \right],$$

*where $\psi^\star(\theta) = \sup_w \theta w - \psi(w)$ is the Fenchel conjugate of $\psi$, $G = \max(h_1, \max_t \|g_t\|)$ and $V = G^2 + \sum_{t=1}^T \|g_t\|^2$.*

*Proof.* Apply Algorithm 5 with REG set to Algorithm 2 using Algorithm 3 with $p = 1/2$ as BASE. The result in 1 dimension then follows from Theorem 16 and Corollary 6. Then by the reduction from $d$-dimensional online learning to 1-dimensional online learning ([16] Theorem 2), the result in high dimensions also follows. $\square$

**Theorem 15.** *Suppose $\psi$ is a symmetric convex function. Suppose that so long as $h_t \geq |\tilde{g}_t|$, REG ensures for some $A, B, C, D, p, \epsilon$:*

$$\sum_{t=1}^T \tilde{g}_t(w_t - w_\star) + a_t(\psi(w_t) - \psi(w_\star))$$

$$\leq C\epsilon h_T^{2p} V_g^{1/2-p} + C\psi(\epsilon)\gamma^{2p} S_a^{1/2-p} + A|w_\star|\sqrt{V_g \log\left(e + \frac{D|x_\star|V_g^p}{h_1^{2p}\epsilon}\right)}$$

$$+ Bh_T|w_\star|\log\left(e + \frac{D|w_\star|V_g^p}{h_1^{2p}\epsilon}\right)$$

$$+ A\psi(w_\star)\sqrt{S_a \log\left[e + \frac{D\psi(w_\star)}{\gamma^{2p}\psi(\epsilon)}S_a^p\right]}$$

$$+ \gamma B\psi(w_\star)\log\left[e + \frac{D\psi(w_\star)}{\gamma^{2p}\psi(\epsilon)}S_a^p\right],$$

*where $V_g = h_T^2 + \sum_{t=1}^T \tilde{g}_t^2$ and $S_a = \gamma^2 + \gamma \sum_{t=1}^T a_t$. Then Algorithm 4 ensures:*

$$S_a \leq \gamma^2 + \gamma^2 \log\left(1 + \min\left[\log\left(\frac{h_T}{h_1}\right), T\right]\right),$$

$$V_g \leq h_T^2 + \sum_{t=1}^T g_t^2,$$

*and:*

$$\sum_{t=1}^{T} g_t(w_t - w_\star) \leq C\epsilon h_T^{2p} V_g^{1/2-p} + C\psi(\epsilon)\gamma^{2p} S_a^{1/2-p} + A|w_\star|\sqrt{V_g \log\left(e + \frac{D|w_\star|V_g^p}{h_1^{2p}\epsilon}\right)}$$

$$+ Bh_T|w_\star|\log\left(e + \frac{D|w_\star|V_g^p}{h_1^{2p}\epsilon}\right)$$

$$+ A\psi(w_\star)\sqrt{S_a \log\left[e + \frac{D\psi(w_\star)}{\gamma^{2p}\psi(\epsilon)}S_a^p\right]}$$

$$+ \gamma B\psi(w_\star)\log\left[e + \frac{D\psi(w_\star)}{\gamma^{2p}\psi(\epsilon)}S_a^p\right]$$

$$+ h_T|u| + \psi(w_\star)S_a$$

$$+ \gamma \log\left(1 + \min\left[\log\left(\frac{h_T}{h_1}\right), T\right]\right)\psi^\star\left(\frac{h_T}{\gamma}\left[1 + \log\left(\frac{h_T}{h_1}\right)\right]\right)$$

*In the special case that* $\psi(x) = \frac{|x|^{1+q}}{1+q}$, *we can replace the final term* $\gamma \log\left(1 + \min\left[\log\left(\frac{h_T}{h_1}\right), T\right]\right)\psi^\star\left(h_T\left[1 + \log\left(\frac{h_T}{h_1}\right)\right]\right)$ *in the above expression by:*

$$\frac{h_T^{1+1/q}\left[1 + \log\left(\frac{h_T}{h_1}\right)\right]^{1/q}}{(1+1/q)\gamma^{1/q}}.$$

*Proof.* We have:

$$\sum_{t=1}^{T} g_t(w_t - u)$$

$$= \sum_{t=1}^{T} \tilde{g}_t(w_t - u) + a_t(\psi(w_t) - \psi(u)) + a_t\psi(u) + (g_t - \tilde{g}_t)(w_t - u) - a_t\psi(w_t),$$

$$\leq \psi(u)\sum_{t=1}^{T} a_t + |u|\sum_{t=1}^{T}|g_t - \tilde{g}_t| + \sum_{t=1}^{T}\tilde{g}_t(w_t - u) + a_t(\psi(w_t) - \psi(u)),$$

$$+ \sum_{t=1}^{T}|g_t - \tilde{g}_t||w_t| - a_t\psi(w_t)$$

Observing that $|g_t - \tilde{g}_t| = h_{t+1} - h_t$:

$$= \psi(u)\sum_{t=1}^{T} a_t + |u|\sum_{t=1}^{T}[h_{t+1} - h_t] + \sum_{t=1}^{T}(h_{t+1} - h_t)|w_t| - a_t\psi(w_t) + \sum_{t=1}^{T}\tilde{g}_t(w_t - u) + a_t(\psi(w_t) - \psi(u)).$$

Next, we will bound the terms $\sum_{t=1}^{T} a_t\psi(u)$ and $|u|\sum_{t=1}^{T}[|g_t| - h_t]_+..$ Moreover, $h_t = h_{t-1} + [|g_t| - h_t]_+$, so that $|u|\sum_{t=1}^{T}|g_t - \tilde{g}_t| \leq |u|h_T$.

Further, notice that for any $s_0, s_1, \ldots, s_T$, $\sum_{t=1}^{T} \log\left(\frac{s_t}{\sum_{i=0}^{t} s_i}\right) \leq \log(s_T/s_0)$, so that:

$$\sum_{t=1}^{T} a_t \leq \gamma \log\left(1 + \sum_{t=1}^{T}\frac{h_{t+1} - h_t}{h_{t+1}}\right)$$

Notice that $[|g_t| - h_t]_+/h_{t+1} \leq 1$, so we also have:

$$\sum_{t=1}^{T}\frac{h_{t+1} - h_t}{h_{t+1}} \leq \min\left[\log\left(\frac{h_T}{h_1}\right), T\right]$$

so that overall:

$$S_a = \gamma^2 + \gamma\sum_{t=1}^{T} a_t \leq \gamma^2 + \gamma^2 \log\left(1 + \min\left[\log\left(\frac{h_T}{h_1}\right), T\right]\right)$$

Next, we bound the terms $(h_{t+1} - h_t)|w_t| - a_t\psi(w_t)$. Let $\psi^\star(w)$ be the Fenchel conjugate of $\psi$. Recall that $\psi$ is symmetric so that $\psi(w_t) = \psi(|w_t|)$. This also implies that $\psi^\star$ is symmetric and is minimized at zero. Thus:

$$
\begin{aligned}
(h_{t+1} - h_t)|w_t| - a_t\psi(w_t) &= (h_{t+1} - h_t)|w_t| - a_t\psi(|w_t|), \\
&= a_t\psi^\star\left(\frac{h_{t+1} - h_t}{a_t}\right), \\
&= a_t\psi^\star\left(\frac{h_{t+1}}{\gamma}\left[1 + \sum_{i=1}^{t}\frac{h_{i+1} - h_i}{h_{i+1}}\right]\right).
\end{aligned}
$$

So, in general we have:

$$
\begin{aligned}
\sum_{t=1}^{T}(h_{t+1} - h_t)|w_t| - a_t\psi(w_t) &\leq \sum_{t=1}^{T} a_t\psi^\star\left(\frac{h_{t+1}}{\gamma}\left[1 + \sum_{i=1}^{t}\frac{h_{i+1} - h_i}{h_{i+1}}\right]\right), \\
&\leq \sum_{t=1}^{T} a_t\psi^\star\left(\frac{h_T}{\gamma}\left[1 + \log\left(\frac{h_T}{h_1}\right)\right]\right), \\
&\leq \gamma\log\left(1 + \min\left[\log\left(\frac{h_T}{h_1}\right), T\right]\right)\psi^\star\left(\frac{h_T}{\gamma}\left[1 + \log\left(\frac{h_T}{h_1}\right)\right]\right).
\end{aligned}
$$

In the special case that $\psi(w) = \frac{|w|^{1+q}}{1+q}$, we have $\psi^\star(h) = \frac{h^{1+1/q}}{1+1/q}$ so that we can improve the logarithmic factors and simplify the calculation:

$$
\begin{aligned}
a_t\psi^\star\left(h_{t+1}\left[1 + \sum_{i=1}^{t}\frac{h_{i+1} - h_i}{h_{i+1}}\right]\right) &= \frac{a_t h_{t+1}^{1+1/q}\left[1 + \sum_{i=1}^{t}\frac{h_{i+1} - h_i}{h_{i+1}}\right]^{1+1/q}}{(1 + 1/q)\gamma^{1+1/q}}, \\
&= \frac{(h_{t+1} - h_t)h_{t+1}^{1/q}\left[1 + \sum_{i=1}^{t}\frac{h_{i+1} - h_i}{h_{i+1}}\right]^{1/q}}{(1 + 1/q)\gamma^{1/q}}, \\
&= \frac{(h_{t+1} - h_t)h_{t+1}^{1/q}\left[1 + \sum_{i=1}^{t}\frac{h_{i+1} - h_i}{h_{i+1}}\right]^{1/q}}{(1 + 1/q)\gamma^{1/q}}, \\
&\leq \frac{(h_{t+1} - h_t)h_T^{1/q}\left[1 + \log\left(\frac{h_T}{h_1}\right)\right]^{1/q}}{(1 + 1/q)\gamma^{1/q}} \\
\sum_{t=1}^{T}(h_{t+1} - h_t)|w_t| - a_t\psi(w_t) &\leq \frac{h_T^{1+1/q}\left[1 + \log\left(\frac{h_T}{h_1}\right)\right]^{1/q}}{(1 + 1/q)\gamma^{1/q}}.
\end{aligned}
$$

Finally, it is clear that $|\tilde{g}_t| \leq h_t$ so the summation $\sum_{t=1}^{T} \tilde{g}_t(w_t - u) + a_t(\psi(w_t) - \psi(u))$ is controlled by the regret bound of REG:

$$
\begin{aligned}
\sum_{t=1}^{T} \tilde{g}_t(w_t - u) + a_t(\psi(w_t) - \psi(u)) &\leq C\epsilon h_T^{2p}V_g^{1/2-p} + C\psi(\epsilon)\gamma^{2p}S_a^{1/2-p} + A|x_\star|\sqrt{V_g\log\left(e + \frac{D|x_\star|V_g^p}{h_1^{2p}\epsilon}\right)} \\
&\quad + Bh_T|x_\star|\log\left(e + \frac{D|x_\star|V_g^p}{h_1^{2p}\epsilon}\right) \\
&\quad + A\psi(x_\star)\sqrt{S_a\log\left[e + \frac{D\psi(x_\star)}{\gamma^{2p}\psi(\epsilon)}S_a^p\right]} \\
&\quad + \gamma B\psi(x_\star)\log\left[e + \frac{D\psi(x_\star)}{\gamma^{2p}\psi(\epsilon)}S_a^p\right].
\end{aligned}
$$

---
**Algorithm 5** Fully Unconstrained Learning
---
**Input:** Symmetric convex function $\psi : \mathbb{R} \to \mathbb{R}$ with $0 = \psi(0)$. Scalars $\epsilon > 0$, $h_1$, $\gamma > 0$, $p \in [0, 1/2]$.
Let REG be an instance of Algorithm 6 with input $\psi$, $\gamma$, $p$, $\epsilon_x = \epsilon$ and $\epsilon_\psi = \psi(\epsilon)$.
Set vector $\vec{w}_1^{direction} = 0$
Send $h_1$ to REG as the first magnitude hint.
**for** $t = 1 \ldots T$ **do**
    // Apply reduction to 1-dimensional learning from [16] using adaptive gradient descent as "direction learner".
    Let $w_t^{magnitude} \in \mathbb{R}$ be the $t$th output of REG.
    Set $\vec{w}_t = w_t^{magnitude} \cdot \vec{w}_t^{direction}$
    Play $w_t$, see feedback $g_t$.
    Set $\vec{w}_{t+1}^{direction} = \Pi_{\|w\| \leq 1} \left[ \vec{w}_t^{direction} - \frac{g_t}{\sqrt{2 \sum_{i=1}^t \|g_i\|^2}} \right]$.
    // Compute feedback for "magnitude learner"
    Set $g_t^{1d} = \langle g_t, d_t \rangle$
    // Apply our new fully unconstrained magnitude learner.
    Set $h_{t+1} = \max(h_t, |g_t^{1d}|)$.
    Set $\tilde{g}_t = \text{clip}_{[-h_t, h_t]} g_t^{1d}$
    Set $a_t = \gamma \frac{(h_{t+1}-h_t)/h_{t+1}}{1+\sum_{i=1}^t (h_{i+1}-h_i)/h_{i+1}}$.
    Send $\tilde{g}_t, a_t$ to REG as $t$th loss and regularization coefficient.
    Send $h_{t+1}$ to REG as the $t + 1$st magnitude hint.
**end for**
---

Finally, we also have:

$$V_g = h_T^2 + \sum_{t=1}^T \tilde{g}_t^2,$$

$$\leq h_T^2 + \sum_{t=1}^T g_t^2.$$

$\square$

## F.1 Full Statement of Main Result in High Dimensions

Throughout this paper, we have considered the special case that $\mathcal{W} = \mathbb{R}$. This suffices due to the reductions of [16] as discussed in Section C. However, here we provide a more complete theorem and algorithm for the case $\mathcal{W} = \mathbb{R}^d$. The pseudocode is provided in Algorithm 5, and the regret bound is stated in Theorem 16. Note that the regret bound follows essentially immediately from Theorem 15.

**Theorem 16.** *There exists universal constants A, B, C, such that Algorithm 5 guarantees for all $T$:*

$$\sum_{t=1}^T \langle g_t, w_t - w_\star \rangle \leq C \epsilon h_T^{2p} V_g^{1/2-p} + C \psi(\epsilon) \gamma^{2p} S_a^{1/2-p} + A \|w_\star\| \sqrt{V_g \log \left( e + \frac{\|w_\star\| V_g^p}{h_1^{2p} \epsilon} \right)}$$

$$+ B h_T \|w_\star\| \log \left( e + \frac{\|w_\star\| V_g^p}{h_1^{2p} \epsilon} \right)$$

$$+ A \psi(\|w_\star\|) \sqrt{S_a \log \left[ e + \frac{\psi(\|w_\star\|)}{\gamma^{2p} \psi(\epsilon)} S_a^p \right]}$$

$$+ \gamma B \psi(\|w_\star\|) \log \left[ e + \frac{\psi(\|w_\star\|)}{\gamma^{2p} \psi(\epsilon)} S_a^p \right]$$

$$+ h_T |u| + \psi(\|w_\star\|) S_a$$

$$+ \gamma \log \left( 1 + \min \left[ \log \left( \frac{h_T}{h_1} \right), T \right] \right) \psi^\star \left( \frac{h_T}{\gamma} \left[ 1 + \log \left( \frac{h_T}{h_1} \right) \right] \right)$$

*where*

$$S_a \leq \gamma^2 + \gamma^2 \log\left(1 + \min\left[\log\left(\frac{h_T}{h_1}\right), T\right]\right)$$

$$V_g \leq h_T^2 + \sum_{t=1}^{T} g_t^2$$

*In the special case that* $\psi(x) = \frac{|x|^{1+q}}{1+q}$, *we can replace the final term* $\gamma \log\left(1 + \min\left[\log\left(\frac{h_T}{h_1}\right), T\right]\right)\psi^\star\left(h_T\left[1 + \log\left(\frac{h_T}{h_1}\right)\right]\right)$ *in the above expression by:*

$$\frac{h_T^{1+1/q}\left[1 + \log\left(\frac{h_T}{h_1}\right)\right]^{1/q}}{(1+1/q)\gamma^{1/q}}.$$

*Proof.* Algorithm 5 is applying the dimension-free-to-one-dimension reduction provided by Theorem 2 of [16]. So overall the reduction tells us that the regret is bounded by

$$\sum_{t=1}^{T}\langle g_t, w_t - w_\star\rangle \leq \sum_{t=1}^{T}\langle g_t^{1d}, w_t^{magnitude} - \|w_\star\|\rangle + \|w_\star\|\sum_{t=1}^{T}\langle g_t, w_t^{direction} - w_\star/\|w_\star\|\rangle$$

In this case, the "direction" learner's iterates $w_t^{direction}$ are generated by standard adaptive gradient descent [13], which guarantees the regret bound: $\sum_{t=1}^{T}\langle g_t, w_t^{direction} - w_\star/\|w_\star\|\rangle \leq 2\sqrt{2\sum_{t=1}^{T}\|g_t\|^2}$.

For the first sum $\sum_{t=1}^{T}\langle g_t^{1d}, w_t^{magnitude} - \|w_\star\|\rangle$, notice that $w^{magnitude}$ is simply an application of Algorithm 4 using an instance of Algorithm 6 The first sum is bounded by application of Theorem 15, noticing that $|g_t^{1d}| \leq \|g_t\|$. So, putting the two bounds together we have the stated result. $\qquad\square$

## G  Technical Lemmas

**Lemma 17.** *Let A, B, C, D, E be positive numbers and let e be the base of the natural logarithm. Then:*

$$\sup_{M} A\sqrt{M\log(e + DM^C)} + B\log(e + DM^C) - EM \leq \left(\frac{A^2}{E} + B\right)\log\left(e + D\left(\frac{2CA^2}{E^2} + \frac{2CB}{E}\right)^C\right)$$

*Proof.* First, by Young inequality $xy \leq \inf_\lambda x^2/2\lambda + \lambda y^2/2$, we have for all $M$:

$$M\log(e + DM^C) \leq \frac{M^2 E^2}{4A^2} + \frac{A^2 \log^2(e + DM^2)}{E^2}$$

Then using the identity $\sqrt{x + y} \leq \sqrt{x} + \sqrt{y}$:

$$\sup_{M} A\sqrt{M\log(e + DM^C)} + B\log(e + DM^C) - EM \leq \sup_{M}\left(\frac{A^2}{E} + B\right)\log(e + DM^C) - \frac{EM}{2}$$

Now, from first order optimality conditions we are looking for a solution to:

$$\frac{\left(\frac{A^2}{E} + B\right)DCM^{C-1}}{e + DM^C} = \frac{E}{2}$$

$$\left(\frac{A^2}{E} + B\right)DCM^{C-1} = \frac{E}{2}e + \frac{E}{2}DM^C$$

Notice that for any $M \geq \frac{2CA^2}{E^2} + \frac{2CB}{E}$ we have:

$$\left(\frac{A^2}{E} + B\right)CD \leq \frac{E}{2}DM$$

$$\left(\frac{A^2}{E} + B\right)DCM^{C-1} \leq \frac{E}{2}DM^C$$

$$\left(\frac{A^2}{E} + B\right)DCM^{C-1} < \frac{E}{2}e + \frac{E}{2}DM^C$$

Therefore, the optimal value for $M$ can be at most $\frac{2CA^2}{E^2} + \frac{2CB}{E}$. Now, notice that $\left(\frac{A^2}{E} + B\right)\log(e + DM^C)$ is strictly increasing in $M$. Thus, our quantity of interest is upper-bounded by substituting in $M = \frac{2CA^2}{E^2} + \frac{2CB}{E}$ into this increasing term:

$$\sup_M \left(\frac{A^2}{E} + B\right)\log(e + DM^C) - EM \leq \left(\frac{A^2}{E} + B\right)\log\left(e + D\left(\frac{2CA^2}{E^2} + \frac{2CB}{E}\right)^C\right)$$

$\square$

**Lemma 18.** *Let $A$, $B < 1$, $C$ be positive numbers. Then:*

$$\sup_M AM^B - CM = \left(\frac{BA}{C^B}\right)^{1/(1-B)}\left(\frac{1-B}{B}\right)$$

*Proof.* We differentiate with respect to $M$:

$$ABM^{B-1} = C$$

$$M = \left(\frac{C}{AB}\right)^{1/(B-1)}$$

So, plugging in this optimal $M$ value we have: $\sup_M AM^B - CM = \left(\frac{C^B}{BA}\right)^{1/(B-1)}\left(\frac{1}{B} - 1\right)$ $\square$

**Lemma 19.** *Let $A$, $B$, $C$, $D$, $E$, $F$, $G < 1$ be positive numbers and let $e$ be the base of the natural logarithm. Then:*

$$\sup_M A\sqrt{M\log\left(e + DM^C\right)} + B\log\left(e + DM^C\right) + FM^G - \frac{EM}{2}$$

$$\leq \left(\frac{4A^2}{E} + B\right)\log\left(e + D\left(\frac{32CA^2}{E^2} + \frac{8CB}{E}\right)^C\right) + \left(\frac{4^G GF}{E^G}\right)^{1/(1-G)}\left(\frac{1-G}{G}\right)$$

*When $G = 0$, the last term $\left(\frac{4^G GF}{E^G}\right)^{1/(1-G)}\left(\frac{1}{G} - 1\right)$ should be replaced with the limiting value $F$.*

*Proof.* Notice that

$$\sup_M A\sqrt{M\log\left(e + DM^C\right)} + Blog\left(e + DM^C\right) + FM^G - \frac{EM}{2}$$

$$\leq \sup_M A\sqrt{M\log\left(e + DM^C\right)} + B\log\left(e + DM^C\right) - \frac{EM}{4} + \sup_M FM^G - \frac{EM}{4}$$

The result now follows from Lemmas 17 and 18. Alternatively, if $G = 0$, clearly $\sup_M FM^G - \frac{EM}{4} = F$. $\square$

**Lemma 20.** *Let $\psi$, $A$, $B$, $C$, $D$, $F$, $S$, $\gamma$, and $p \leq 1/2$ be positive numbers with $S \geq \gamma^2$, and let $e$ be the base of the natural logarithm. Then:*

$$\inf_E \sup_V A(\psi + E)\sqrt{V\log\left(e + \frac{D(\psi + E)}{\gamma^{2p}}V^p\right)} + B\gamma(\psi + E)\log\left(e + \frac{D(\psi + E)}{\gamma^{2p}}V^p\right) + F\gamma^{2p}V^{1/2-p} - \frac{EV}{2\gamma} + \frac{ES}{2\gamma}$$

$$\leq (1/2 + 16A^2)\psi\sqrt{S\log\left(e + \frac{2D\psi S^p}{\gamma^{2p}}(128pA^2\psi + 16pB)^p\right)}$$

$$+ \gamma\psi\left(16A^2 + 2B\right)\log\left(e + \frac{2D\psi S^p}{\gamma^{2p}}(128pA^2\psi + 16pB)^p\right)$$

$$+ \left((16A^2 + 2B)\log\left(e + 2DF\left(128pA^2 + 16pB\right)^p\right) + \frac{1}{2} + \frac{(2p+1)(2-4p)^{\frac{1-2p}{1+2p}}}{2}\right)\gamma^{2p}FS^{1/2-p}$$

*Proof.* By Lemma 19, we have:

$$\sup_V A(\psi + E)\sqrt{V\log\left(e + \frac{D(\psi+E)}{\gamma^{2p}}M^p\right)} + B\gamma(\psi+E)\log\left(e + \frac{D(\psi+E)}{\gamma^{2p}}M^p\right) + F\gamma^{2p}V^{1/2-p} - \frac{EV}{2\gamma}$$

$$\leq \left(\frac{4A^2(\psi+E)^2\gamma}{E} + B\gamma(\psi+E)\right)\log\left(e + \frac{D(\psi+E)}{\gamma^{2p}}\left(\frac{32p\gamma^2A^2(\psi+E)^2}{E^2} + \frac{8p\gamma^2B(\psi+E)}{E}\right)^p\right)$$

$$+ \left(\frac{4^{1/2-p}(1/2-p)F\gamma^{2p}}{(E/\gamma)^{1/2-p}}\right)^{\frac{2}{1+2p}}\frac{2p+1}{1-2p}$$

$$= \underbrace{\left(\frac{4A^2(\psi+E)^2\gamma}{E} + B\gamma(\psi+E)\right)\log\left(e + D(\psi+E)\left(\frac{32pA^2(\psi+E)^2}{E^2} + \frac{8pB(\psi+E)}{E}\right)^p\right)}_{(*)}$$

$$+ \underbrace{\gamma\frac{(2p+1)(2-4p)^{\frac{1-2p}{1+2p}}}{2}\left(\frac{F}{E^{1/2-p}}\right)^{\frac{2}{1+2p}}}_{(**)}$$

Now, set:

$$E = \max\left[\min\left[\psi, \frac{\gamma\psi}{\sqrt{S}}\sqrt{\log\left(e + \frac{2D\psi S^p}{\gamma^{2p}}(128pA^2 + 16pB)^p\right)}\right], \frac{\gamma^{1+2p}F}{S^{\frac{1+2p}{2}}}\right]$$

We will bound the above expression by first considering $(*)$ and then $(**)$. Now, if $E = \psi$, we have:

$$(*) = \gamma\psi\left(16A^2 + 2B\right)\log\left(e + 2D\psi\left(128pA^2 + 16pB\right)^p\right)$$

recalling that $S \geq \gamma^2$:

$$\leq \gamma\psi\left(16A^2 + 2B\right)\log\left(e + \frac{2D\psi S^p}{\gamma^{2p}}(128pA^2 + 16pB)^p\right)$$

Alternatively, if $E = \frac{\gamma\psi}{\sqrt{S}}\sqrt{\log\left(e + \frac{2D\psi S^p}{\gamma^{2p}}(128pA^2 + 16pB)^p\right)}$, then we have $E + \psi \leq 2\psi$ and so:

$$(*) \leq \left(\frac{16A^2\psi^2\gamma}{E} + 2B\gamma\psi\right)\log\left(e + 2D\psi\left(\frac{128pA^2\psi^2}{E^2} + \frac{16pB\psi}{E}\right)^p\right) \tag{19}$$

Before we bound this expression, let us consider just the value inside the logarithm:

$$\frac{128pA^2\psi^2}{E^2} + \frac{16pB\psi}{E} \leq 128pA^2\psi\frac{S}{\gamma^2} + 16pB\frac{\sqrt{S}}{\gamma}$$

now, since $S \geq \gamma^2$:

$$\leq (128pA^2\psi + 16pB)\frac{S}{\gamma^2}$$

So, putting this back in the previous expression:

$$\log\left(e + 2D\psi\left(\frac{128pA^2\psi^2}{E^2} + \frac{16pB\psi}{E}\right)^p\right) \leq \log\left(e + \frac{2D\psi S^p}{\gamma^{2p}}(128pA^2\psi + 16pB)^p\right)$$

from which we conclude:

$$(*) \leq 16A^2\psi\sqrt{S\log\left(e + \frac{2D\psi S^p}{\gamma^{2p}}(128pA^2\psi + 16pB)^p\right)}$$

$$+ 2B\gamma\psi\log\left(e + \frac{2D\psi S^p}{\gamma^{2p}}(128pA^2\psi + 16pB)^p\right)$$

Finally, let us consider the case $E = \frac{\gamma^{1+2p}F}{S^{\frac{1+2p}{2}}}$. To handle this situation, we will work with two more subcases: either $E \le \psi$ or not. If $E \le \psi$, then $E + \psi \le 2\psi$. Therefore:

$$(*) \le \left( \frac{16A^2\psi^2\gamma}{E} + 2B\gamma\psi \right) \log \left( e + 2D\psi \left( \frac{128pA^2\psi^2}{E^2} + \frac{16pB\psi}{E} \right)^p \right)$$

However, if $E \le \psi$, then it must be that $E \ge \frac{\gamma\psi}{\sqrt{S}} \sqrt{\log \left( e + \frac{2D\psi S^p}{\gamma^{2p}} (128pA^2 + 16pB)^p \right)}$. Thus by the exact same analysis following equation (19), we again have

$$(*) \le 16A^2\psi \sqrt{S \log \left( e + \frac{2D\psi S^p}{\gamma^{2p}} (128pA^2\psi + 16pB)^p \right)}$$
$$+ 2B\gamma\psi \log \left( e + \frac{2D\psi S^p}{\gamma^{2p}} (128pA^2\psi + 16pB)^p \right)$$

So, for our final subcase we consider $E = \frac{\gamma^{1+2p}F}{S^{\frac{1+2p}{2}}}$ and also $E \ge \psi$. Then $E + \psi \le 2E$, which yields:

$$(*) \le \gamma E (16A^2 + 2B) \log \left( e + 2DE \left( 128pA^2 + 16pB \right)^p \right)$$

Since $S \ge \gamma^2$, $E \le F$ and so:

$$\le \gamma F (16A^2 + 2B) \log \left( e + 2DF \left( 128pA^2 + 16pB \right)^p \right)$$

So, in all cases we have:

$$(*) \le 16A^2\psi \sqrt{S \log \left( e + \frac{2D\psi S^p}{\gamma^{2p}} (128pA^2\psi + 16pB)^p \right)}$$
$$+ \gamma\psi \left( 16A^2 + 2B \right) \log \left( e + \frac{2D\psi S^p}{\gamma^{2p}} (128pA^2\psi + 16pB)^p \right)$$
$$+ \gamma F (16A^2 + 2B) \log \left( e + 2DF \left( 128pA^2 + 16pB \right)^p \right)$$

Where the last term $\gamma F (16A^2 + 2B) \log \left( e + 2DF \left( 128pA^2 + 16pB \right)^p \right)$ is only present if $p \ne 1/2$.

Notice that we must have $E \ge \frac{\gamma^{1+2p}F}{S^{\frac{1+2p}{2}}}$. Therefore:

$$\gamma \frac{F^{\frac{2}{1+2p}}}{E^{\frac{1-2p}{1+2p}}} \le \gamma^{2p} F S^{1/2-p}$$
$$(**) \le \frac{(2p+1)(2-4p)^{\frac{1-2p}{1+2p}}}{2} \gamma^{2p} F S^{1/2-p}$$

So, overall it holds that:

$$(*) + (**) \le 16A^2\psi \sqrt{S \log \left( e + \frac{2D\psi S^p}{\gamma^{2p}} (128pA^2\psi + 16pB)^p \right)}$$
$$+ \gamma\psi \left( 16A^2 + 2B \right) \log \left( e + \frac{2D\psi S^p}{\gamma^{2p}} (128pA^2\psi + 16pB)^p \right)$$
$$+ \gamma F (16A^2 + 2B) \log \left( e + 2DF \left( 128pA^2 + 16pB \right)^p \right)$$
$$+ \frac{(2p+1)(2-4p)^{\frac{1-2p}{1+2p}}}{2} \gamma^{2p} F S^{1/2-p}$$

To conclude, let us bound $\frac{ES}{2\gamma}$. If $E \ne \frac{\gamma^{1+2p}F}{S^{\frac{1+2p}{2}}}$, then it must be that $E \le \frac{\gamma\psi}{\sqrt{S}} \sqrt{\log \left( e + \frac{2D\psi S^p}{\gamma^{2p}} (128pA^2 + 16pB)^p \right)}$. Therefore:

$$\frac{ES}{2\gamma} \le \frac{\psi}{2} \sqrt{S \log \left( e + \frac{2D\psi S^p}{\gamma^{2p}} (128pA^2 + 16pB)^p \right)}$$

Alternatively, if $E = \frac{\gamma^{1+2p}F}{S^{\frac{1+2p}{2}}}$. In this case:

$$\frac{ES}{2\gamma} = \frac{\gamma^{2p}FS^{1/2-p}}{2}$$

So, combining all these facts, we have when $p < 1/2$:

$$\inf_E \sup_V A(\psi + E)\sqrt{V\log\left(e + \frac{D(\psi + E)}{\gamma^{2p}}M^p\right)} + B\gamma(\psi + E)\log\left(e + \frac{D(\psi + E)}{\gamma^{2p}}M^p\right) + F\gamma^{2p}V^{1/2-p} - \frac{EV}{2\gamma} + \frac{ES}{2\gamma}$$

$$\leq \inf_E (*) + (**) + \frac{ES}{2\gamma}$$

$$\leq 16A^2\psi\sqrt{S\log\left(e + \frac{2D\psi S^p}{\gamma^{2p}}(128pA^2\psi + 16pB)^p\right)}$$

$$+ \gamma\psi\left(16A^2 + 2B\right)\log\left(e + \frac{2D\psi S^p}{\gamma^{2p}}(128pA^2\psi + 16pB)^p\right)$$

$$+ \gamma F(16A^2 + 2B)\log\left(e + 2DF\left(128pA^2 + 16pB\right)^p\right)$$

$$+ \frac{(2p+1)(2-4p)^{\frac{1-2p}{1+2p}}}{2}\gamma^{2p}FS^{1/2-p}$$

$$+ \frac{\gamma^{2p}FS^{1/2-p}}{2} + \frac{\psi}{2}\sqrt{S\log\left(e + \frac{2D\psi S^p}{\gamma^{2p}}\left(128pA^2 + 16pB\right)^p\right)}$$

grouping terms, and using $\gamma \leq \gamma^{2p}S^{1/2-p}$:

$$\leq (1/2 + 16A^2)\psi\sqrt{S\log\left(e + \frac{2D\psi S^p}{\gamma^{2p}}(128pA^2\psi + 16pB)^p\right)}$$

$$+ \gamma\psi\left(16A^2 + 2B\right)\log\left(e + \frac{2D\psi S^p}{\gamma^{2p}}(128pA^2\psi + 16pB)^p\right)$$

$$+ \left((16A^2 + 2B)\log\left(e + 2DF\left(128pA^2 + 16pB\right)^p\right) + \frac{1}{2} + \frac{(2p+1)(2-4p)^{\frac{1-2p}{1+2p}}}{2}\right)\gamma^{2p}FS^{1/2-p}$$

$$\square$$

**Lemma 21.** *Suppose $g_1, \ldots, g_t$ and $0 < h_1 \leq h_2 \leq \cdots \leq h_T$ are such that $|g_t| \leq h_t$ for all t. Define $V_t = ch_t^2 + g_{1:t-1}^2$. Define $\alpha_t = \frac{\epsilon}{\sqrt{c + \sum_{i=1}^{t-1} g_i^2/h_i^2}\log^2\left(c + \sum_{i=1}^{t-1} g_i^2/h_i^2\right)}$ for some $c \geq 3$. Then:*

$$\sum_{t=1}^T \frac{\alpha_t g_t^2}{\sqrt{V_t}} \leq = 4\epsilon h_T$$

*Proof.* Let $1 = \tau_1, \ldots, \tau_k \leq T$ be the set of indices such that $h_{\tau_{i+1}} > 2h_{\tau_i}$ and $h_{\tau_{i+1}-1} \leq 2h_{\tau_i}$, with $\tau_{k+1}$ defined equal to $T + 1$ for convenience. Note that this implies $h_{\tau_{k-1}} < h_{\tau_k}/2^i$. Further, $h_{\tau_k} \leq h_T$, so overall we have for all $i$ $h_{\tau_{k-i}} \leq h_T/2^i$. We will show that

$$\sum_{t=\tau_i}^{\tau_{i+1}-1} \frac{\alpha_t g_t^2}{\sqrt{V_t}} \leq 2\epsilon h_{\tau_i} \tag{20}$$

Once established, this implies:

$$\sum_{t=1}^{T} \frac{\alpha_t g_t^2}{\sqrt{V_t}} = \sum_{i=1}^{k} \sum_{t=\tau_i}^{\tau_{i+1}-1} \frac{\alpha_t g_t^2}{\sqrt{V_t}}$$

$$\le 2\epsilon \sum_{i=1}^{k} h_{\tau_i}$$

$$= 2\epsilon \sum_{i=0}^{k-1} h_{\tau_{k-i}}$$

$$\le 2\epsilon h_T \sum_{i=0}^{k-1} 2^{-i}$$

$$= 4\epsilon h_T$$

So, to establish (20), we observe that for any $t \in [\tau_i, \tau_{i+1} - 1]$ we have

$$V_t \ge (c-1)h_t^2 + \sum_{j=\tau_i}^{t} g_j^2$$

$$\ge \frac{c-1}{2} h_{\tau_{i+1}-1}^2 + \sum_{j=\tau_i}^{t} g_j^2$$

and also:

$$\alpha_t \le \frac{\epsilon}{\sqrt{c-1 + \sum_{j=\tau_i}^{t} g_j^2/h_j^2} \log^2 \left(c-1 + \sum_{j=\tau_i}^{t} g_j^2/h_j^2\right)}$$

$$\le \frac{\epsilon}{\sqrt{\frac{c-1}{2} + \sum_{j=\tau_i}^{t} g_j^2/h_{\tau_{i+1}-1}^2} \log^2 \left(\frac{c-1}{2} + \sum_{j=\tau_i}^{t} g_j^2/h_{\tau_{i+1}-1}^2\right)}$$

$$\le \frac{\epsilon h_{\tau_{i+1}-1}}{\sqrt{\frac{c-1}{2} h_{\tau_{i+1}-1}^2 + \sum_{j=\tau_i}^{t} g_j^2} \log^2 \left(\frac{c-1}{2} + \sum_{j=\tau_i}^{t} g_j^2/h_{\tau_{i+1}-1}\right)}$$

Combining these yields:

$$\frac{\alpha_t g_t^2}{\sqrt{V_t}} \le \frac{\epsilon h_{\tau_{i+1}-1} g_t^2}{\left(\frac{c-1}{2} h_{\tau_{i+1}-1}^2 + \sum_{j=\tau_i}^{t} g_j^2\right) \log^2 \left(\frac{c-1}{2} + \sum_{j=\tau_i}^{t} g_j^2/h_{\tau_{i+1}-1}\right)}$$

$$= \epsilon h_{\tau_{i+1}-1} \frac{g_t^2/h_{\tau_{i+1}-1}^2}{\left(\frac{c-1}{2} + \sum_{j=\tau_i}^{t} g_j^2/h_{\tau_{i+1}-1}^2\right) \log^2 \left(\frac{c-1}{2} + \sum_{j=\tau_i}^{t} g_j^2/h_{\tau_{i+1}-1}\right)}$$

using $c \ge 3$

$$\le \epsilon h_{\tau_{i+1}-1} \frac{g_t^2/h_{\tau_{i+1}-1}^2}{\left(1 + \sum_{j=\tau_i}^{t} g_j^2/h_{\tau_{i+1}-1}^2\right) \log^2 \left(1 + \sum_{j=\tau_i}^{t} g_j^2/h_{\tau_{i+1}-1}\right)}$$

$$\le 2\epsilon h_{\tau_i} \frac{g_t^2/h_{\tau_{i+1}-1}^2}{\left(1 + \sum_{j=\tau_i}^{t} g_j^2/h_{\tau_{i+1}-1}^2\right) \log^2 \left(1 + \sum_{j=\tau_i}^{t} g_j^2/h_{\tau_{i+1}-1}\right)}$$

So, now if we define $x_s = g_{s+\tau_i-1}^2/h_{\tau_{i+1}-1}^2$, then we have:

$$\sum_{t=\tau_i}^{\tau_{i+1}-1} \le 2\epsilon h_{\tau_i} \sum_{s=1}^{\tau_{i+1}-\tau_i} \frac{x_s}{\left(1 + \sum_{s'=1}^{s} x_{s'}\right) \log^2 \left(1 + \sum_{s'=1}^{s} x_{s'}\right)}$$

And, by [1] Lemma 4.13:

$$\sum_{s=1}^{\tau_{i+1}-\tau_i} \frac{x_s}{\left(1 + \sum_{s'=1}^{s} x_{s'}\right) \log^2 \left(1 + \sum_{s'=1}^{s} x_{s'}\right)} \le \int_{0}^{\sum_{s=1}^{\tau_{i+1}-\tau_i} x_s} \frac{dx}{(1+x) \log^2(1+x)}$$

$$= \left. \frac{-1}{\log(1+x)} \right|_{0}^{\sum_{s=0}^{\tau_{i+1}-\tau_i} x_s}$$

$$\le 1$$

So, in the end we have $\sum_{t=\tau_i}^{\tau_{i+1}-1} \le 2\epsilon h_{\tau_i}$ as desired. $\qquad \square$

**Lemma 22.** *Suppose $g_1, \ldots, g_t$ and $0 < h_1 \le h_2 \le \cdots \le h_T$ are such that $|g_t| \le h_t$ for all $t$. Define $V_t = ch_t^2 + g_{1:t-1}^2$. Define $\alpha_t = \frac{\epsilon}{(c + \sum_{i=1}^{t-1} g_i^2/h_i^2)^p}$ for some $c \ge 1$ and $p \in [0, 1/2)$. Then:*

$$\sum_{t=1}^{T} \frac{\alpha_t g_t^2}{\sqrt{V_t}} \le \frac{2\epsilon h_T^{2p} \left(\sum_{t=1}^{T} g_t^2\right)^{1/2-p}}{1-2p}$$

*Proof.* Similar to the proof of Lemma 21, we have:

$$V_t \ge (c-1)h_t^2 + \sum_{j=1}^{t} g_j^2$$

$$\ge \sum_{j=\tau_i}^{t} g_j^2$$

and also:

$$\alpha_t \le \frac{\epsilon}{\left(\sum_{j=1}^{t} g_j^2/h_j^2\right)^p}$$

$$\le \frac{\epsilon h_t^{2p}}{\left(\sum_{j=1}^{t} g_j^2\right)^p}$$

Combining these yields:

$$\frac{\alpha_t g_t^2}{\sqrt{V_t}} \le \frac{\epsilon h_T^{2p} g_t^2}{\left(\sum_{j=1}^{t} g_j^2\right)^{1/2+p}}$$

Further, by [1] Lemma 4.13 we have:

$$\sum_{t=1}^{T} \frac{g_t^2}{\left(\sum_{j=1}^{t} g_j^2\right)^{1/2+p}} \le \int_0^{\sum_{t=1}^{T} g_t^2} \frac{dx}{x^{1/2+p}}$$

$$\le \frac{\left(\sum_{t=1}^{T} g_t^2\right)^{1/2-p}}{1/2-p}$$

from which the conclusion immediately follows. $\qquad\square$

## H  Regularized Regret via Full-Matrix Bound With Constraints

In this section, we provide an alternative approach to solving the "epigraph-based regularized regret" game specified by Protocol 3. Our approach actually involves a generic improvement to the class of so-called "full-matrix" regret bounds, and so may be of independent interest.

Specifically, we will provide an algorithm for online learning with "magnitude hints" (Protocol 4) that ensures the regret bound:

$$\sum_{t=1}^{T} \langle g_t, w_t - w_\star \rangle \le O\left(\epsilon h_{T+1} + \sqrt{d \sum_{t=1}^{T} \langle g_t, w_\star \rangle^2 \log(\|w_\star\| T/\epsilon)}\right). \tag{21}$$

This type of bound is sometimes called a "full-matrix" bound as the term inside the square root can be expressed as $w_\star^T \Sigma w_\star$ where $\Sigma$ is the matrix of gradient outer products $\Sigma = \sum_{t=1}^{T} g_t g_t^\top$. Bounds of this form have appeared before in the literature. For the case that $\mathcal{W}$ is an entire vector space, [16, 18] both provide full-matrix bounds. For the case in which $\mathcal{W}$ is *not* an entire vector space, [31] provides to our knowledge the only full-matrix bound. However, their algorithm suffers a suboptimal logarithmic factor: the $\log(T)$ term appears *outside* rather than inside the square root. We provide a method that fixes this issue.

However, before delving into the technical details of our approach, let us explain how achieving a full-matrix bound allows us to solve Protocol 3. The argument is nearly immediate: observe that in the 2-d game, we would have $w_\star \mapsto (w_\star, \psi(w_\star))$ and $g_t \mapsto (\tilde{g}_t, a_t)$. Then the bound (6) is immediate from (21). So, without further ado, let us provide our bound and analysis.

## H.1 Full Matrix Algorithm and Analysis

Assume that $\mathcal{W} \subset \mathbb{R}^d$ is a closed convex set that contains the origin within its interior. Further, let $\Phi_{\mathtt{bar}}$ be a self-concordant barrier for $\mathcal{W}$ with parameter $\mu > 0$. In this section, we present an algorithm that achieves (21). The algorithm is Follow-The-Regularized-Leader (FTRL) with a specific regularizer we define next.

**Regularizers.** For $\Sigma \in \mathbb{R}^{d \times d}$ and $Z, \sigma, \varepsilon > 0$, define the regularizer:

$$\Phi(w; \Sigma, Z, \sigma, \varepsilon) = \sup_{\lambda \geq 0} \sqrt{w^\top (\Sigma + \lambda I) w} \cdot X\left( w^\top (\Sigma + \lambda I) w e^{-\lambda Z} \cdot \frac{\det(\sigma^{-2}\Sigma)}{\varepsilon^2} \right), \qquad (22)$$

where $X(\theta) \coloneqq W(\theta)^{1/2} - W(\theta)^{-1/2}$ and $W$ is the Lambert function; $W(x)$ is defined as the principal solution to $W(x)e^{W(x)} = x$.

**Lemma 23.** *For any $\Sigma \in \mathbb{R}^{d \times d}$ and $Z, \varepsilon, \sigma > 0$, the Fenchel dual of the function $\Phi(\cdot; \Sigma, Z, \sigma, \varepsilon)$ in* (22) *satisfies:*

$$\forall G \in \mathbb{R}^d, \quad \Phi^\star(G; \Sigma, Z, \sigma, \varepsilon) = \inf_{\lambda \geq 0} \frac{\varepsilon \cdot \exp\left( \frac{1}{2} G^\top (\Sigma + \lambda I)^{-1} G + \frac{\lambda Z}{2} \right)}{\sqrt{\det(\sigma^{-2}\Sigma)}}.$$

*Proof.* See [18]. $\qquad \square$

**FTRL.** We will consider the FTRL algorithm with regularizer $\Phi(\cdot; \Sigma, Z, \sigma, \varepsilon) + \Phi_{\mathtt{bar}}(\cdot)$, for some choices of $\Sigma$, $Z$, $\sigma$, and $\varepsilon$. To specify these choices, let

$$\rho(\gamma) = \sqrt{2} \cdot \left( 1 - e^{\frac{1}{2\gamma} - \frac{1}{2}} \right), \qquad (23)$$

for $\gamma > 1$. With this, and given the history of gradients $g_1, \ldots, g_{t-1}$ up to round $t - 1$ and parameters $\gamma, \sigma, \varepsilon > 0$ and hint $h_t > 0$, the algorithm outputs:

$$\widehat{w}_t \in \operatorname*{argmin}_{w \in \mathbb{R}^d} \langle G_{t-1}, w \rangle + \Psi(-w; V_{t-1}, h_t, \sigma, \varepsilon), \qquad (24)$$

where

$$\Psi(w; V, h, \sigma, \varepsilon) \coloneqq \Phi(w; \sigma^2 I + \gamma V, \rho(\gamma)^2 / h^2, \sigma, \varepsilon) + \Phi_{\mathtt{bar}}(-w), \qquad (25)$$

and

$$G_\tau \coloneqq \sum_{s=1}^{\tau} g_s, \quad \text{and} \quad V_\tau \coloneqq \sum_{s=1}^{\tau} g_s g_s^\top. \qquad (26)$$

**Remark 1** (Connection to `Matrix-FreeGrad`)**.** We note that without the barrier term $\Phi_{\mathtt{bar}}$ in (25), the iterates in (24) can be computed in closed-form; in this case, the iterates exactly matches those of the `Matrix-FreeGrad` algorithm by [18] for *unconstrained* Online Convex Optimization (the connection to FTRL was not made explicit in [18]). The advantage of adding a barrier $\Phi_{\mathtt{bar}}$ is that it ensure that the iterates $(\widehat{w}_t)$ are always in the feasible set without requiring any sophisticated constrained-to-unconstrained reductions that may lead to sub-optimal logarithmic terms in the regret [46] (see Remark 2 in the sequel).

**Lemma 24** (Monotocity of potential)**.** *Let $\sigma, \varepsilon > 0$ and $\gamma > 1$ be given. For all $g_t \in \mathbb{R}^d$ and $h_t > 0$ such that $\|g_t\| \leq h_t$, we have*

$$\langle g_t, \widehat{w}_t \rangle \leq \Psi^\star(G_{t-1}; V_{t-1}, h_t, \sigma, \varepsilon) - \Psi^\star(G_t; V_t, h_t, \sigma, \varepsilon). \qquad (27)$$

*where $G \mapsto \Psi^\star(G; V, h, \sigma, \varepsilon)$ denotes the Fenchel dual of $w \mapsto \Psi(w; V, h, \sigma, \varepsilon)$.*

The proof of the lemma is in Appendix H.3. By summing (27) over $t$ and using Fenchel duality, we obtain the following regret bound for the FTRL iterates in (24).

**Theorem 25** (Regret with valid hints)**.** *Let $\sigma, \varepsilon > 0$ and $\gamma > 1$ be given. The FTRL iterates $(\widehat{w}_t)$ in* (24) *in response to any sequence $(g_t)$ such that $\|g_t\| \leq h_t$, for all $t \geq 1$, satisfy: for all $T \in \mathbb{N}$ and $w \in \operatorname{int} \mathcal{W}$:*

$$\sum_{t=1}^{T} \langle g_t, w_t - w \rangle \leq \varepsilon + \Phi_{\mathtt{bar}}^\star(0) + \Phi_{\mathtt{bar}}(w) + \sqrt{Q_T^w \ln_+\left( \det(\sigma^{-2}\Sigma_T) \cdot Q_T^w \right)}, \qquad (28)$$

*where* $\ln_+(\cdot) \coloneqq 0 \vee \ln(\cdot)$, $\Sigma_T = \sigma^2 I + \gamma V_T$, *and*

$$Q_T^w \coloneqq \max\left\{ w^\top \Sigma_T w, \frac{1}{2}\left( \frac{h_T^2 \|w\|^2}{\rho(\gamma)^2} \ln\left( \det(\sigma^{-2}\Sigma_T) \frac{h_T^2 \|w\|^2}{\varepsilon^2 \rho(\gamma)^2} \right) + w^\top \Sigma_T w \right) \right\}. \tag{29}$$

**Remark 2** (Comparison to previous "full-matrix" bounds in the constrained setting). We note that by having the $O(\log T)$ factor in (28) inside the square root, the bound in (28) improves on previous "full-matrix" bounds in the constraint setting [46], which have the log factor outside.

## H.2 Implementation Considerations

As stated in Remark 2, if we remove $\Phi_{\mathtt{bar}}$ from the regularizer, then iterates in (24) match those of `Matrix-FreeGrad`, which are available in *closed-form*. Unfortunately, in the presence of $\Phi_{\mathtt{bar}}$ (which ensures that the iterates are always in the feasible set $\mathcal{W}$), the iterate $\widehat{w}_t$ in (24) no longer admits a closed-form expression, and computing $\widehat{w}_t$, for $t \in [T]$, now requires solving a convex optimization problem. This is not ideal from a computational perspective; most first-order OCO algorithms require only $O(d)$ operation per round. It might be possible (at least in the case where $\mathcal{W}$ is bounded) to efficiently approximate $(\widehat{w}_t)$ without solving an optimization problem at each step and without sacrificing the regret by much using *Newton steps* such as in the recent works of [47, 48, 49]. We leave this investigation for future work.

## H.3 Proof of Lemma 24

*Proof.* By Lemma 23, we have that for all $V$ and $h$, $\Psi^\star(\cdot; V, h, \sigma, \varepsilon)$ satisfies

$$\Psi^\star(G; V, h, \sigma, \varepsilon) = \inf_{u \in \mathbb{R}^d} \Phi^\star(G - u; \gamma V, \rho(\gamma)^2/h^2, \sigma, \varepsilon) + \Phi_{\mathtt{bar}}^\star(-u),$$

$$= \inf_{\lambda \geq 0, u \in \mathbb{R}^d} \frac{\varepsilon \cdot \exp\left( \frac{1}{2}(G-u)^\top (\sigma^2 I + \gamma V + \lambda I)^{-1}(G-u) + \frac{\lambda \rho(\gamma)^2}{2h^2} \right)}{\sqrt{\det(I + \sigma^{-2}\gamma V)}} + \Phi_{\mathtt{bar}}^\star(-u). \tag{30}$$

We will use this to prove (27).

Let $(\lambda_\star, u_\star) \in \mathbb{R}_{\geq 0} \times \mathbb{R}^d$ be the minimizer in the problem $\Psi^\star(G_{t-1}; V_{t-1}, h_t, \sigma, \varepsilon)$. With this notation, we have

$$\widehat{w}_t = \operatorname*{argmin}_{w \in \mathbb{R}^d} \langle G_{t-1}, w \rangle + \Psi(-w; V_{t-1}, h_t),$$

$$= \operatorname*{argmax}_{w \in \mathbb{R}^d} \langle G_{t-1}, -w \rangle - \Psi(-w; V_{t-1}, h_t),$$

$$= -\operatorname*{argmax}_{v \in \mathbb{R}^d} \left\{ \langle G_{t-1}, v \rangle - \Psi(v; V_{t-1}, h_t) \right\},$$

$$= -\nabla \Psi^\star(G_{t-1}; V_{t-1}, h_t, \sigma, \varepsilon),$$

and so by Lemma 26,

$$= -\left(\sigma^2 I + \gamma V_{t-1} + \lambda_\star I\right)^{-1}(G_{t-1} - u_\star) \cdot \Phi^\star(G_{t-1} - u_\star; \sigma^2 I + \gamma V_{t-1}, \rho(\gamma)^2/h_t^2, \sigma, \varepsilon). \tag{31}$$

Moving forward, we define

$$G_{t-1,\star} \coloneqq G_{t-1} - u_\star \quad \text{and} \quad G_{t,\star} \coloneqq G_t - u_\star.$$

To prove the lemmsa, it suffices to prove the stronger statement obtained by picking the sub-optimal choice $(\lambda, u) = (\lambda_\star, u_\star)$ for the problem $\Psi^\star(G_t, V_t, h_t, \sigma, \varepsilon)$; that is,

$$\langle \widehat{w}_t, g_t \rangle$$

$$\leq \frac{\varepsilon \cdot \exp\left( \frac{1}{2} G_{t-1,\star}^\top \left( \sigma^2 I + \gamma V_{t-1} + \lambda_\star I \right)^{-1} G_{t-1,\star} + \frac{\lambda_\star \rho(\gamma)^2}{2h_t^2} \right)}{\sqrt{\det(I + \sigma^{-2}\gamma V_{t-1})}} + \Phi_{\mathtt{bar}}^\star(-u_\star)$$

$$\quad - \frac{\varepsilon \cdot \exp\left( \frac{1}{2} G_{t,\star}^\top \left( \sigma^2 I + \gamma V_t + \lambda_\star I \right)^{-1} G_{t,\star} + \frac{\lambda_\star \rho(\gamma)^2}{2h_t^2} \right)}{\sqrt{\det(I + \sigma^{-2}\gamma V_t)}} - \Phi_{\mathtt{bar}}^\star(-u_\star),$$

$$= \Phi^\star(G_{t-1,\star}; \sigma^2 I + \gamma V_{t-1}, \rho(\gamma)^2/h_t^2, \sigma, \varepsilon) - \Phi^\star(G_{t,\star}; \sigma^2 I + \gamma V_t, \rho(\gamma)^2/h_t^2, \sigma, \varepsilon),$$

and so dividing by $\Phi^\star(G_{t-1,\star}; \sigma^2 I + \gamma V_{t-1}, \rho(\gamma)^2/h_t^2, \sigma, \varepsilon)$ and using (31), this becomes

$$
\begin{aligned}
& - g_t \cdot \left(\sigma^2 I + \gamma V_{t-1} + \lambda_\star I\right)^{-1} G_{t-1,\star} \\
& \leq 1 - \frac{\exp\left(\frac{1}{2} G_{t,\star}^\top (\sigma^2 I + \gamma V_t + \lambda_\star I)^{-1} G_{t,\star} + \frac{\lambda_\star \rho(\gamma)^2}{2h_t^2} - \frac{1}{2}\ln\det\left(I + \sigma^{-2}\gamma V_t\right)\right)}{\exp\left(\frac{1}{2} G_{t-1,\star}^\top (\sigma^2 I + \gamma V_{t-1} + \lambda_\star I)^{-1} G_{t-1,\star} + \frac{\lambda_\star \rho(\gamma)^2}{2h_t^2} - \frac{1}{2}\ln\det\left(I + \sigma^{-2}\gamma V_{t-1}\right)\right)}.
\end{aligned}
$$

Let us abbreviate $\Sigma = \sigma^2 I + \gamma V_{t-1} + \lambda_\star I$. The matrix determinant lemma and monotonicity of matrix inverse give

$$
\ln \frac{\det\left(I + \sigma^{-2}\gamma V_t\right)}{\det\left(I + \sigma^{-2}\gamma V_{t-1}\right)} \;=\; \ln\left(1 + \gamma g_t^\top\left(\sigma^2 I + \gamma V_{t-1}\right)^{-1} g_t\right) \;\geq\; \ln\left(1 + \gamma g_t^\top \Sigma^{-1} g_t\right).
$$

Then Sherman-Morrison gives

$$
G_{t,\star}^\top\left(\sigma^2 I + \gamma V_t + \lambda_\star I\right)^{-1} G_{t,\star} \;=\; G_{t,\star}^\top \Sigma^{-1} G_{t,\star} - \gamma\frac{(g_t^\top \Sigma^{-1} G_{t,\star})^2}{1 + \gamma g_t^\top \Sigma^{-1} g_t}
$$

and splitting off the last round $G_{t,\star} = G_{t-1,\star} + g_t$ gives

$$
G_{t,\star}^\top\left(\sigma^2 I + \gamma V_t + \lambda_\star I\right)^{-1} G_{t,\star} \;=\; G_{t-1,\star}^\top \Sigma^{-1} G_{t-1,\star} + \frac{2 G_{t-1,\star}^\top \Sigma^{-1} g_t + g_t^\top \Sigma^{-1} g_t - \gamma(g_t^\top \Sigma^{-1} G_{t-1,\star})^2}{1 + \gamma g_t^\top \Sigma^{-1} g_t}.
$$

All in all, it suffices to show

$$
-g_t^\top \Sigma^{-1} G_{t-1,\star} \;\leq\; 1 - \exp\left(\frac{2 G_{t-1,\star}^\top \Sigma^{-1} g_t + g_t^\top \Sigma^{-1} g_t - \gamma(g_t^\top \Sigma^{-1} G_{t-1,\star})^2}{2(1 + \gamma g_t^\top \Sigma^{-1} g_t)} - \frac{1}{2}\ln\left(1 + \gamma g_t^\top \Sigma^{-1} g_t\right)\right).
$$

Introducing scalars $r = g_t^\top \Sigma^{-1} G_{t-1,\star}$ and $z = g_t^\top \Sigma^{-1} g_t$, this simplifies to

$$
-r \;\leq\; 1 - \exp\left(\frac{2r + z - \gamma r^2}{2(1 + \gamma z)} - \frac{1}{2}\ln(1 + \gamma z)\right)
$$

Being a square, $z \geq 0$ is positive. In addition, optimality of $\lambda_\star$ ensures that $\left\|\Sigma^{-1} G_{t-1,\star}\right\| = \frac{\rho(\gamma)}{\sqrt{2}h_t}$; this follows from the fact that $\frac{d}{d\lambda}\left. G_{t-1,\star}^\top (\sigma^2 I + \gamma V + \lambda I)^{-1} G_{t-1,\star}\right|_{\lambda=\lambda_\star} = \|\Sigma^{-1} G_{t-1,\star}\|^2$. In combination with $\|g_t\| \leq h_t$, we find

$$
|r| \leq \rho(\gamma)/\sqrt{2} = 1 - e^{\frac{1}{2\gamma} - \frac{1}{2}} < 1. \tag{32}
$$

The above requirement may hence be further reorganized to

$$
2r - \gamma r^2 \;\leq\; -z + (1 + \gamma z)\left(\ln(1 + \gamma z) + 2\ln(1 + r)\right).
$$

The convex right hand side is minimized subject to $z \geq 0$ at

$$
z \;=\; \max\left\{0, \frac{e^{\frac{1}{\gamma} - 1 - 2\ln(1+r)} - 1}{\gamma}\right\}
$$

so it remains to show

$$
2r - \gamma r^2 \;\leq\; \begin{cases} \frac{1}{\gamma} - (1 + r)^{-2} e^{\frac{1}{\gamma} - 1}, & \text{if } \frac{1}{\gamma} - 1 \geq 2\ln(1 + r); \\ 2\ln(1 + r), & \text{otherwise.} \end{cases} \tag{33}
$$

Note that by (32), we have $2\log(1 + r) \geq \frac{1}{\gamma} - 1$, and so the condition in the previous display reduces to the second case; that is,

$$
2r - \gamma r^2 \leq 2\log(1 + r), \quad \forall |r| \leq 1 - e^{\frac{1}{2\gamma} - \frac{1}{2}}, \tag{34}
$$

which is satisfied for the hardest case, where $r = e^{\frac{1}{2\gamma} - \frac{1}{2}} - 1$. $\qquad\square$

## H.4 Proof of Theorem 25

*Proof.* Fix $w \in \mathbb{R}^d$. Using that $\Psi^\star(G; V, h, \sigma, \varepsilon)$ is decreasing in $h$, we can telescope (27) in Lemma 24 to obtain

$$\sum_{t=1}^{T} g_t^\top \widehat{w}_t \;\le\; \Psi^\star(0; 0, h_1, \sigma, \varepsilon) - \Psi^\star(G_T; V_T, h_T, \sigma, \varepsilon)$$

By (30), we have $\Psi^\star(0; 0, h_1, \sigma, \varepsilon) \le \varepsilon + \Phi_{\mathtt{bar}}^\star(\mathbf{0})$, yielding:

$$
\begin{aligned}
\sum_{t=1}^{T} g_t^\top \widehat{w}_t &\le \varepsilon + \Phi_{\mathtt{bar}}^\star(\mathbf{0}) - \Psi^\star(G_T; V_T, h_T, \sigma, \varepsilon), \\
&\le \varepsilon + \Phi_{\mathtt{bar}}^\star(\mathbf{0}) + \inf_{u \in \mathbb{R}^d} \langle G_T, u \rangle + \Psi(-u; V_T, h_T, \sigma, \varepsilon), \\
&= \varepsilon + \Phi_{\mathtt{bar}}^\star(\mathbf{0}) + \inf_{u \in \mathbb{R}^d} \langle G_T, u \rangle + \Phi(-u; \sigma^2 I + \gamma V_T, \rho(\gamma)^2/h_T^2, \sigma, \varepsilon) + \Phi_{\mathtt{bar}}(u), \\
&\le \varepsilon + \Phi_{\mathtt{bar}}^\star(\mathbf{0}) + \langle G_T, w \rangle + \Phi(-w; \sigma^2 I + \gamma V_T, \rho(\gamma)^2/h_T^2, \sigma, \varepsilon) + \Phi_{\mathtt{bar}}(w), \quad (\text{setting } u = w) \\
&= \varepsilon + \Phi_{\mathtt{bar}}^\star(\mathbf{0}) + \langle G_T, w \rangle + \sup_{\lambda \ge 0} \sqrt{w^\top(\Sigma_T + \lambda I)w} \cdot X\!\left( w^\top(\Sigma_T + \lambda I)w e^{-\lambda Z_T} \cdot \frac{\det(\sigma^{-2}\Sigma_T)}{\varepsilon^2} \right) \\
&\quad + \Phi_{\mathtt{bar}}(w), \tag{35}
\end{aligned}
$$

where $\Sigma_T := \sigma^2 I + \gamma V_T$ and $Z_T := \rho(\gamma)^2/h_T^2$. Zero derivative of the above objective for $\lambda$ occurs at

$$\lambda \;=\; \frac{\ln \frac{\|w\|^2}{Z_T}}{2 Z_T} - \frac{w^\top \Sigma_T w}{2 \|w\|^2},$$

and hence the optimum for $\lambda$ is either at that point or at zero, whichever is higher, with the crossover point at $\frac{\|w\|^2}{Z_T} \ln \frac{\|w\|^2}{Z_T} = w^\top \Sigma_T w$. Plugging that in, we find that for $C := \frac{\|w\|^2}{Z_T} \ln \frac{\|w\|^2}{Z_T}$, we have

$$
\begin{aligned}
&\sup_{\lambda \ge 0} \sqrt{w^\top(\Sigma_T + \lambda I)w} \cdot X\!\left( w^\top(\Sigma_T + \lambda I)w e^{-\lambda Z_T} \cdot \frac{\det(\sigma^{-2}\Sigma_T)}{\varepsilon^2} \right) \\
&= \begin{cases}
\sqrt{\tfrac{1}{2}(C + w^\top \Sigma_T w)} \cdot X\!\left( \tfrac{1}{2}(C + w^\top \Sigma_T w) e^{-\frac{\ln \frac{\|w\|^2}{Z_T}}{2} + \frac{Z_T w^\top \Sigma_T w}{2\|w\|^2}} \cdot \frac{\det(\sigma^{-2}\Sigma_T)}{\varepsilon^2} \right), & \text{if } C \ge w^\top \Sigma_T w; \\[2ex]
\sqrt{w^\top \Sigma_T w} \cdot X\!\left( w^\top \Sigma_T w \cdot \frac{\det(\sigma^{-2}\Sigma_T)}{\varepsilon^2} \right), & \text{otherwise.}
\end{cases} \\
&\le \sqrt{Q_T^w} \cdot X\!\left( \frac{\det(\sigma^{-2}\Sigma_T)}{\varepsilon^2} Q_T^w \right), \tag{36}
\end{aligned}
$$

where $Q_T^w := \max\left\{ w^\top \Sigma_T w, \tfrac{1}{2}\left( \frac{\|w\|^2}{Z_T} \ln \frac{\|w\|^2}{Z_T} + w^\top \Sigma_T w \right) \right\}$; in the last inequality, we used that $X(\theta)$ is increasing to drop the exponential in its argument. Combining (36) with (35) and using that $X(\theta) \le \sqrt{\ln_+(\theta)}$ (see Lemma 27), we obtain the desired bound. $\qquad\square$

## H.5 Helper Lemmas for Full-Matrix Analysis

**Lemma 26.** *Let $\mathcal{W} \subseteq \mathbb{R}^d$ and $\mathcal{Y} \subseteq \mathbb{R}$. Further, let $f : \mathcal{X} \times \mathcal{Y} \to \mathbb{R}$ be a differentiable function such that for all $x \in \mathcal{X}$, the problem $\inf_{y \in \mathcal{Y}} f(x, y)$ has a unique minimizer $y(x)$. Then,*

$$\nabla_x f(x, y(x)) = \partial_x f(x, y(x)). \tag{37}$$

**Lemma 27.** *For $\theta \ge 0$, define $X(\theta) := \sup_\alpha \alpha - e^{\frac{\alpha^2}{2} - \frac{1}{2}\ln \theta}$. Then $X(\theta) = (W(\theta))^{1/2} - (W(\theta))^{-1/2} = \sqrt{\ln \theta} + o(1)$.*

*Proof.* The fact that $X(\theta) = (W(\theta))^{1/2} - (W(\theta))^{-1/2}$ follows from [50, Lemma 18]. Recall that

$$\sup_x yx - e^x \;=\; y \ln y - y$$

Hence

$$
\begin{aligned}
X(\theta) &= \sup_{\alpha} \; \alpha - e^{\frac{\alpha^2}{2} - \frac{1}{2}\ln\theta} \\
&= \sup_{\alpha}\inf_{\eta} \; \alpha - \eta\left(\frac{\alpha^2}{2} - \frac{1}{2}\ln\theta\right) + \eta\ln\eta - \eta \\
&= \inf_{\eta} \; \frac{1}{2\eta} + \frac{\eta}{2}\ln\theta + \eta\ln\eta - \eta \\
&\le \; \min\left\{\sqrt{\ln\theta} - \frac{1 + \frac{1}{2}\ln\ln\theta}{\sqrt{\ln\theta}}, \; \frac{\sqrt{\theta}}{2} - \frac{1}{\sqrt{\theta}}\right\} \\
&\le \; \sqrt{\ln_+ \theta}
\end{aligned}
$$

where we plugged in the sub-optimal choices $\eta = \frac{1}{\sqrt{\ln\theta}}$ (this requires $\theta \ge 1$) and $\eta = \frac{1}{\sqrt{\theta}}$. When we stick in $\eta = \frac{1}{\sqrt{\ln(e^{e^{-2}}+\theta)}}$ we find

$$
X(\theta) \; \le \; \frac{\ln(e^{e^{-2}}+\theta) + \ln\theta - \ln\!\left(\ln(e^{e^{-2}}+\theta)\right) - 2}{2\sqrt{\ln(e^{e^{-2}}+\theta)}} \; \le \; \sqrt{\ln(e^{e^{-2}}+\theta)}
$$

Note that $e^{e^{-2}} = 1.14492$. This is less than 2, the value of $\theta$ where $\sqrt{\theta}/2 - 1/\sqrt{\theta}$ becomes positive. $\qquad\square$

# I   Complete Psuedocode for Regularized 1-Dimensional Learning

In Algorithm 6, we provide a self-contained implementation of an algorithm for regularized online learning (Protocol 2). The algorithm is obtained by combing Algorithm 3 with Algorithm 2.

## I.1   Efficient Projections for $\psi(z) = z^2$

Our algorithms for regularized online learning via epigraphs (Protocol 3) require projections to the set $\{y \ge \psi(x)\}$. While in general this projection may be expensive, for simple function $\psi$ of interest, such as $\psi(z) = z^2$, this projection is relatively straightforward. In the following we provide a formula for this projection that is easy to compute (if a little ungainly to look at).

**Proposition 28.** *Let $\psi : \mathbb{R} \to \mathbb{R}$ be given by $\psi(x) = x^2$. Define the norm $\|(x,y)\|^2 = hx^2 + \gamma^2 y^2$, the function $S(\hat{x},\hat{y}) = \inf_{y \ge \psi(x)} \|(x,y) - (\hat{x},\hat{y})\|$, and the projection $P(\hat{x},\hat{y}) = argmin_{y \ge \psi(x)} \|(x,y) - (\hat{x},\hat{y})\|$. Then for any $\hat{y} < \psi(\hat{x})$, we have $P(\hat{x},\hat{y}) = (x,y)$ with $y = x^2$ and:*

$$
x = \frac{2^{1/3}(G^2 - 2\gamma^2\hat{y})}{Z^{1/3}} - \frac{Z^{1/3}}{6 \cdot 2^{1/3}\gamma^2}
$$

*with*

$$
Z = -108G^2\gamma^4\hat{x} + 2\sqrt{2916G^2\gamma^8\hat{x}^2 + (6G^2\gamma^2 - 12\gamma^4\hat{y})^3}
$$

*Moreover,* $\nabla S(\hat{x},\hat{y}) = \left(\frac{G^2(\hat{x}-x)}{\sqrt{G^2(x-\hat{x})^2 + \gamma^2(\hat{}y)^2}}, \frac{\gamma^2(\hat{y}-y)}{\sqrt{G^2(x-\hat{x})^2 + \gamma^2(\hat{y}-y)^2}}\right)$

*Proof.* Since the $(x,y)$ is on the boundary of the constraint, we clearly have $y = x^2$. Note that $(x,y) = argmin_{y \ge \psi(x)} \|(x,y) - (\hat{x},\hat{y})\|^2$. Thus, by LaGrange multipliers, we have for some $\lambda$:

$$
2G^2(x - \hat{x}) = \lambda\psi'(x) = 2\lambda x
$$
$$
2\gamma^2(y - \hat{y}) = -\lambda
$$

**Algorithm 6** Regularized 1-dimensional learner (REG) for Protocol 2

**Input:** Non-negative convex function $\psi : \mathbb{R} \to \mathbb{R}$. Parameters $\gamma > 0$, $p \in [0, 1/2]$, $\epsilon_x > 0$ and $\epsilon_\psi > 0$

Initialize $k = 3$.

**if** $p = 1/2$ **then**
   Define constant $c = 3$
**else**
   Define constant $c = 1$
**end if**

**for** $t = 1 \ldots T$ **do**
   Receive $h_t \geq h_{t-1} \in \mathbb{R}$.
   Set $h_t^x = 3h_t$.
   Set $h_t^y = 3\gamma$
   Define $V_t^x = 9h_t^2 + \sum_{i=1}^{t-1}(g_i^x)^2$.
   Define $V_t^y = 9\gamma^2 + \sum_{t=1}^{t-1}(g_i^y)^2$
   **if** $p = 1/2$ **then**
     Set $\alpha_t^x = \dfrac{\epsilon}{\sqrt{c + \sum_{i=1}^{t-1}(g_i^x)^2/(h_i^x)^2}\,\log^2\!\big(c + \sum_{i=1}^{t-1}(g_i^x)^2/(h_i^x)^2\big)}$
     Set $\alpha_t^y = \dfrac{\psi(\epsilon)}{\sqrt{c + \sum_{i=1}^{t-1}(g_i^y)^2/(h_i^y)^2}\,\log^2\!\big(c + \sum_{i=1}^{t-1}(g_i^y)^2/(h_i^y)^2\big)}$
   **else**
     Define $\alpha_t^x = \dfrac{\epsilon}{\big(c + \sum_{i=1}^{t-1}(g_i^x)^2/(h_i^x)^2\big)^p}$
     Define $\alpha_t^y = \dfrac{\psi(\epsilon)}{\big(c + \sum_{i=1}^{t-1}(g_i^y)^2/(h_i^y)^2\big)^p}$
   **end if**
   Define $\Theta_t^x = \begin{cases} \dfrac{\left(\sum_{i=1}^{t-1} g_i^x\right)^2}{4k^2 V_t^x} & \text{if } \left|\sum_{i=1}^{t-1} g_i^x\right| \leq \dfrac{2kV_t^x}{h_t^x} \\[2ex] \dfrac{\left|\sum_{i=1}^{t-1} g_i^x\right|}{kh_t^x} - \dfrac{V_t^x}{(h_t^x)^2} & \text{otherwise} \end{cases}$
   Define $\Theta_t^y = \begin{cases} \dfrac{\left(\sum_{i=1}^{t-1} g_i^y\right)^2}{4k^2 V_t^\psi} & \text{if } \left|\sum_{i=1}^{t-1} g_i^y\right| \leq \dfrac{2kV_t^y}{h_t^y} \\[2ex] \dfrac{\left|\sum_{i=1}^{t-1} g_i^y\right|}{kh_t^y} - \dfrac{V_t^y}{(h_t^y)^2} & \text{otherwise} \end{cases}$
   Set $\hat{x}_t = -\text{sign}\left(\sum_{i=1}^{t-1} g_i^x\right)\alpha_t^x\left(\exp(\Theta_t^x) - 1\right)$
   Set $\hat{y}_t = -\text{sign}\left(\sum_{i=1}^{t-1} g_i^y\right)\alpha_t^x\left(\exp(\Theta_t^y) - 1\right)$
   Define the norm $\|(x, y)\|_t^2 = h_t^2 x^2 + \gamma^2 y^2$, with dual norm $\|(g, a)\|_{\star, t}^2 = \dfrac{g^2}{h_t^2} + \dfrac{a^2}{\gamma^2}$.
   Define $S_t(\hat{x}, \hat{y}) = \inf_{\hat{y} \geq \psi(\hat{x})} \|(x, y) - (\hat{x}, \hat{y})\|_t$
   Compute $x_t, y_t = \text{argmin}_{y \geq \psi(x)} \|(x_t, y_t) - (\hat{x}, \hat{y})\|_t$.
   Output $w_t = x_t$, receive feedback $g_t \in [-h_t, h_t]$, $a_t \in [0, \gamma]$, such that $a_t = 0$ unless $|g_t| = h_t$.
   Compute $(\delta_t^x, \delta_t^y) = \|g_t\|_{\star, t} \cdot \nabla S_t(\hat{x}_t, \hat{y}_t)$
   Set $g_t^x = g_t + \delta_t^x$.
   Set $g_t^y = a_t + \delta_t^y$.
**end for**

This implies:

$$x = \frac{G^2 \hat{x}}{G^2 - \lambda}$$

$$y = \hat{y} - \frac{\lambda}{2\gamma^2}$$

Moreover, we also must have $y = x^2$, so that:

$$\frac{G^4 \hat{x}^2}{(G^2 - \lambda)^2} = \hat{y} - \frac{G^2}{2\gamma^2} + \frac{G^2 - \lambda}{2\gamma^2}$$

$$\frac{(G^2 - \lambda)^3}{2\gamma^2} + \left( \hat{y} - \frac{G^2}{2\gamma^2} \right)(G^2 - \lambda)^2 - G^4 \hat{x}^2 = 0$$

This is clearly a cubic equation in $\lambda$, and so we can apply the cubic formula (via Mathematica) to obtain the following result:

$$\lambda = \frac{2G^2}{3} + \frac{2\gamma^2 \hat{y}}{3} - \frac{2^{5/3} G^2 \gamma^2 + 2^{11/3} G^2 \gamma^4 \hat{y} + 2^{11/3} \gamma^6 \hat{y}^2}{Z^{2/3}} - \frac{Z^{2/3}}{9 \cdot 2^{5/3} \gamma^2}$$

where

$$Z = -108 G^2 \gamma^4 \hat{x} + 2\sqrt{2916 G^2 \gamma^8 \hat{x}^2 + (6 G^2 \gamma^2 - 12 \gamma^4 \hat{y})^3}$$

which yields:

$$x = \frac{2^{1/3}(G^2 - 2\gamma^2 \hat{y})}{Z^{1/3}} - \frac{Z^{1/3}}{6 \cdot 2^{1/3} \gamma^2}$$

and $y = x^2$.

The expression for $\nabla S(\hat{x}, \hat{y})$ follows directly from [16] Theorem 4. $\qquad \square$

