# OpenReview forum: "Fully Unconstrained Online Learning"
_NeurIPS.cc/2024/Conference — NeurIPS 2024 poster_

### Official Review · Reviewer_itTv · 2024-06-25

**Soundness:** 3
**Presentation:** 4
**Contribution:** 3
**Rating:** 7
**Confidence:** 4

**Summary:**

The paper presents the first $\tilde{O}(G \lVert w_\star \rVert \sqrt{T} + \lVert w_\star \rVert^2 + G^2)$ guarantee for online convex optimization without assuming known-in-advanced bounds of either the gradients or comparator norms. This result matches the best possible rate given known bounds in the main regime of interest where sub-linear regret is possible (while $G \lVert w_\star \rVert$ are not overly small). Additional results include a generalization of the method to a frontier of bounds using parametric regularization and a complementary lower bound.

**Strengths:**

1. The paper unrolls a detailed discussion and comparison with previous work, providing an in-depth understanding of the problem, including both results and techniques.

2. The guarantee is new and possesses several properties that are more appealing than previous ''fully parameter-free'' results, including a nicer symmetry between $G$ and $\lVert w_\star \rVert$ without a dependence in $T$ on the excess terms.

**Weaknesses:**

3. As the authors mention, an ideal result would also include previous results that are incomparable with the new bound.

**Questions:**

4. Considering the stochastic case (with online-to-batch) with crude bounds of the comparator and stochastic gradient norms, [1] can obtain a price-of-adaptivity of $\widetilde O(R/\sqrt{T})$ instead of the $\widetilde O(\max\{L,R\}/\sqrt{T})$ mentioned in the discussion. Can such knowledge be used to obtain a similar result using the approach presented in the paper?

5. Again considering the stochastic case (with online-to-batch), recent results [2-4] with crude bounds depend on the noise bound (a stronger noise assumption) instead of the bound of the stochastic gradient norms. Is there a possibility for a better online-to-batch conversion which enjoys such refined guarantees using online parameter-free algorithms? Or, alternatively, a direct application of the techniques in the stochastic setting?

Overall, the paper presents a clear picture of the current state of ''fully parameter-free'' optimization for online learning and presents a new guarantee with desirable properties (and a more general frontier using parametric regularization), both of value to the online parameter-free literature.

[1] Ashok Cutkosky. “Artificial Constraints and Hints for Unbounded Online Learning”. In: Proceedings of the Thirty-Second Conference on Learning Theory. 2019, pp. 874–894.

[2] Attia, A. and Koren, T., 2024. How Free is Parameter-Free Stochastic Optimization?. arXiv preprint arXiv:2402.03126.

[3] Khaled, A. and Jin, C., 2024. Tuning-Free Stochastic Optimization. arXiv preprint arXiv:2402.07793.

[4] Kreisler, I., Ivgi, M., Hinder, O. and Carmon, Y., 2024. Accelerated Parameter-Free Stochastic Optimization. arXiv preprint arXiv:2404.00666.

**Limitations:**

Incomparable results with previous work, which are discussed in the paper.

---

> ### Author Rebuttal · Authors · 2024-08-05
>
> Thanks for your review!
>
> Q1: Yes, in a certain sense. The way to to obtain the improved PoA using [1] is simply to only do the clipping suggested by [1] and not the artificial constraints - instead just rely upon the coarse bound on $\|w_\star\|$. We could do the same thing by just replacing our regularization with a constraint to the coarse bound, which makes the two algorithms the same. That said, it would better to have a "cleaner" way to do this.
>
> Q2: Yes, our bound implies this sort of result immediately. In general, "variance of gradients"-style bounds for smooth losses are implied by any online algorithm whose regret bound depends on $\sum_{t=1}^T \|g_t\|^2$.
>
> To see this, let $L$ be the expected loss. Then we have $\sum_{t=1}^T \|g_t\|^2 \sim \sum_{t=1}^T \|\nabla L(w_t)\|^2 + T\sigma^2 \le \sum_{t=1}^T L(w_t) -L(w_\star) + T\sigma^2$. So, our regret bound looks like $\sum_{t=1}^T L(w_t) - L(w_\star) \le O(\sqrt{T\sigma^2 + \sum_{t=1}^T L(w_t) - L(w_\star)})$, which implies $\sum_{t=1}^T L(w_t) - L(w_\star) \le O(1+\sqrt{T\sigma^2})$. This is one motivation for why we wanted to achieve the $\sum_{t=1}^T \|g_t\|^2$ in the regret bound rather than $G\sum_{t=1}^T \|g_t\|$ as obtained by the previous work.

---

> > ### Comment · Reviewer_itTv · 2024-08-08
> >
> > I thank the authors for their response.
> >
> > I have read the reviews and responses and would like to keep my score.

---

### Official Review · Reviewer_1CHH · 2024-06-29

**Soundness:** 3
**Presentation:** 1
**Contribution:** 3
**Rating:** 6
**Confidence:** 2

**Summary:**

This paper studies online convex optimization without prior knowledge of the Lipschitz constant of the losses or constraints on the magnitude (in $\ell_2$-norm) of the competitor. They provide a regret bound that almost matches the optimal regret with knowledge of these quantities (with some additive terms independent of the time-horizon), which improve on similar bounds from prior work. They also provide matching lower-bounds.

**Strengths:**

- The regret bounds significantly improve upon prior work in certain settings, e.g. when $||g_t||=G$ for all $t$ or can have milder dependence on the $\gamma$-parameter.
- A lower-bound is provided to establish the tightness of their bounds.
- They exploit reductions from prior works that make some aspects of the analysis easier to present and follow. Their proposed regularisation of the losses  is an interesting technical contribution.

**Weaknesses:**

- The improvement upon prior work is only relevant under specific conditions.  In particular, if $||g_t||$ is not constant and many are small (or even 0) and only a few are large, it is unclear which bound is better. It would also be interesting to have a comparison of the bounds for the “optimal” value of $\gamma$ (which of course is not of practical use but may help provide some intuition).
- The focus is only on the constraints expressed w.r.t. the $\ell_2$-norm and it is unclear if these results extend to arbitrary norms.

The presentation and clarity of some parts of the paper: the introduction is rather clear but the links between the different parts of the rest of the paper could be improved:
- It could be made clearer when introducing protocol 2 that you will use the original problem to generate these values of $h_t$ and $a_t$ to feed to an algorithm solving protocol 2. As it stands, as a reader it is unclear how/why “nature” generates or reveals these things. I wonder if going down this line of using a reduction is really the best way to present it and if immediately explicitly giving what these quantities will be ($h_t, a_t$) is not a clearer (and simpler) way to do it. This is explained in lines 142-144 but something like this should be explained at least at the beginning of Section 3 or 3.1. The link between protocol 1 and 2 does not come across that clearly (and in particular that Protocol 2 is an easier problem). It is also not clear why (4) is our goal - I guess the point is that this reduction and achieving (4) implies (2) but this is not explicitly said when presenting Protocol 2. (again this is discussed somewhat in the proof of Theorem 1 (lines 139-140) but Protocol 2 is introduced beforehand and it would help to have some context).
- Similarly, in the introduction, your results are presented in (2) and in the main paper in Theorem 1 but the relation between the two is not mentioned. In particular, Theorem 1 is quite a complicated looking bound and so it would be nice to have the implication explicitly stated.
- The paper is presented as solving two main challenges, not knowing $||g_t||$ and $||w_\star||$ ahead of time. However, dealing with the former seems to be already well understood (e.g. lines 147-157) and it is really the latter that is the challenge in this paper (lines 107-108 - correct me if I have misunderstood). The distinction between these two is not clear from reading the introduction.
- I am confused by lines 109-111: if $a_t = 0$ then $\tilde{g}_t + a_t \nabla \psi(w_t) = \tilde{g_t}$ ?
- At the start of the proof of Theorem 1 (section 3.2), you define the notation REG for the algorithm solving protocol 2. But then in lines 186-188, I believe you want to show a regret bound for REG on protocol 2 (i.e get the bound (4)), why not use the notation you defined to link these different parts and provide better guidance on the steps of the proof? In this section, you also show a “weaker” version of (4), which to me seems stronger because the presence of $\gamma < 1$ means it is a smaller upper-bound than (4) - maybe I am misunderstanding your use of “weaker” (this is also the case in section 3.3 for (6) vs (7)) ? In any case, if the proven bound is (7) instead of (6) (or the one in line 188 with S instead of (4)) and this is a smaller upper-bound, why not only consider this one ? Is there a specific reason why the ones presented in (4) and (6) are more interesting ?
- Similarly to the comment on linking Theorem 1 to (2), it would help to have something like this for Theorems 2 and 3 (and 4). For example, does Theorem 2 also lead to (2) or if not how do they differ in these simplified forms ? Given the amount of terms in these theorems, it is hard to assess if Theorem 4 is tight compared to the other Theorems.
- It would also be useful to have pointers to where the proofs of some of the later Theorems could be found.
- It would be nice to have a dedicated related works section that contextualises your results in the context of other works. This is done im some parts of the paper but having a dedicated section solely to this would improve clarity.
- Calling the (sub)-gradients of the losses $g_t$ loss vectors can be a bit confusing at first (especially that $\partial \ell_t$ is not defined anywhere I believe).

Typos:
- Line 66: “in our bound (3)” - which should be (2).

**Questions:**

See some questions in the weaknesses section.
- What happens if the competitor $w_\star$ becomes very small (or even exactly 0), do the bounds still hold ? Similarly if the $g_t$’s are all $0$, then the bound (3) goes to 0 but your bound does not, is this a problem with your analysis or is there an easy fix ?
- The bounds apply to the linearised version of the regret, which usually means that properties of the losses such as strong convexity cannot be exploited to achieve better than $\sqrt{T}$ regret. How would your bounds / algorithm behave in such settings ?

**Limitations:**

Yes

---

> ### Author Rebuttal · Authors · 2024-08-05
>
> Thank you so much for your very detailed review and you careful comments on the presentation. We will work hard to incorporate your comments into the final version. In the interest of brevity, below we respond to just a few of your comments:
>
> * For the "optimal" value of $\gamma$ our bound is always better. This is because for the optimal $\gamma$ value the previous bound is $O\left[\|w_\star\|\sqrt{\sum_{t=1}^T \|g_t\|^2} + \|w_\star\| G^{2/3}\left(\sum_{t=1}^T \|g_t\|\right)^{1/3}\right]$ while ours is $\tilde O\left[\|w_\star\|\sqrt{\sum_{t=1}^T \|g_t\|^2} + G\|w_\star\|\right]$ where $G=\max_t\|g_t\|$. Thanks for suggesting this comparision!
> * We can actually deal with arbitrary Banach spaces (so any reasonable norm). This is because the reduction in Section 3 of https://arxiv.org/pdf/1802.06293 shows that to solve an online learning problem in a Banach space, you need only be able to solve it in 1-dimension. We focused on the familiar L2 case here to tame the complexity a little bit.
> * There are by now many algorithms that deal with either unknown $G$ or unknown $\|w_\star\|$ ahead of time - it is dealing with both at once that is somewhat more rare. That said, the unknown $\|w_\star\|$ case is arguably more challenging. Our particular approach works by improving algorithms that deal with known $\|w_\star\|$, which is why our exposition focuses on these problems.
> * Regarding Lines 109-110: That's right. In these lines we are suggesting a straw-man algorithm: replace $\tilde g_t$ with $\hat g_t= \tilde g_t + a_t \nabla \psi(w_t)$. Then we could try to solve the problem by considering the pure linear losses $\langle \hat g_t, w\rangle$, which would be equivalent to the case $a_t=0$.
> * The weaker bound (7) vs the bound (4). We believe (4) is actually smaller than (7), not the other way around. This is because (7) depends on $\gamma \sum_{t=1}^T a_t$ while (4) depends on $\sum_{t=1}^T a_t^2$, and $a_t\le \gamma$ for all $t$. We chose to present (4) first because it is the true "ideal" bound, and we can in fact achieve it with a more computationally expensive algorithm.
>
>
> Questions:
> 1. Yes, the bounds hold for all $w_\star$ simultaneously, including 0. In the case $w_\star=0$, our bound achieve *constant* regret, while the previous approach achieves $O(\sqrt{T})$ regret. For the case $G\to 0$, you're right our bound appears to not go to zero. However, we believe this is an artifact of analysis: notice that *all algorithms* achieve zero regret when $G\to 0$. In our particular case, the algorithm will set $w_t=0$ for all $t$, as one might intuitively expect.
> 2. The unconstrained strongly convex setting is perhaps understudied in online learning: even the standard $G^2\log(T)$ bound doesn't really make sense since $G$ is unbounded in the unconstrained setting. We would suffer from this same issue. It's an interesting open problem to deal with this!

---

> > ### Comment · Reviewer_1CHH · 2024-08-08
> >
> > Thanks to the authors for their response. Apologies, I had missed the square on the $a_t$ in (4) - this clarifies my query. I will keep my score but acknowledge that it could be raised to a 7 with an improved presentation of the final version.

---

> > > ### Author Response · Authors · 2024-08-09
> > > **presentation**
> > >
> > > Unfortunately, this year are not allowed to submit revisions during the review process.
> > > However, we overall agree with your suggestions regarding the presentation. In particular,  we will commit to expanding our brief overview of the sequence of reductions at the start of section 3 to introduce $h_t$ and $a_t$  earlier, as well a higher level motivation for the approach. In particular, we will explan how algorithms that typically require a bound on $G$ can usually be modified to work with a value of $G$ that is not static but actually increases over time, and how we approximate  such a scenario with $h_t$. Then we will discuss that the remaining regret from this approach can be "cancelled" by adding an  additional quadratic regularization given by $a_t$. We will motivate this technique with the following example: suppose that for all but the *very last iteration*, it holds that $\|g_t\|\le H$ for some known value $H$. Then on the first $T-1$ iterations we can run a known-$G$ algorithm with $G=H$  and obtain low regret. However, on the last round we may discover that $G\gg  H$.  The maximum extra regret we can experience in this last round $(G-H)\|w_T - u\|\le G\|w_T\|+G\|u\|\le \frac{(G-H)^2}{2a_t } + \frac{a_t\|w_T\|^2}{2} + (G-H)\|u\|$. So, our approach is to add quadratic regularization to offset the $\frac{a_t \|w_T\|^2}{2}$  at the cost of an additional $\frac{a_t\|u\|^2}{2}$ in the regret.
> > >
> > > Your suggestion to add  more discussion after Theorem 1 linking back to equation (2) is also a good one and we will implement it as well.

---

### Official Review · Reviewer_Wuav · 2024-07-11

**Soundness:** 3
**Presentation:** 3
**Contribution:** 3
**Rating:** 6
**Confidence:** 3

**Summary:**

The authors propose a new algorithm for parameter-free online convex optimization without knowing the Lipschitzness of losses. Here, parameter-free online learning refers to a framework that achieves the optimal regret upper bound without knowing the magnitude of an (optimal) comparator.
The new parameter-free bound in the paper achieves a regret upper bound very close to the optimal regret upper bound $\|| w_* \|| G \sqrt{T}$ obtained when the magnitude of the comparator $\|| w_* \||$ and the Lipschitz continuity $G$ of losses are given before the game starts. The resulting regret upper bound is the first second-order bound in parameter-free learning without knowing the Lipschitzness of losses, and the dependency on the user-defined tradeoff parameter $\gamma$ is improved. To achieve this upper bound, the authors consider sequentially reducing general online convex optimization to one-dimensional online convex optimization, to a regularized online learning with magnitude hints, and to an epigraph-based regularized online learning setting, and derive the desired regret upper bound by effectively combining the latest techniques in existing online convex optimization.

**Strengths:**

The authors make a solid contribution to parameter-free online learning under the assumption that both the magnitude of the comparator $\|| w_* \||$ and the Lipschitz continuity $G$ of the loss are unknown. In particular, the second-order bound, which is one of the main differences from existing bounds, is an important contribution.

The quality of the paper is high. Although the presentation is very dense, it is clear, and the flow of the sequence of reductions and their motivations are explained in a comprehensive manner. The proof sketch of Theorem 1 in Section 3.2 is clearly given while explaining the existing techniques in online convex optimization. I have only partially reviewed the proofs in the appendix, but they appear to be correct.

**Weaknesses:**

A potential weakness of this paper is the density of the presentation.
It appears that the authors assume a significant level of knowledge from the readers about parameter-free online learning and general online convex optimization. The sequence of reductions is particularly dense and closely related to very recent research, which is not widely familiar within the community. It would be beneficial to provide more explanation especially concerning Epigraph-based Regularized Online Learning (in Protocol 3) (and Protocol 2 if possible).

The sequence of reductions begins with the reduction to one-dimensional OCO in Algorithm 1 in Appendix B. However, $w^{1d}_t$ does not seem to be defined in or around Algorithm 1. Can the authors provide an explanation for this?

Minor issues:
- Line 86: [15] Theorems 2 and 3 (similar typos are present in several places in the appendix)

**Questions:**

It is expected that the authors will address the questions mentioned in the Weakness section.

---

> ### Author Rebuttal · Authors · 2024-08-05
>
> Thanks for your work reviewing our paper!
>
> W1 (Regarding the density of the presentation): We chose a presentation intended to communicate all of the ideas, but it might become difficult to follow and we will work to make things clearer in the revision. For the epigraph-based learning, the idea is the following:
> Suppose you are interested in an online learning problem with losses of the form $\ell_t(w) = \langle g_t, w\rangle + a_t \psi(w)$ for some *known* function $\psi$. We are guaranteed $\|g_t\|\le 1$ and $a_t \in[0,1]$, but $\psi$ may not be Lipschitz. This means that we cannot apply the standard linearization trick $\ell_t(w)-\ell_t(w_\star) \le \langle \hat g_t, w-w_\star\rangle$ with $\hat g_t= g_t +a_t \nabla \psi(w)$ and then run an OLO algorihtm using $\hat g_t$ because $\hat g_t$ is not bounded. To fix this, we instead consider the 2-dimensional problem with parameter $(w_t, y_t)$ subject to the constraint $y_t \ge \psi(w_t)$ and *linear* losses $\hat \ell_t(w_t, y_t) = \langle g_t, w_t\rangle + a_t y_t$. This has the property that $\hat \ell(w_t, y_t) - \hat \ell_t(w_\star, \psi(w_\star)) \ge \ell_t(w_t) - \ell_t(w_\star)$, so it suffices to control the regret on the constrained linear problem.
>
> For protocol 2, the idea is that when one checks the analysis of essentially any algorithm that assumes a fixed Lipschitz constant, it is usually fairly easy to change the algorithm to allow for the Lipschitz constant to grow over time. Protocol 2 captures this by letting $h_t$ describe the growth in the Lipschitz constant. Of course, in general we do not know how it might grow over time, which is what the clipping technique in our final analysis addresses.
>
> W2 (definition of $w^{1d}_t$ in Algorithm 1): We apologize for the typo here: $w^{1d}$ is the same as $w^{\text{magnitude}}$. In the revision we will only mention $w^{\text{magnitude}}$ and convert all the $1D$ superscripts to $\text{magnitude}$. Intuitively, the idea is that the 1-dimensional learner $\mathcal{A}^{1D}$ is responsible for learning the magnitude of the comparison point $u$, while the direction $u/\|u\|$ is learned by a standard mirror descent algorithm in $w^{\text{direction}}$. See also the development in Sections 3 and 4 of https://arxiv.org/pdf/1912.13213, which is where we got these reductions.

---

> > ### Comment · Reviewer_Wuav · 2024-08-10
> >
> > Thank you for very much for the author’s response to the review. The authors' reply clearly addresses my questions. I will maintain my current positive score.

---

### Official Review · Reviewer_q7Dc · 2024-07-12

**Soundness:** 3
**Presentation:** 3
**Contribution:** 4
**Rating:** 7
**Confidence:** 3

**Summary:**

The authors consider the task of unbounded online convex optimization
without prior knowledge of the magnitude of the comparator nor the largest gradient.
In this context, they propose a parameter-free method whose regret against any arbitrary comparator
$w_*$ matches the optimal bound of $\mathcal{O}(G||w_{\*}||\sqrt{T})$ up to logarithmic factors when considering
$G$-Lipschitz losses. Their method has access to internally constructed hints at each time step -- which serve to control the gradients in the learning process, and regret bound. From prior works, learning with such hints is enough to derive reasonable regret bounds when the magnitude of the learners actions $||w_{t}||$ is small. To this end, the authors focus on controlling $||w_{t}||$ in the regret bound. They propose to optimize a regularized loss, which for a specific choice of the regularization parameter eliminates $||w_{t}||$ for the bound at the cost of additional factors of $||w_{\*}||$ and the largest hint. Finally, they implement this regularization trick with constraints and prove tightness of their result with lower-bounds.

**Strengths:**

1. This paper contributes to the budding line of work on near-optimal parameter-free online convex optimization. The authors build on the seminal work of Ashok Cutkosky on Artificial Constraints and Hints for Unbounded Online Learning, and address, in a unique way, a significant constraint in that work.
2. The paper is technically sound. The authors are honest about their methodology and adequately cite related works. Also, the submission was clearly written.
3. Overall, the authors address an important yet complex task by combining existing techniques in an unconventional way.

**Weaknesses:**

1. The authors discuss a number of preliminary ideas and independently interesting techniques which might be useful
	for some to understand their thought process, but distracting for others who are more interested in the main claims, method and result.

**Questions:**

I have some clarification questions, and comments below.
1. Just for clarification, can the authors explain why it is necessary to replace the regularization terms in the loss with constraints?
2. From the analysis of [1], it seems plausible that, for quadratic
	losses at least, regularization can be applied to completely eliminate the
	magnitude of the iterates in the regret bound at the cost of an additional
	factor of the magnitude of the comparator, but not the largest gradient.
        Actually, having the largest gradient in the upper-bound might not be good, for example in the case of quadratic losses, as the gradients may scale with $||w_{t}||$ as well. What are the authors thoughts on the dependence of $G$ in the regret bound especially for quadratic losses?
3. Can this method be directly extended to work with noisy gradients?

**Comments**
1. In Equation 1, the inequality should be an equality. Same holds for Equation 4, the expression in Theorem 1, e.t.c.
2. In line 66, do you mean "... in our bound (2)"?
3. Line 4 of Protocol 2 should read "loss vector $\tilde{g}\_{t}$" not
	"loss $\tilde{g}\_{t}$".
4. Line 231 should be "... the outputs
	$w_1,w_2,\cdots\in\mathbb{R}^d$..."
5. Line 104 should be "... but is not a major focus..."

[1] Neu, G., & Okolo, N. (2024). Dealing with unbounded gradients in stochastic saddle-point optimization. arXiv preprint arXiv:2402.13903.

**Limitations:**

Nil

---

> ### Author Rebuttal · Authors · 2024-08-05
>
> Thanks for your detailed review! Below we answer your questions.
>
> Q1: It is may not be strictly "necessary" to replace the regularization with a constraint - we suspect that in fact a suitable variation on e.g. FTRL based algorithms would also achieve the same goals. However, the constraint approach led to an algorithm that does not require solving a complex subproblem, and we also feel it is inherently interesting as it is a technique less commonly deployed in online optimization.
>
> Q2: We agree, the technique of [1] is essentially to apply an extra quadratic regularizer to an FTRL update. We believe an approach along these lines could achieve the same results we have for the quadratic case without using constraints. However, it would probably not remove the $G^2$ dependence on its own because the $G^2$ dependence arises from the calculuation that makes eliminating $\|w_t\|^2$ a useful thing to do.

---

> ### Comment · Reviewer_q7Dc · 2024-08-08
> **Response to Authors Rebuttal**
>
> I thank the authors for their response and am happy to keep my score.
>
> That said, I suggest the authors to conisder extending their method to work with quadratic losses, i.e getting rid of $G$ which can scale with $\|w_t\|$ while retaining the optimal $\mathcal{O}(\|w^*\|\sqrt{T})$ scaling in the bound, at least in cases where sub-linear regret is achievable. This would be an interesting future direction.

---

### Official Review · Reviewer_xTkL · 2024-07-14

**Soundness:** 3
**Presentation:** 2
**Contribution:** 3
**Rating:** 7
**Confidence:** 2

**Summary:**

This paper considers the fully unconstrained online convex optimization. At each round, the learner needs to choose a vector w_t and then observes the loss vector g_t and suffers the loss of the inner product of w_t and g_t. The goal is to design a learning algorithm to minimize the regret with respect to a comparator w*.

Previous work provides algorithms that require prior knowledge of either the length of w*, or the maximum length of loss vectors g_t. This work provides an online learning algorithm that achieves the regret bound of G^2+|w*|^2+ |w^*| G sqrt{T} where G is the maximum length of loss vectors.

**Strengths:**

1 This paper provides an algorithm that improved the regret bound for the fully unconstrained online convex optimization. It achieves the optimal bound for all sublinear regret regimes. The previous work for fully unconstrained online learning has a larger additive term G|w*|^3, which can not get the optimal bound for all sublinear regimes.

**Weaknesses:**

1 The paper introduces a lot of notations before defining them properly. It is not easy to read and have a general idea about the techniques.

2 The paper did not provide a detailed discussion about why the prior knowledge of the length of w* and the maximum length of the less vectors is unavailable in practice. It makes it hard for the reader to understand the merit of the new algorithm in this problem and the real-world applications. Especially, the bound in previous work for a fully unconstrained setting also gets the optimal bound for a wide-range regime. What is the regime in which the new bound is significantly better than the previous bound? How should I think about those regimes and connect them to some specific examples or applications?

-------------------
after the response:
They discussed the regimes where their bound is much better than the previous bounds. They also explained the unavailability of prior knowledge in practice.

**Questions:**

1 What is the regime that the new bound is significantly better than the previous bound? How should I think about those regimes and connect them to some specific examples or applications?

**Limitations:**

Yes

---

> ### Author Rebuttal · Authors · 2024-08-05
>
> Thanks for your work reviewing our paper! Regarding the unvailability of prior knowledge of $\|w_\star\|$ and $G$ in practice, let's think about the stochastic optimization setting. In this case, the unavailability of $G$ in practice is *exactly* the problem that popular methods like AdaGrad (the precurser to Adam) actually solve. However, these algorithms still require tuning a learning rate scalar, which roughly corresponds to selecting the value for $\|w_\star\|$. In fact, we actually have trouble coming up with many compelling examples in which $\|w_\star\|$ *is* available ahead of time.
>
> Regarding your question, the new bound is better in two important regimes:
> 1. When the $\|w_\star\|$ is large or $\gamma$ is small, so that the difference between the $\|w_\star\|^3/\gamma^2$ penalty incurred by the previous bounds is large compared to the $\|w_\star\|^2/\gamma$ penalty we incur. See also the response to review 1CHH, in which we observe that if both our bound and the previous bound are provided the optimal tuning for $\gamma$, our bound is always better.
> 2. For a more intuitive setting, consider a stochastic optimization problem in which the losses are *smooth* and $\ell_t(w_\star)=0$ for all $t$. In this case, by applying standard self-bounding arguments (e.g. see section 4.2 of [A modern Introduction to Online Leraning](https://arxiv.org/pdf/1912.13213)), we will actually obtain *constant* regret due to our dependence on $\sum_{t=1}^T \|g_t\|^2$, while the previous bound that depends on $\sum_{t=1}^T G \|g_t\|$ will at best obtain $O(T^{1/3})$ regret.

---

> > ### Comment · Reviewer_xTkL · 2024-08-12
> >
> > Thanks the authors for their detailed response, which answers my questions. I will raise my score.

---

### Decision · Program_Chairs · 2024-09-25

**Decision:**

Accept (poster)

**Comment:**

This paper studies online learning without knowing either an upper bound on the norm of the loss vector or the norm of the optimal solution and provides a protocol with regret that, in some regimes, can match the optimal regret bounds when these quantities are known. Though the reviewers agree that the theoretical contribution of the paper is qualitatively significant, there is also a majority consensus that the presentation of the paper needs attention.

I suggest the authors incorporate reviewer feedback to improve the overall presentation and strengthen the potential impact of the work.